



# Carbon Dioxide and Methane Measurements from the Los Angeles Megacity Carbon Project: 1. Calibration, Urban Enhancements, and Uncertainty Estimates

Kristal R. Verhulst[1], Anna Karion[2], Jooil Kim[3], Peter K. Salameh[3], Ralph F. Keeling[3], Sally Newman[4], John Miller[5,6], Christopher Sloop[7], Thomas Pongetti[1], Preeti Rao[1], Clare Wong[1,4,*], Francesca M. Hopkins[1], Vineet Yadav[1], Ray F. Weiss[3], Riley M. Duren[1], and Charles E. Miller[1]

[1]NASA Jet Propulsion Laboratory, California Institute of Technology, Pasadena, CA, USA
[2]National Institute of Standards and Technology (NIST), Gaithersburg, MD, USA
[3]Scripps Institution of Oceanography, University of California, San Diego, La Jolla, CA, USA
[4]Division of Geological and Planetary Sciences, California Institute of Technology, Pasadena, California, USA
[5]NOAA/ESRL/GMD, Boulder, CO, USA
[6]CIRES, University of Colorado, Boulder, CO, USA
[7]Earth Networks, Inc., Germantown, MD, USA
*Now at: California State University, Northridge, Northridge, California, USA

*Correspondence to*: K. R. Verhulst (Kristal.R.Verhulst@jpl.nasa.gov)

**Abstract.** We report continuous surface observations of carbon dioxide ($CO_2$) and methane ($CH_4$) from the Los Angeles
(LA) Megacity Carbon Project during 2015. We devised a calibration strategy, methods for selection of background air
masses, calculation of urban enhancements, and a detailed algorithm for estimating uncertainties in urban scale $CO_2$ and $CH_4$
measurements. These methods are essential for understanding carbon fluxes from the LA megacity and other complex urban
environments globally. We estimate background mole fractions entering LA using observations from four "extra-urban" sites
including: two "coastal/marine" sites, one "continental" site in the high desert northeast of LA, and one "continental/mid-
troposphere" site located in the San Gabriel Mountains. We find that a local marine background can be established to within
roughly 1 ppm $CO_2$ and 10 ppb $CH_4$ using these local measurement sites. We also show that continental sites may not be
relevant for selecting background observations during summer months due to the prevalence of onshore flow, which could
transport $CO_2$ and $CH_4$ from the LA Basin to relatively remote sites. Overall, atmospheric carbon dioxide and methane levels
are highly variable across Los Angeles. "Urban" and "suburban" sites show moderate to large $CO_2$ and $CH_4$ enhancements
relative to a marine background to estimate. An urban site near Downtown LA has a median enhancement of roughly 20
ppm $CO_2$ and 150 ppb $CH_4$ during all hours, and roughly 15 ppm $CO_2$ and 80 ppb $CH_4$ during midday hours (roughly 12-
16:00 LT, local time), which is the typical period of focus for flux inversions. The estimated measurement uncertainty is
typically better than 0.1 ppm $CO_2$ and 1 ppb $CH_4$ based on the repeated standard gas measurements from the LA sites during
the last 1-2 years, similar to Andrews et al. (2014). The largest component of the measurement uncertainty is due to the
observations being elevated relative to the single-point calibration method; however the uncertainty in the background mole
fraction is much larger than the measurement uncertainty. The approach to identifying background mole fractions described





here results in uncertainty ranging from roughly 5 and 15% of the enhancement near downtown LA for $CO_2$ and $CH_4$, respectively, during afternoon hours. Overall, analytical and background uncertainties are small relative to the local $CO_2$ and $CH_4$ enhancements, however, our results suggest that reducing the uncertainty to less than 5% of the enhancement will require detailed assessment of the impact of meteorology on background conditions.

**1 Introduction**

Improved understanding of carbon dioxide ($CO_2$) and methane ($CH_4$) emissions from cities has been identified as a priority for both carbon cycle science and to support climate mitigation efforts (Hutyra et al., 2014; Pacala et al., 2011). More than half of the global population currently resides within cities, with the fraction living in urban areas projected to increase in the future (United Nations, 2014). Currently, more than 70% of anthropogenic greenhouse gases (GHG) are

emitted from cities globally (IEA, 2008). The combination of carefully designed urban-scale atmospheric $CO_2$ and $CH_4$ monitoring networks, tracer transport modelling, and functionally resolved emissions data sets has the potential to offer significant advances in understanding and managing urban carbon emissions (Duren and Miller, 2012).

Carbon fluxes can be estimated using "top-down" or "bottom-up" methods. Both approaches are complementary to one another and can be beneficial for informing policy. Top-down approaches typically attempt to estimate carbon sources

and sinks from measured patterns of variability based on atmospheric observations. By contrast, bottom-up methods require an investigation of local processes and/or construction of models, such as combining fossil fuel usage data from each source sector with estimates of the carbon content of the fuel type (Gurney et al., 2009, 2012). An integrated top-down approach can be very useful, especially given the complex mixtures of anthropogenic and biogenic $CO_2$ and $CH_4$ sources found in urban ecosystems, which may be difficult to quantify using bottom-up methods (Duren and Miller, 2012; Hutyra et al.,

2014). Top-down measurements are also advantageous in that they can be reported with fully traceable and rigorously defined uncertainties. In this way, measurement records with both high precision and long-term stability are crucial to the objective evaluation of reported emissions at local, regional, and continental scales (roughly $10^2$-$10^6$ km$^2$; e.g. Andrews et al., 2014).

In recent years, there has been growing international interest in using top down atmospheric approaches to quantify

urban GHG fluxes (e.g. Duren and Miller, 2012; McKain et al., 2012, 2015). Large, organized urban greenhouse gas monitoring projects have emerged in many cities, including Paris ($CO_2$-Megaparis: http://co2-megaparis.lsce.ipsl.fr; e.g. Bréon et al., 2015; Xueref-Remy et al., 2016), Boston (McKain et al., 2015), Indianapolis (Influx: http://influx.psu.edu; e.g. Turnbull et al., 2015), Salt Lake City (http://lair.utah.edu/page/project/uta/pilot/; e.g. McKain et al., 2012), and, in this study, the Los Angeles Megacity (https://megacities.jpl.nasa.gov/portal/; see also Feng et al., 2016). To date, most of these

research efforts have been largely disconnected. More information flow between existing urban observational networks and the science and applications communities is needed to understand greenhouse gas emissions from cities. The data and





methods for greenhouse gas monitoring in urban regions should be fully disclosed and documented with a small degree of latency to make the best use of these atmospheric data for emissions verification and/or for informing policies more generally.

The Megacities Carbon Project was established through a multi-agency and multi-institution collaboration to
develop and demonstrate policy-relevant carbon monitoring in some of the world's largest and most complex cities, and to help address gaps in our knowledge of greenhouse gas emissions (Duren and Miller, 2012, and https://megacities.jpl.nasa.gov/portal/). The LA pilot project involves continuous and discrete flask sampling of air to monitor greenhouse and trace gas concentrations, together with isotopic ratios of $CO_2$ at multiple surface sites. This study describes the Los Angeles surface measurement network. The LA project has dramatically expanded the number of
greenhouse gas observing sites in the South Coast Air Basin since 2013, allowing unprecedented spatio-temporal measurement coverage in this region. In this study, we describe the Los Angeles Megacity surface network, sampling strategy, and calibration methods. We also discuss some preliminary results on $CO_2$ and $CH_4$ enhancements in the LA Basin and some detailed metrics for evaluating uncertainties in our observations.

The Los Angeles Megacity is home to >15 million residents and spans roughly 17,100 $km^2$ in California's South
Coast Air Basin (SCB, Figure 1). Observations from the LA network will also be useful for future assessment of GHG emissions in the South Coast Air Basin, which encompasses more than 43% of the CA statewide population. Policies and strategies for mitigation of $CO_2$ and $CH_4$ emissions are currently being implemented in California, with measures being passed at the state and local levels. The California Global Warming Solutions Act of 2006 (AB 32) requires California to reduce its GHG emissions to 1990 levels by 2020, a 15% reduction below emissions expected under a business-as-usual
scenario.

The SCB presents unique challenges in terms of the complexity of the land surface, meteorology, and spatial-temporal variability of its $CO_2$ and $CH_4$ emissions. Urban and suburban areas in the SCB have high population densities and a large variety of anthropogenic $CO_2$ and $CH_4$ emissions sources, as well as non-zero $CO_2$ fluxes expected from the terrestrial biosphere (Feng et al., 2016; Newman et al., 2013, 2016) and potential for $CH_4$ from natural geologic seeps (e.g.
Peischl et al., 2013). The SCB is bordered by the Pacific Ocean to the west and by mountains to the north and east. The mesoscale circulation patterns observed over the LA megacity are challenging to represent in atmospheric transport models (e.g. Angevine et al., 2012; Conil and Hall, 2006; Feng et al., 2016). Complex topography within the Basin can allow formation of micrometeorological zones, which may result in concomitant transport complexity. Prior studies suggest a dense measurement network with a high-degree of spatial and temporal resolution is required to provide robust, spatially-
resolved greenhouse gas flux estimates for the Los Angeles megacity (Kort et al., 2013).



Urban areas such as Los Angeles contain a complex mixture of sources. Urban $CO_2$ emissions can originate from both anthropogenic and biospheric processes. Urban anthropogenic $CO_2$ sources mainly reflect fossil fuel usage (including combustion of gasoline in cars and combustion of natural gas for electricity production, including seasonal cooling and heating), while biospheric $CO_2$ sources include photosynthesis and above- and below-ground respiration (Djuricin et al.,
2010; Hutyra et al., 2014). $CH_4$ can be produced via biogenic and thermogenic processes. Biogenic $CH_4$ is produced as a result of microbial decomposition of organic matter under anaerobic conditions (e.g. due to waste disposal in landfills and wastewater treatment plants), and is also produced via enteric fermentation in the gut of livestock and from manure. Thermogenic $CH_4$ is derived from natural geologic processes that produce fossil fuels, and therefore is naturally present in fossil fuel deposits including coal beds, oil fields, and geologic seeps (Etiope and Ciccioli, 2009). Thermogenic $CH_4$ can also
be emitted through intentional venting and fugitive leaks in the extraction, storage, refining, transport, and use of natural gas, as well as from incomplete combustion of fossil fuels.

In the LA Basin, many anthropogenic sources of $CO_2$ and $CH_4$ are co-located with each other and with potential natural sources. LA is a major industrial and shipping hub, with a dense network of roads and freeways for transport, the Port of Los Angeles, the Los Angeles International Airport, and also has extensive oil drilling infrastructure, with more than 10
local oil refineries and storage facilities. The LA Basin is also known for its naturally occurring geologic seeps, such as the La Brea Tar Pits. In addition to extensive natural gas pipeline networks, LA also has a variety of other $CH_4$ sources, including landfills, wastewater treatment plants, fossil fuel extraction and refining, natural gas storage facilities, compressor stations, and vehicle-fueling stations, and dairy agriculture, all of which can result in fugitive emissions (e.g. Hopkins et al., 2016; Peischl et al., 2013; Viatte et al., 2016; Wennberg et al., 2012). The complex mixture of sources and intense human
impacts of urbanization complicate $CO_2$ and $CH_4$ source attribution in the LA Basin.

Several previous efforts have been made to characterize $CO_2$ and $CH_4$ in LA using in situ and remote sensing observations. Some of the earliest published measurements of $CO_2$ in Los Angeles date back to the 1970s (Newman et al., 2008). Since then, there have been numerous studies investigating atmospheric $CO_2$ and $CH_4$ in the LA Basin using in situ observations, including continuous and flask-based sampling from Mt Wilson (MWO; Hsu et al., 2010; Wennberg et al.,
2012), Pasadena (CIT) and Palos Verdes Peninsula (PVP; Newman et al., 2008, 2013, 2016), and remote-sensing studies, including ground-based and space-based measurements (Kort et al., 2012; Viatte et al., 2016; Wong et al., 2015, 2016, Wunch et al., 2009, 2016). Periodic intensive field campaigns using aircraft have allowed brief "snap-shot" assessments (days to weeks in duration) of $CO_2$ and $CH_4$ levels and emissions in LA, including the campaigns ARCTAS-CA in 2008 (Jacob et al., 2010) and CalNex-LA in 2010 (Brioude et al., 2013; Cui et al., 2015; Peischl et al., 2013; Ryerson et al., 2013),
which were major field studies involving collaboration between the California Air Resources Board (ARB) and several partner agencies to improve the accuracy of emissions inventories for greenhouse gases and atmospheric pollutants, as well as a smaller, more recent campaign (Conley et al., 2016).





Prior studies have consistently reported robust enhancements of $CO_2$ (e.g. 30–100 ppm $CO_2$ at the surface and roughly 2–8 ppm $XCO_2$ column averaged dry-air mole fraction) and $CH_4$ (e.g. 10's to 100's of ppb $CH_4$ at the surface and roughly 0.2-50 ppb $XCH_4$ column averaged dry-air mole fraction), with significant temporal variability of the signals (Kort et al., 2012; Newman et al., 2013, 2016; Viatte et al., 2016; Wecht et al., 2014; Wennberg et al., 2012; Wong et al., 2015;

Wunch et al., 2009). For $CO_2$, radiocarbon ($^{14}C$) isotopic tracer measurements have also been made at a limited number of sites in Southern California (Djuricin et al., 2010, 2012, Newman et al., 2013, 2016; Riley et al., 2008). Djuricin et al. (2010) demonstrated that fossil fuel combustion contributed up to 50-70% to $CO_2$ sources during winter, while aboveground biological respiration was found to contribute more $CO_2$ than other sources during spring, when fossil fuel contributions were smaller. Recently, Newman et al. (2016) determined that fossil fuel combustion is the dominant source of $CO_2$ for

inland Pasadena using three-isotope approach, using $^{14}C$ along with $^{13}C$ and $^{18}O$ stable isotopes, similar to Djuricin et al., (2010). For $CH_4$, emissions estimates based on top down methods indicate that bottom-up methods systematically underestimate $CH_4$ emissions in the LA megacity by roughly 30% to >100% (Cui et al., 2015; Jeong et al., 2013; Peischl et al., 2013; Wecht et al., 2014; Wennberg et al., 2012; Wong et al., 2015, 2016; Wunch et al., 2009). Recent evidence from stable isotopes of $CH_4$ and light alkanes (e.g., ethane, propane, and butane) suggest that fossil emissions are the predominant

source of $CH_4$ (Hopkins et al., 2016; Peischl et al., 2013; Wennberg et al., 2012; Townsend-Small et al., 2012), particularly leakage from natural gas infrastructure and from local fossil $CH_4$ sources.

In contrast to some of these earlier studies, the monitoring network described here provides near-continuous and systematic monitoring of in situ $CO_2$ and $CH_4$ levels (as well as CO, which is not discussed in this work) at multiple sites in the LA metropolitan area. The LA network allows continuous spatial and temporal measurement coverage at multiple sites,

spanning multiple years, which can be used in future top-down atmospheric inversion studies. The first part of this study focuses on the sampling strategy and calibration method (Section 2). Next, we estimate hourly average $CO_2$ and $CH_4$ mole fractions (Section 3) and discuss observation-based selection criteria for determining the background $CO_2$ and $CH_4$ mole fractions using data from "extra-urban" sites (Section 4). One important result from this analysis is the near equivalence of continental and marine boundary layer background estimates for this region. We then use a marine background estimate to

calculate urban $CO_2$ and $CH_4$ enhancements from the LA surface network during afternoon hours, the typical period of focus for atmospheric flux inversions (Section 5). We also present a framework for estimating detailed time-dependent uncertainties in the enhancement based on the combined uncertainty in the air sample data collected from the measurement system and the background estimate (Section 6). We also compare data collected from analyzers in the field and independent data collected at the NOAA/ESRL and Scripps Institution of Oceanography laboratories to estimate measurement

uncertainties. In addition to providing a foundation for subsequent flux studies for LA, the sampling strategy, calibration methods, and uncertainty calculations described here are intended to be extensible to other surface observation networks in complex cities around the world.





## 2 Methods

### 2.1 Site selection criteria

The Los Angeles network design strategy began with a preliminary analysis based on a network receptor footprint sensitivity analysis using WRF-STILT (Kort et al., 2013) and Vulcan (Gurney et al., 2009, 2012), which found that a minimum of eight optimally located in-city surface observation sites were required for accurate monitoring of fossil fuel $CO_2$ emissions in the LA megacity. Such a network was estimated to distinguish fluxes to within approximately 12 g C m$^{-2}$ d$^{-1}$ (roughly 10% of average peak fossil $CO_2$ flux in the LA domain) on 8-week time scales and 10 km spatial scales (Kort et al., 2013). We initially assessed the logistics of deploying instruments at or near each of the locations specified by Kort et al. (2013). Site evaluation and siting criteria involved one or more of the following steps: (1) visual inspection of maps and satellite imagery to investigate whether suitably tall structures were available and to assess potential impacts of terrain and nearby strong greenhouse gas emission sources; (2) on-site surveys; (3) mobile measurement surveys in the region of interest (Hopkins et al., 2016); and/or (4) short-term deployment of a continuous CRDS analyzer on a short tower (approx. 10 m) for roughly 1-2 weeks prior to more permanent, fixed installation.

Where possible, measurement locations were sought on open-lattice communications towers. These structures were favored as they tend to reduce the influence of perturbed airflow from the supporting structure itself and remote locations minimize the influence of nearby emissions (Prasad et al., 2013). In the SCB, access to tall towers (>100 meters above ground level) was limited to the surrounding mountain ranges, which would present unique complexities for modelling and interpretation of the data. Therefore, towers within the Basin were limited to shorter cellular tower sites (<60 m), where available. Although there are a large number of shorter cellular towers in the SCB, these structures were often inaccessible due permitting or other restrictions. When no tower sites were available in a critical sampling area, we sought secure locations on the rooftops of tall, multi-story buildings in the area of interest. The siting criteria and sampling design framework were based on recommendations from Prasad et al. (2013) and McKain et al. (2015). In cases where rooftop sites were evaluated, Large Eddy Simulations were performed to study the impact of recirculation and nearby structures on the flow field around a building rooftop (Prasad et al., 2013).

### 2.2 Sampling locations

We established a network of eleven new surface observation sites distributed throughout three counties in the SCB (Figure 1). The geographic coordinates, inlet heights, species measured, and installation dates are summarized in Table 1. The tower sites include: Compton (COM), Granada Hills (GRA), Ontario (ONT), Victorville (VIC), and San Clemente Island (SCI). The building/rooftop sites are all located on university campuses in the following cities: Los Angeles (USC, University of Southern California), Pasadena (CIT, California Institute of Technology), Fullerton (FUL, California State University Fullerton), and Irvine (UCI, University of California, Irvine). The La Jolla site (LJO) is located on Scripps pier,





near a flask sampling location that has been discussed previously in the literature (e.g. Graven et al., 2012). The Palos Verdes Peninsula (PVP) and Pasadena (CIT) measurements have been described previously in the literature, but are not discussed in this study (Newman et al., 2013, 2016).

The measurement methods discussed below apply to the eleven new observation sites discussed here. All are equipped with similar instrumentation and use an internally consistent sampling protocol and calibration strategy (see Section 2.3-2.4). The LJO, SCI, VIC, and MWO sites are located outside the SCB boundary and are considered here as "extra-urban" sites, which can be used to estimate background or boundary condition for the SCB (Figure 1). We use an observation-based method to select background mole fractions from "extra-urban" sites, in part due to their remote locations (see Sections 3 and 4 for further discussion).

**2.3 Instrumentation**

The Los Angeles Megacity greenhouse gas-monitoring network utilizes wavelength-scanned cavity ring-down spectroscopy (CRDS) instruments (Picarro Inc., series G2301 and G2401; Rella et al., 2013; Welp et al., 2013) . All the CRDS instruments measure $CO_2$, $CH_4$, and water vapor, while sites with Picarro G2401 instruments also measure CO (Table 1). There are 3 standard configurations for the sites discussed in this study: 1) towers with a single inlet height, 2) towers

with multiple inlet heights, and 3) rooftop sites, which follow a 4-corner sampling strategy. Table 1 also indicates the site type, number of air inlets, and approximate heights for the air inlets. Air inlet heights vary from 13 to 100 meters above ground level (m agl) for tower sites, and from 20 to 55 m agl for the rooftop sites. Many of the measurement sites discussed in this study were installed, maintained, and/or operated by Earth Networks (EN, Germantown, MD, https://www.earthnetworks.com/).

The gas-handling configuration for the EN greenhouse gas monitoring stations is shown in the Appendix (Figure A1, adapted from Welp et al., 2013). The Earth Networks Sample Module houses a Valco 8-port low-pressure, dead-end flow path selector with standard bore size of 0.75 mm (VICI, Valco Instruments Co. Inc., http://vici.com/vval/sd.php), housed inside a heated box. The selector valve determines the sample type entering the CRDS cell (either outside air or standard/calibration gases).

All tower and rooftop sites are equipped with EN meteorological stations (Weatherbug, Inc., http://download.aws.com/manuals/RedBugBoxInstall.pdf), which measure wind speed, wind direction, ambient pressure, ambient temperature, humidity, dew point temperature, and incident solar radiation. Rain gauges are installed below the gas inlets. For tower sites, the wind measurements are co-located with the uppermost air inlet for the in situ greenhouse gas analyzers. For rooftops, the air inlets and wind sensors are installed on the four corners of the building, with masts typically

positioned roughly 3-5 m above the roofline and roughly 90 degrees from the walls or edge of the building's rooftop. Co-located meteorological measurements will allow better determination of the sensitivity of rooftop sites to local and regional



emissions (i.e. when the winds are stronger or more consistent), relative to potential emissions from the building itself (i.e. when the winds are calm).

The EN sample modules used in the LA surface network include a Nafion dryer housed in a thermostatic box (see Figure A1 and description by Welp et al., 2013). The drying system consists of a 183 cm (72-inch)-long Nafion membrane dryer (PermaPure, Inc., model MD-050-72S-1). An MKS640 pressure controller maintains a constant pressure to the Nafion dryer during routine sampling of ambient air and calibration gases (set point roughly 800 mb, 600 Torr). Both sample air and reference gases pass through a Nafion dryer before entering the CRDS cavity (see Appendix, Figure A1). The water vapor concentrations in the sample and standard gases are roughly $0.1\pm0.01\%$ $H_2O$ after passing through the Nafion dryer. The analyzer pump redirects roughly 30% of the dry gas exiting the Nafion to the outer shell side of the dryer. Welp et al. (2013) provide further discussion the design, testing, and implementation of this drying inlet system. Both the sample air and reference gases are delivered to the Nafion at the same pressure in order to reduce the drying bias due to permeation through the Nafion during routine operation, based on recommendations from Welp et al. (2013). The CRDS water vapor correction and uncertainty due to the treatment of water vapor are described in more detail in Section 6.

Before each analyzer was deployed, the Picarro factory default orifice (O'Keefe A-18-NY) was replaced with a smaller one (O'Keefe A-9-NY) to reduce the flow to about 70 sccm (cm$^3$/minute at STP). A second critical orifice (O'Keefe A-6-NY) was installed downstream of the Nafion to reduce the counterflow rate to about 30 sccm, and filters were added upstream of the critical orifice to prevent particles from disrupting the flow. A separate small pump (ALITA AL-6SA Air pump) module is installed for each air inlet and delivers a constant stream of sample air at 10 standard liters per minute (sL/min) to the EN sample module. The air inlets consist of 9.525 mm (3/8") Synflex tubing and an air intake filter consisting of either a stainless steel or titanium wire mesh screen (100 Mesh SS or Monel mesh).

The CRDS analyzers communicate data directly with a Linux mini-computer on-site that receives the data stream through a TCP connection. The site computer runs software (GCWerks, http://www.gcwerks.com), which controls the port sampling sequence in the EN sample module. The software acquires all the high-frequency data points from the CRDS (i.e. roughly 2.5 second time interval), EN sample module, and weather stations at each site, records extensive engineering data. GCWerks also sends out pre-programmed email alarms so that instrument issues can be diagnosed remotely. All high-resolution data (Level 0 data) are retained. The GCWerks software then applies some basic automated quality control flags and filters to the Level 0 data (the uncorrected, roughly 2.5 second resolution CRDS reading) and also rejects some data points to create higher-level data products (see Appendix A1 and Table A1).

**2.4 Calibration gases and sampling**

Each measurement site is equipped with two natural air standard gas tanks that are calibrated on the World Meteorological Organization (WMO) scales for $CO_2$ and $CH_4$. In the field, Parker Veriflo regulators (p/n: 45100653, Model:





95930S4PV3304) are used to deliver gas from the calibration tanks, and are connected to the Earth Networks sample module via 0.16 cm O.D. (1/16") SS tubing. Field standards are prepared by the National Oceanic and Atmospheric Administration Earth System Research Laboratory (NOAA/ESRL) and/or Scripps Institute of Oceanography (SIO) laboratory and are calibrated relative to WMO scales before and after deployment in the field. The NOAA/ESRL ambient-level standards are

natural air tanks filled at Niwot Ridge, Colorado and calibrated against standards on the WMO-scale maintained by NOAA/ESRL (X2007 for $CO_2$, X2004A for $CH_4$, http://www.esrl.noaa.gov/gmd/ccl/; (Dlugokencky, 2005; Zhao and Tans, 2006). For all standard tanks, we retrieve the most recent tank assignments from the NOAA Central Calibration Laboratory (http://www.esrl.noaa.gov/gmd/ccl/refgas.html). The SIO standards are filled using a similar procedure, except tanks are filled with natural coastal air from Scripps Pier in La Jolla, California. All mole fractions are reported in units of µmol gas

per mol dry air (ppm) or nmol gas per mol dry air (ppb). Both ambient-level tanks have mole fractions close to clean-air ambient conditions (roughly 400 ppm $CO_2$ and 1850 ppb $CH_4$). Our calibration strategy ensures compatibility within the LA surface network, and with other global atmospheric observations tied to the WMO scales.

The current calibration strategy for the LA surface network relies on a single-point calibration, tied to the WMO/NOAA scale. One of the near-ambient tanks is assigned as the calibration standard, and the other tank is a target

standard, which is treated as an unknown sample. This calibration framework has been used extensively for calibration of gas chromatography (GC-MS) instruments in remote monitoring networks, such as the ALE/GAGE/AGAGE network (e.g. Prinn et al., 2001). The details of the calibration gas composition will be discussed in a separate publication

The CRDS analyzer samples each standard tank approximately every 22 hours (i.e. approximately daily). The target tank measurement is staggered roughly 8-12 hours after the calibration gas (as well as the high mole fraction tank, where

applicable). All tanks are sampled for 20 minutes. The first 10 minutes of each tank run are rejected and only the data from the last 10 minutes of any are used in the calibration of $CO_2$ and $CH_4$ mole fractions (Welp et al., 2013). Variations in the measured target values and deviations from the assigned values are used to track the performance of the analyzer over time and determine uncertainties for the air data (Section 6.1).

The instrument sensitivity (S) is calculated for each standard tank (the calibration tank, the target tank, and the high

mole fraction tank) and is determined as the ratio between the uncorrected CRDS reading and the tank's assigned value on the WMO scales ($Xassign_{cal}$):

$$S = X'_{cal} / Xassign_{cal} \qquad (1)$$

where $X'_{cal}$ is the uncorrected CRDS reading (the dry mole fraction of the species of interest, in units ppm or ppb for $CO_2$ and $CH_4$, respectively). The sensitivity of the calibration tank is used to correct the air sample data, as described below.

Sensitivities for the target tank (and high-concentration tank, where available) are also tracked over time, however these





tanks are not used in the calibration of the air data.

The CRDS analyzer provides a nominal mole fraction value, which we take as an uncalibrated measurement. We then calibrate the uncorrected dry air sample mole fraction readings from the CRDS analyzer ($X'_{air}$) using the single point drift-correction method:

$Xcorr = X'_{air} * (Xassign_{cal} / X'_{cal})$            (2)

        $= X'_{air} / S$

where $Xcorr$ is the calibrated air data, $X'_{cal}$ is the dry mole fraction measurement of the calibration tank, and $Xassign_{cal}$ is the assigned value of the calibration standard on the WMO scales (which is constant in time). For each instrument, we interpolate the daily runs of the field calibration gas standard in time to provide a time stamp for $X'_{cal}$ at the time of the air

sample measurement. The units of $Xcorr$ are in ppm $CO_2$ or ppb $CH_4$.

In addition to the ambient-level calibration and target tanks, the VIC and LJO sites had high mole-fraction standard tanks installed at the time of this study. These tanks were prepared by NOAA/ESRL and calibration assignments were provided prior to deployment (roughly 500 ppm $CO_2$ and 2600 ppb $CH_4$). We treat the high mole-fraction tanks as an unknown target tank. The sensitivity (S) of the high mole fraction tank is also tracked over time, providing a check on the

analyzer stability at higher mole fractions. We use the high mole fraction tanks at these sites to estimate the uncertainty associated with our single-point calibration strategy by calculating the residual of repeated measurement of the high mole fraction tank from its assigned value. In Section 6.1 we discuss the individual components of uncertainty in the air measurements, including the extrapolation uncertainty, which is the uncertainty due to our assumption that S is not dependent on the mole fraction (see Section 6.1.1). In Appendix A2, we discuss an "Alternative calibration method" using

limited measurements of a high mole fraction tank installed at the La Jolla (LJO) and Victorville (VIC) sites in 2016.

### 3 Results

### 3.1 $CO_2$ and $CH_4$ observations

Atmospheric $CO_2$ and $CH_4$ mole fractions can vary on timescales ranging from less than 1 hour, to annual, and inter-annual cycles. Figure 2 shows the 1 hour average observations collected from nine sites in the surface network

between January 1, 2013 and December 31, 2015.

Generally, each site exhibits the expected seasonal cycle for $CO_2$ and $CH_4$, with wintertime maxima and summertime minima. The Downtown LA (USC), Compton (COM), and Fullerton (FUL) sites exhibit the highest average $CO_2$ levels during 2015 (Table 2). The annual average level was 421.6±17.5 ppm (USC, mean±1σ S.D.), 418.6±14.9 (FUL),



and 418.0±16.9 ppm $CO_2$ (COM) during afternoon hours (midday, roughly 12-16:00 LT or UTC-8, with no local adjustment for daylight savings time). On average, the same three sites showed the highest average $CH_4$ levels during 2015, in addition to the "suburban" Granada Hills site (GRA). During afternoon hours, the annual average $CH_4$ level was 2009.9±116.4 ppb (USC), 1985.6±130.5 ppb (GRA) 1978.2±100.2 ppb (FUL) and 1977.2±109.8 ppb $CH_4$ (COM). While USC exhibits the

highest levels of both gases, $CH_4$ exhibits a somewhat different spatial pattern relative to $CO_2$, with the GRA site showing the second largest $CH_4$ enhancements during midday. Overall the maximum 1 hour average measurement during afternoon hours was 558 ppm $CO_2$ and 3568 ppb $CH_4$ at the COM and GRA sites, respectively (Figure 2 and Tables 2 and 3).

Victorville and San Clemente Island (VIC and SCI) show less variability in their annual average $CO_2$ and $CH_4$ levels compared to the other sites that are within the South Coast Air Basin (Figure 1). During 2015, $CO_2$ levels at SCI

ranged from 391.2-425.2 ppm $CO_2$, with an average level of 402.4±4.4 ppm $CO_2$ during mid-afternoon hours. During mid-afternoon hours, $CH_4$ levels ranged from 1824.7-2231.4 ppm $CH_4$, with an average of 1900.9±37.9 ppb $CH_4$. At VIC, $CO_2$ levels at midday ranged from 395.9-442.6 ppm $CO_2$, with an average of 404.5±3.7 ppm $CO_2$, while $CH_4$ levels ranged from 1832.7-2105.3 ppb $CH_4$, with an average of 1898.6±32.9 ppb $CH_4$. We find that SCI and VIC are the cleanest sites in terms of their annual $CO_2$ and $CH_4$ variability. Feng et al. (2016) used a forward modelling framework to explore variability in

modelled $CO_2$ mole fractions during the CalNex period (May-June 2010). Their results based on modelled $CO_2$ "pseudo-data" are generally in agreement with the observations from these two "extra domain" sites. A third extra domain site is located outside the SCB boundary, at La Jolla (LJO). On average, LJO appears to have more variability and higher $CO_2$ and $CH_4$ levels compared to the SCI and VIC sites. The LJO site is outside the innermost model domain used Feng et al. (2016) and was not discussed in that study.

The annual average $CO_2$ variability observed at the IRV site is in the same range as other suburban sites, such as GRA and FUL, based on the 2015 observation record. This result is somewhat in contrast with a result presented by Feng et al., (2016), which showed IRV was a relatively clean site with respect to $CO_2$ using pseudo-data. Although both IRV and LJO are suburban sites and somewhat near the coast, on average, the $CO_2$ and $CH_4$ levels at LJO are typically lower than at IRV. We note that differing prevailing meteorological conditions during spring/summer months compared to the rest of the

year could influence the $CO_2$ and $CH_4$ observations, especially for coastal sites such as IRV and LJO. These sites typically exhibit less $CO_2$ and $CH_4$ variability during spring/summer when onshore flow may be stronger or more consistent. This may explain the difference between our results and those reported by Feng et al., (2016).

The heterogeneous mixture of sources in urban LA complicates sectoral attribution of $CO_2$ and $CH_4$ sources. The variability at each site is likely a reflection of the site's footprint, or its sensitivity to sources in the area. Measurement

footprints are typically variable and generally larger during the daytime than at night, and as such footprints are also more difficult to quantify during stable night-time conditions (Djuricin et al., 2010; Turnbull et al., 2015). Tables 2 and 3 show the median and interquartile ranges for the $CO_2$ and $CH_4$ observations, respectively. At most sites, the data distributions are skewed and have long-tails, where a relatively small fraction of observations exhibit significantly elevated $CO_2$ and/or $CH_4$



levels (see also Section 5, where we discuss the long-tail distribution with regards to the enhancement above background). Generally, high concentration spikes can occur at night and in the early morning, when the atmosphere is more stable, and when the site is more sensitive to nearby sources. One example is the suburban GRA site, which shows many high concentration $CH_4$ spikes since data collection began in 2013 (Figure 2, right panels, note scale difference on the y-axis).

Many of the $CH_4$ spikes throughout the GRA record occur at night, suggesting contributions from a nearby source that the other measurement sites are not sensitive to. In general, we do not expect the surface sites to be equally sensitive to $CO_2$ and $CH_4$, as the network design was only optimized for detection of fossil-fuel $CO_2$ emissions (Kort et al., 2013). Resolving the fine-scale structure of $CO_2$ and $CH_4$ emissions at the sectoral level will likely require footprint analysis and additional tracer measurements, which are planned as part of future work.

In addition to emissions, it has been demonstrated previously that meteorology plays an important role in controlling the variability of $CO_2$ (and $CH_4$) observations within the planetary boundary layer (e.g. Feng et al., 2016; Newman et al., 2013; Xueref-Remy et al., 2016). Diurnal variations are driven in part by changes in the height of the PBL. Newman et al. (2013) demonstrated this for $CO_2$ using observations from the CIT site. A stable PBL prevents surface emissions from mixing with the atmosphere above. Therefore, given a constant flux, the $CO_2$ and $CH_4$ mole fraction

observed within the PBL will increase or decrease as the PBL height falls or rises, respectively. Observations from midday hours show less variance in the within-hour $CO_2$ and $CH_4$ values and a smaller inter-quartile range relative to all hours (Table 2). The reduced variability in the $CO_2$ and $CH_4$ observations during midday hours is in part due to the stability of the PBL depth during the mid/late afternoon. Rahn and Mitchell (2016) evaluated Aircraft Meteorological Data Relay (AMDAR) automated weather reports from three major international airports in Southern California (LA, Ontario, and San

Diego) between 2001 and 2014. Overall, they found that PBL depth observations from LA (in the western LA Basin) showed the least variability (smallest interquartile range) during the hours just before sunset (~21:00 UTC to 03:00 UTC) indicating a fairly regular range of boundary layer height at this time (Rahn and Mitchell, 2016). $CO_2$ and $CH_4$ observations are also more likely to be sensitive to local sources when the PBL is shallow and the atmosphere is less well mixed (and at low wind speeds). The stability of the PBL height may also vary with season. Southern California is characterized by a well-

defined boundary layer during the spring and summer months due to strong temperature inversions associated with large-scale subsidence. During the autumn and winter, the large-scale subsidence is less prominent and the presence of a weak temperature inversion (or one that extends down to near the surface) makes it more difficult to identify a boundary layer (Rahn and Mitchell 2016). As part of future work, we plan to evaluate the diurnal and seasonal variability in the $CO_2$ and $CH_4$ signals with PBL depth measurements from a mini micropulse lidar instrument installed near the location of the CIT

measurement site (Ware et al., 2016).

Wind speed is also an important factor controlling variability in observed $CO_2$ (and $CH_4$) mole fractions, as has been demonstrated previously for $CO_2$ (e.g. Newman et al., 2013; Xueref-Remy et al., 2016). This is also related to the measurement footprint, as discussed earlier. For example, at low wind speeds, observations within the PBL are more likely



to reflect sources and sinks in close proximity to the site (with distances of roughly 10 km or less), while at higher wind speeds, the observation site will become more sensitive to transported emissions from more distant sources (d ~ 10-100 km), while the influences from nearby sources will appear more diluted. We do not go into further detail on the impacts of meteorology on the $CO_2$ and $CH_4$ signals as part of this analysis. Future work will explore the impacts of meteorology and

PBL height on the $CO_2$ and $CH_4$ (and CO) signals observed by the network using footprint analysis, and weather reanalysis products.

There are three potential signals of interest for urban and regional greenhouse gas studies. All may be potentially relevant for utilizing greenhouse gas measurements in local or regional inverse modelling studies: (1) diurnal changes in the measured mole fraction at one location over a 24-hour period; (2) gradients in the measured mole fraction between locations;

and (3) the local enhancement (referred to here as $\Delta CO_2$ and $\Delta CH_4$), which is the difference between an observed mole fraction at one location and an defined background mole fraction. In the remainder of this paper, we focus on the third type of signal discussed above, the enhancement above background.

### 3.2 Calculating $CO_2$ and $CH_4$ enhancements

The enhancement relative to the background mole fraction can be useful for evaluating local additions of $CO_2$ and

$CH_4$ from urban regions. We define the enhancement or excess signal ($\Delta X$) as follows:

$$\Delta X = X_{OBS} - X_{BG} \tag{3}$$

where $X_{OBS}$ is the calibrated $CO_2$ or $CH_4$ mixing ratio at the site of interest, $X_{BG}$ is the background mole fraction (i.e. the mole fraction from an air mass entering the domain or region of interest), all with units of ppm $CO_2$ or ppb $CH_4$.

### 4 Estimating background mole fractions

A critical goal for the LA Megacity Carbon Project is to identify an optimized background measurement location (or locations). Prior studies in the LA region have used either a coastal marine boundary layer background derived from observations from La Jolla, CA (32.87°N; 117.25°W, 0 m asl; Graven et al., 2012), or Palos Verdes Peninsula (33.74 °N; 118.35°W, 116 m asl; Newman et al., 2013, 2016), or a continental, free-troposphere background based on night-time flask measurements from the mountaintop site at Mt Wilson, CA in the San Gabriel mountains bordering the northern edge of the

LA Basin (MWO, 34.22°N; 118.06°W, 1670 m asl; Figure 1). Prior studies attempting to constrain $CH_4$ emissions in California have also estimated background mixing ratios along their model domain boundary using particle trajectory endpoints from WRF-STILT footprint simulations as a look-up for a latitudinally averaged, 3-D marine boundary layer (MBL) "curtain" product (Jeong et al., 2012, 2013; Zhao et al., 2009).

Evaluating the composition of a background air mass depends in part on the application. For example, in forward

and inverse modelling studies, the location and scale of the domain of interest will determine the background requirements.





A model that is used to estimate the enhancement due to local emissions should account for influences from sources both within and outside the domain of interest, as well as recirculation effects (i.e. when air exits the domain and returns a short while later). There is obviously no single background that is representative for all cases. There may also be cases when a single background site is not appropriate for estimating enhancements throughout the Basin. Out-of-domain sites may help

resolve within-domain emissions under some conditions, however the appropriate background site will also depend on the prevailing meteorological conditions. For Los Angeles, if the prevailing wind is from the land (offshore), then a continental background may be most appropriate, whereas if the wind is from the western coastal boundary (onshore), then a marine background may be most appropriate. Out-of-domain influences can also lead to spatial gradients that are independent of within-domain emissions, and will be more difficult to discern or characterize. In such cases, within domain sites may

occasionally be useful for characterizing background conditions.

In this study, the domain of interest is defined by the South Coast Air Basin boundary (Figure 1). The sites most suitable for characterizing background (or upwind) conditions are SCI, LJO, VIC, and MWO, which are all located outside this SCB domain. Overall, SCI, VIC, and LJO are most similar to the mole fractions of the remote MBL in terms of their annual average $CO_2$ and $CH_4$ mole fractions (Table 2). LJO is a coastal, suburban site in La Jolla, CA (as described above);

SCI is an offshore island site located on San Clemente Island, CA, just southwest of LA (32.92°N; 118.49°W, 480 m asl). VIC is a rural, desert site located outside the city of Victorville, CA (34.61°N; 117.29°W, 1370 m asl); and MWO is a mountaintop site, as described above. LJO and SCI are potentially useful for characterizing the Pacific marine boundary layer background values; VIC for characterizing a continental background; and MWO for characterizing a continental, mid-tropospheric background. At best, background conditions may only be observed intermittently from any of these sites

because each site can also be influenced by local and within-domain emissions under certain meteorological conditions. In Section 4.1, we use an observation-based method to select background observations at the LJO, SCI, VIC, and MWO sites and in Section 4.2 we compare these estimates. In Section 4.3, we discuss some air mass back trajectories and the implications for background estimates for the LJO, SCI, VIC, and MWO sites.

### 4.1 Background methods

Estimating greenhouse gas enhancements at the local scale requires measurements that resolve variability in background air masses (e.g. Graven et al., 2012; Turnbull et al., 2015). In the literature, several methods have been demonstrated for identifying background observations, including applying statistical filters to look for periods with stable measurements, filtering for meteorological conditions and/or chemical parameters, or using modelled and/or reanalysis products in combination with observations to estimate gradients (e.g. Alden et al., 2016; Ruckstuhl et al., 2012; Thoning et

al., 1989). Methods relying on chemical filtering techniques involve monitoring multiple species to identify pollution events or to inform about the sensitivity of a site to local pollution, while methods relying on meteorological filters assume some prior knowledge about the transport of polluted air masses to the site.





In this study, we used a data selection approach based on simple statistical filtering criteria, where steady $CO_2$ and $CH_4$ mole fractions are used as an indicator of background air. Using this approach, we aim to estimate a local continental and marine background that can be used to estimate $CO_2$ and $CH_4$ enhancements in Los Angeles with relatively low-latency (i.e. with reduced delays such that near-real time atmospheric monitoring of the enhancement signal will be possible). Our
data selection approach relies on several criteria: (1) a small degree of variability within a 1 hour period, and 2) small hour-to-hour variability, and (3) persistence of the first two conditions for several hours. Based on these criteria, we should be able to exclude observations that are impacted by local emissions or recirculation effects at the continuous observation sites. This data filtering approach does not rely on the availability of any other observations (i.e. winds, boundary layer height, etc.). In this sense, we consider this background selection algorithm to be very operational in that it can be used to estimate
background mole fractions in real time or near-real time.

        **LJO and SCI "Marine" Background and VIC – "Continental" Background Estimates**: The LJO, SCI, and VIC air observations were filtered according to the same criteria. Our data filtering criteria loosely follow the preliminary selection criteria discussed by Thoning et al. (1989) and were as follows: (1) Check for stability of the $CO_2$ and $CH_4$ observations within 1-hour and only retain measurements if the 1-hour SD is <0.3 ppm $CO_2$ and <5 ppb $CH_4$; (2) Check for
large hour-to-hour changes in $CO_2$ concentration and retain measurements if the hour-to-hour difference is less than 0.25 ppm $CO_2$ (no hour-to-hour criteria were used for $CH_4$); (3) Retain only those observations with six or more consecutive hours that meet criteria 1 and 2. After applying the selection criteria respective to each site, the CCGCRV curve fitting software was used to estimate a "smooth curve" fit to the remaining observations (Thoning et al., 1989; http://www.esrl.noaa.gov/gmd/ccgg/mbl/crvfit/crvfit.html). The curve-fitting parameters are given in Appendix A3. The full
time series, selected data and "smooth curve" results are shown in Figure 3 and the final smooth curve results are shown in Figure 4 (top panels).

        **MWO "Continental, Mid-Troposphere" Background Estimate:** Mt. Wilson (MWO) is a mountaintop observatory overlooking the South Coast Air Basin, approximately 1670 m agl (Figure 1). At night, the PBL is shallow and the MWO site is more likely to be influenced by air from the free-troposphere. During the daytime, the MWO $CO_2$ and $CH_4$
mole fractions can be influenced by emissions from the Basin either due to upslope winds or due to the rising of the PBL above MWO. Calibrated continuous in situ observations from Mt Wilson were not available at the time of this study. Instead, we used the MWO night-time flask record from NOAA/ESRL to produce a smooth curve background estimate using a similar approach to that described above for the SCI, LJO, and VIC sites. Flask samples have been collected at MWO approximately every 3-4 days since 2010. Only flask samples collected between 23:00 and 05:00 hours LST (local
standard time) were used in the smooth curve fit because only night-time samples are likely to be representative of background conditions. The curve fitting parameters are given in Appendix A3. The final smooth curve results are shown in Figure 4 (top panels).



**Pacific MBL Background:** The Pacific MBL reference surface was developed using weekly flask air samples from the NOAA's Global Greenhouse Gas Reference Network (see http://www.esrl.noaa.gov/gmd/ccgg/mbl/ and Masarie and Tans, 1995). The MBL reference surface is a data product smoothed in time and over latitude that uses NOAA measurements from samples that are predominantly influenced by well-mixed MBL air (typically remote, marine sea level locations with prevailing onshore winds). The Pacific MBL product provides a 2-D (latitude and time) representation of $CO_2$ and $CH_4$ mixing ratios along the Pacific boundary of North America based on the subset of GGGRN MBL sites in the Pacific Basin. We compare the results from SCI, LJO, VIC, and MWO to the Pacific MBL reference surface in Figure 4.

We note that the method used to estimate background would fail to give a measure of influences from outside the domain under some conditions. Below we compare the background estimates described above (Section 4.2) and discuss some meteorological considerations for background estimation (Section 4.3).

### 4.2 Comparison of background estimates

We compared the background estimates derived from the SCI, LJO, VIC, and MWO sites during 2014-2015 with the 2-D Pacific marine boundary layer (MBL) reference from roughly 33.4°, 36.9°, and 40.5° N (Figure 4). There are small but systematic differences in the background curves determined for each site.

For $CO_2$, the seasonal cycle at SCI and LJO is more similar to the Pacific MBL estimates than the MWO and VIC results. The SCI and LJO background estimates show more pronounced $CO_2$ minima in the summer relative to VIC and MWO, similar to the MBL estimate from 33.4° N. This suggests that under the appropriate filtering criteria, the LJO and SCI observations can be used to derive a marine background estimate for $CO_2$. During summer months, the background derived from VIC and MWO are differ from the MBL estimates by up to ~7 ppm $CO_2$.

Overall, the differences $CO_2$ background estimates from SCI, LJO, VIC, and MWO are less than ±6.5 ppm $CO_2$ in summer and ±3 ppm $CO_2$ in winter relative to the Pacific MBL estimate from 40.5° N. During summer 2015, the marine background estimates from LJO and SCI are slightly higher than the Pacific MBL estimate from 40.5° N, with a maximum $CO_2$ difference of roughly +3 and +5 ppm $CO_2$ for LJO and SCI, respectively (Figure 4). During winter 2015, the marine background estimates from LJO and SCI are slightly lower than the Pacific MBL estimate from 40.5° N (maximum differences of roughly -3 ppm $CO_2$). This is somewhat surprising given that there is more variability in the origin of the incoming air masses during winter months (Figure 5).

$CH_4$ background estimates from SCI, LJO, VIC, and MWO are less similar to one another during summer compared to other months (Figure 4). Overall, the differences from the Pacific MBL estimate from 40.5° N range from -20 and +60 ppb $CH_4$ during summer months and ±30 ppb $CH_4$ during all other months. The SCI and LJO background estimates are more similar to the Pacific MBL background during almost all times of year compared to VIC and MWO. For SCI, the





differences from the Pacific MBL estimate ranging from -31 to +2 ppb $CH_4$ during summer and from -35 to +10 ppb $CH_4$ during the rest of the year. During summer 2015, the $CH_4$ background derived from LJO is very different from the other marine background estimate. A landfill near the LJO site could influence the observations at this site. This hypothesis could be confirmed using footprint analysis. Further refinement of the data selection algorithm could also provide better agreement

between the $CH_4$ background estimates.

Our results show that during most of the year, the differences in background mole fractions estimated from each of the remote "extra domain" sites are small relative to the enhancement (discussed further in Section 5). The median differences between the Pacific MBL estimate from 40.5° N and the other background estimates from SCI, LJO, VIC, and MWO are: -0.8, -0.8, -0.5 and -1.3 ppm $CO_2$ and -3.5, +1.6, -7.8, and -11.5 ppb $CH_4$. As shown below in Sections 5 and 6,

this is ~15-17% of the median $CO_2$ enhancement and ~10-13% of the median $CH_4$ enhancement at the USC and FUL sites.

### 4.3 Back-trajectory analysis

Our approach for estimating background mole fractions thus far has ignored variations in atmospheric transport. In reality, winds transport air masses in and through the LA Basin on various timescales. Therefore, the optimal background site for selecting observations could vary diurnally, weekly, monthly, and/or seasonally. Wind back trajectories can be useful

for selecting a primary background site, based on the prevailing winds. We performed a simple back trajectory analysis and below discuss some preliminary conclusions based on that analysis. Results in Figure 5 are shown for 14:00 LST (local standard time), however, in general, the back trajectories computed for 12:00 and 16:00 LST show similar results.

We computed twenty-four hour back trajectories for winds arriving in Pasadena at 14:00 LST using NOAA's HYSPLIT model (Figure 5; Stein et al., 2015; Rolph, 2016). During the warmer months (spring/summer, or roughly May

through September), winds enter the Basin almost exclusively on-shore, originating over the ocean. These air masses generally travel south along the coast before being directed inland. Conversely, during the cooler months (fall/winter months, roughly November to March), there is much more variety in the provenance of the air masses (Figure 5). A significant fraction of days have off-shore winds (i.e. from the north to northeast, and originating from the Mojave desert region over the mountains), or could have Santa Ana-like conditions, which are a typical mode of variability for the Los

Angeles area during November to March (e.g. Conil and Hall, 2006). During offshore wind conditions, coastal sites such as La Jolla or San Clemente Island may not be relevant choices for selecting background observations as these sites may be subject to outflow and recirculation of an air mass from over land. Coastal ("Catalina") eddies are also common occurrence along the CA bight, which is the mostly convex part of the Southern California coastline (Figures 1 and 5). Conditions that favor coastal eddies are most common between April and September, though they develop at almost any time of the year

(Rahn and Mitchell, 2016). During such conditions, a site northwest of the Los Angeles Basin may be a more relevant choice for background. However, as we showed in Section 4.3, the MBL background derived using the SCI or LJO sites was very similar to the Pacific MBL reference surface between roughly 33-40° N.





At least some of the differences in our background estimates from the LA sites can be explained by differences in the prevailing meteorological conditions and a lag in the transport of air masses between the sites. The VIC and MWO sites show larger differences from the marine background estimates during summer months for both $CO_2$ and $CH_4$. For VIC, there is virtually no $CO_2$ or $CH_4$ data meeting the selection criteria during the summer and early fall months (Figure 3). The

back trajectory analysis showed that onshore flow conditions were more consistent from roughly May to September, during the same period when our background algorithm failed to find VIC observations meeting the stability criteria (Figures 3 and 4). The VIC inlet elevation (1370 m asl, 100 m agl inlet height) is only roughly 200 m lower than MWO (1670 m asl). The smooth curve estimate from VIC is similar to that of MWO, suggesting these two sites may be sensitive to similar air masses. Therefore, we conclude that the VIC and MWO sites may not be relevant choices for background during summer,

when onshore flow patterns dominate.

Our back trajectory analysis does not have the temporal resolution necessary to evaluate diurnal land-sea breezes. The spatial resolution of the NAM12 meteorological data used by HYSPLIT is 12 km. From this analysis, we can certainly see seasonal variations of the wind direction and the incoming air masses for the LA basin. We do not compare the day/night differences in meteorology, such as land/sea breezes, in our analysis, though we note that these circulation patterns

could be important for understanding the greenhouse gas variability (especially at coastal sites such as SCI, LJO, and possibly IRV). Such analysis would require a higher resolution model, such as the 1.3 km resolution WRF-Chem model discussed by Feng et al. (2016), which is beyond the scope of this study. Feng et al. (2016) found that sea breeze prevailed over the LA megacity at ~14:00 LST during the May/June 2010 (CalNex) study. Furthermore, the modelled topography of the Palos Verdes Peninsula was found to divide the sea breeze into west and southwest onshore flows that later converged in

the Central Basin. In general, transport models do not do well overnight ((Feng et al., 2016), which makes evaluation of diurnal variations challenging using modelled $CO_2$ or $CH_4$ output. Future modelling studies that overlap with the $CO_2$ and $CH_4$ records will be needed to evaluate the impact of land/sea-breezes on $CO_2$ and $CH_4$ observations from coastal sites and could also improve our understanding of the impacts winds induced by topography on the greenhouse gas observations.

Feng et al. (2016) used results from a forward model simulation to explore correlations in $CO_2$ concentrations in a

model framework. They showed that $CO_2$ is trapped and accumulates due to the mountain barrier, leading to $CO_2$ enhancements at in-basin sites relative to the desert site at VIC. Feng et al. also found that while the modelled $CO_2$ levels at the VIC desert site were mainly anti-correlated with the LA Basin sites, $CO_2$ that accumulated in the Basin could occasionally be pushed over the mountains and into the desert due to episodic strong sea breezes and onshore flow conditions. This supports our conclusions that VIC and MWO (night-time) observations may not always provide

representative background mole fractions, particularly during summer months when onshore flow conditions prevail. It is important to note that our approach for evaluating background mole fractions from MWO relied on night-time flask observations only, which were collected between 23:00 and 05:00 hours LST. Feng et al. refer to MWO as "western basin"





site, exhibiting spatial $CO_2$ correlations similar to the GRA, CIT, USC, and COM sites. Feng et al. (2016), but do not discuss day/night differences in the sensitivity of the MWO site. At night, we expect the PBL to be shallower, reducing the likelihood that air from the SCB will be transported to the MWO site. In future work, we plan to analyze continuous observations in conjunction with the night-time flask record from MWO to evaluate the diurnal variability in $CO_2$ and $CH_4$

observations at this site. While the simulations discussed by Feng et al., (2016) only cover a brief period during spring/summer 2010, future modelling studies over longer periods (e.g. one year) could improve our understanding of variations in the mesoscale circulation in the LA megacity and the impacts on the observed $CO_2$ and $CH_4$ mole fractions. The variety and complexity of meteorology in the South Coast Air Basin suggests that a more sophisticated background selection algorithm is needed to determine the site that is "upwind" during different prevailing wind conditions. Future

model analyses could also help determine when our observation sites are most relevant for estimating background.

Overall, the LJO and SCI background estimates establish a marine sector background to within roughly 1 ppm $CO_2$ and 10 ppb $CH_4$ (excluding the period during summer 2015 discussed above). SCI is the most representative of local marine background conditions for both $CO_2$ and $CH_4$ throughout the year. Therefore, we use SCI as the background reference site to calculate $CO_2$ and $CH_4$ enhancements for the LA surface sites (see below).

**5 $CO_2$ and $CH_4$ enhancements**

We calculated the average enhancement at each site using the SCI marine background reference. Moderate to large $CO_2$ and $CH_4$ enhancements ($\Delta CO_2$ and $\Delta CH_4$) are observed above the background mole fractions. Tables 4 and 5 show statistics regarding the enhancement at each site estimated for all hours and midday hours (12:00-16:00 LT, not including adjustment for daylight savings time) during 2015. Figure 6 shows the $\Delta CO_2$ and $\Delta CH_4$ values at 9 sites for all hours and

midday hours, with sites arranged by latitude. We do not discuss the results from the Ontario site (ONT) in detail because measurements were only available from Sept-Dec 2015 and therefore are not representative of the annual average.

The median enhancement during all hours was 21.4, 18.8, 16.3, 15.1, 10.7 ppm $\Delta CO_2$ and 121.0, 148.0, 106.1, 100.6, and 74.1 ppb $CH_4$ during 2015 at the USC, FUL, COM, GRA and IRV sites, respectively. During midday hours, the period that is most relevant for flux inversions, the median enhancement was 13.8, 12.2 10.1, 10.4, 5.8 ppm $\Delta CO_2$ for the

USC, FUL, COM, GRA and IRV sites, respectively (Figure 6 and Table 4).

The median $CH_4$ enhancement was 148, 106, 101, 121, and 75 ppb $\Delta CH_4$ for the USC, FUL, COM, GRA and IRV sites, respectively. During midday hours, the median enhancement during 2015 was 81.4, 58.6, 52, 70.4, and 40.2 ppb $\Delta CH_4$ at the USC, FUL, COM, GRA and IRV sites, respectively (Figure 6 and Table 5).

Overall, the results suggest that the $CO_2$ and $CH_4$ enhancements are characterized by a large degree of spatial and





temporal variability (Figures 2 and 6). In general, the enhancements of both gases are more pronounced in winter relative to spring and summer months (Figure 2). Prior studies have shown that anthropogenic (fossil) $CO_2$ sources dominate in winter months due to increased emissions from the residential and electric production sectors. On average, more urbanized areas such as the USC site near Downtown LA exhibit larger median $\Delta CO_2$ and $\Delta CH_4$ values during 2015 (Figure 6 and Tables 4

and 5). $CH_4$ shows a slightly different spatial distribution in the median enhancement relative to $CO_2$, with the second largest $CH_4$ enhancements observed at the GRA site, which is a suburban site located in the San Fernando Valley.

The $CO_2$ and $CH_4$ enhancements also exhibit long-tail distributions, a reason we report the median and interquartile range in Tables 4 and 5 in addition to the other statistics. As mentioned earlier, relatively large $CH_4$ excursions, on the order of 4 ppm above background or more, are observed throughout the GRA time series (Figure 2). The GRA site also exhibits a

long-tail distribution with respect to the $CH_4$ enhancements, which is more pronounced compared to the other sites, even during midday hours (see Supplemental materials Figure S6, which shows the outliers in addition to the median and interquartile range). Many of the larger enhancements occur during night-time/early morning hours. The smaller enhancements during midday hours relative to night suggest that GRA may be sensitive to a local $CH_4$ source at night, when the PBL becomes shallower and could be more stratified (Figures 2 and 6 and Tables 2–5). The long-tail distribution for

$CH_4$ in Los Angeles and the prevalence of fugitive $CH_4$ emissions across the LA urban landscape was previously demonstrated by Hopkins et al. (2016), using extensive mobile surveys. Hopkins et al. (2016) identified 75% of methane hotspots to be of fossil origin, while 20% were biogenic, and of 5% of indeterminate source using the ratio of ethane to methane ($C_2H_6/CH_4$). They also found that fossil fuel sources accounted for 58-65% of methane emissions and suggested that there are widely distributed methane sources, primarily of fossil origin, that are not included in bottom-up inventories.

In future work, detailed analysis of winds, measurement footprints, and tracer/tracer analyses will be used to evaluate the origin of the anomalous $CH_4$ enhancements.

**6 Uncertainty in the $CO_2$ and $CH_4$ enhancements ($U_{Enhancement}$)**

Both analytical uncertainty and imperfect knowledge of the composition of background air limit the precision of observation-based estimates of local- or regional-scale greenhouse gas enhancements (e.g. Graven et al., 2009; 2012a;

2012b; Turnbull et al., 2009; 2015). We estimate the uncertainty in the enhancement as follows:

$$(U_{Enhancement})^2 = (U_{air})^2 + (U_{BG})^2 \qquad (4)$$

where $U_{Enhancement}$ is the total uncertainty in the enhancement of $CO_2$ or $CH_4$ and is proportional to the quadrature sum of the uncertainty in the air measurement ($U_{air}$) and the uncertainty in the background mole fraction ($U_{BG}$). We note that $U_{BG}$ is not statistically independent of $U_{air}$ because $U_{BG}$ is derived from measured values. In the remainder of this study, we explore the

analytical uncertainty in our measurement approach and calibration strategy ($U_{air}$) using data from the LJO site (Section 6.1) and the uncertainty in the background mole fraction using the marine reference background from SCI (Section 6.2).





### 6.1 Measurement uncertainty analysis ($U_{air}$)

We model the analytical uncertainty in the air measurements following the general methods outlined in Andrews et al. (2014), using the quadrature sum of multiple uncertainty components:

$$(U_{air})^2 = (u_{extrap})^2 + (u_{h2o})^2 + (u_M)^2 \qquad (5)$$

5 *where*

$$u_M = u_{TGT} \qquad (6)$$

*or*

$$(u_M)^2 = (u_p)^2 + (u_b)^2 + (u_{scale})^2 \qquad (7)$$

(whichever is greater).

10 Equation 5 describes $U_{air}$, the total uncertainty in the reported air mole fractions, and its individual components, which have units in mole fraction $CO_2$ or $CH_4$ (ppm or ppb). In Eq. 5, $u_{extrap}$ is the extrapolation uncertainty, or the uncertainty introduced because the measured mole fraction of the air sample differs from the value of the calibration standard (Section 6.1.1), and $u_{h2o}$ is uncertainty from the treatment of water vapor (Section 6.1.2). In Eqs. 6-7, $u_M$ is the greater of two terms, defined by either $u_{TGT}$ the uncertainty determined by the target tank measurements or the quadrature 15 sum of several terms: $u_p$, the analyzer precision (Section 6.1.4), $u_b$, the analyzer calibration baseline uncertainty (6.1.5), and $u_{scale}$ the scale reproducibility (Section 6.1.6). In Eq. 6, $u_{TGT}$ is equivalent to a Root Mean Square Error (RMSE), and is estimated using the corrected target tank residual over 10 days, similar to Andrews et al. (Section 6.1.3).

Overall, Eqs. 5 to 7 describe a generic algorithm that can be applied to other analyzers, as well as for CO measurements. Time-dependent monitoring of $u_b$, $u_p$ and $u_{TGT}$ is useful when tracking analyzer performance. Although the 20 overall measurement uncertainty is typically small, an increase in any of these values ($u_b$, $u_p$ and $u_{TGT}$) may indicate problems with a specific analyzer. Thus, this system could be used to generate alerts for the data user to identify periods when an analyzer is performing poorly or to indicate periods when the measurements may not be useful for atmospheric inverse modelling studies.

### 6.1.1 Extrapolation uncertainty ($u_{extrap}$)

25 We corrected the air measurements in Figure 2 using a one-point calibration method. As a result, any air measurement that is different from the value of the calibration standard is subject to an extrapolation uncertainty, $u_{extrap}$, which is the uncertainty introduced because the measured mole fraction of the air sample differs from (and in many cases is





larger than) the value of the calibration standard (around 400 ppm $CO_2$ and 1850 ppb $CH_4$). We estimate $u_{extrap}$ as follows:

$$u_{extrap} = |\varepsilon| \times |Xcorr - Xassign_{cal}| \tag{8}$$

where $\varepsilon$ (described below) has units of ppm/ppm or ppb/ppb and is multiplied by the absolute value of the difference between the sampled air concentration and the and assigned calibration tank value ($|Xcorr - Xassign_{cal}|$).

Our approach relies on independent estimates of $\varepsilon$, the error due to the single-point calibration method, to determine the magnitude of the systematic and random components of the error in our calibration method. Ideally, initial estimates of $\varepsilon$ would be determined empirically via testing each analyzer in a laboratory prior to deployment in the field to provide estimates of the magnitude of the extrapolation uncertainty (e.g. Andrews et al., 2014; Richardson et al., 2012). At the time of this study, it was not possible to test many of the CRDS analyzers in a laboratory prior to deployment in the field because
high mole fraction standards spanning the range of $CO_2$ and $CH_4$ measurements expected in LA were not available.

Since a suite of calibration standards was not available at the time of this study, we determined $\varepsilon$ using the average "correction" slope determined from analysis of a series of standard tanks at different mole fraction tanks on a suite of CRDS analyzers. Within the LA network, only the LJO and VIC analyzers had field calibration data from high mole fraction tanks available at the time of this study. We used the limited measurements of these high mole fraction tanks (approximately 500
ppm $CO_2$ and 2600 ppb $CH_4$) to compute an average $\varepsilon$ over the period when a high mole fraction tank was available. We also investigated laboratory calibration data from the seven additional Picarro CRDS model G2401 and G2401-m analyzers, as described below. These analyzers are not part of the network, but are similar to the CRDS analyzers used in the field in the LA network.

Calibration analyses for the seven independent analyzers were performed at NOAA/ESRL during 2014 to 2015 with
between 3 and 7 reference tanks calibrated on the WMO scales for each gas (up to approximately 470 ppm $CO_2$ and 3060 ppb $CH_4$). A single standard tank (the tank with a $CO_2$ value closest to 400 ppm) was set as the calibration standard ($Xassign_{cal}$) and was used to correct the CRDS reading for the other standard gases using Eq. 2. Next, we plotted the residual of the corrected mole fraction for each tank measurement and its assigned value ($Xcorr - Xassign$) as a function of the difference in the assigned mole fraction between a given tank and the calibration tank ($Xassign_{span} - Xassign_{cal}$). The slope
of this relationship is equivalent to $\varepsilon$ for a given analyzer. Estimates of the correction factor, $\varepsilon$, and regression statistics for these seven analyzers are summarized in Tables 6 and 7 (the data are shown in the Supplemental materials, Figures S1 and S2).

The values of the slope correction ($\varepsilon$) are 0.0027 and 0.0018 ppm/ppm for $CO_2$ and 0.0012 and 0.0060 ppb/ppb for $CH_4$, for the LJO and VIC analyzers respectively. These results are compared with the other analyzers in Table 6. For $CH_4$,
all analyzers show a clear linear relationship between the error and the mole fraction of the tank, and there is very little





difference in the slope between different analyzer units (see Supplementary materials, Figure S2). Interestingly, for $CO_2$, we find that the two older analyzers (CFKBDS-2007 and -2008) have larger slopes, while the majority of the analyzers have very little dependence on the mole fraction and have errors close to zero (see Figure S1). The results in Table 6 are used to estimate the magnitude of the error in the corrected air sample mole fractions caused by assuming a constant analyzer

sensitivity, or slope correction. The average value of $\varepsilon$ from all 9 analyzers was used to estimate an extrapolation to our single-point calibration and the uncertainty in this correction ($u_{extrap}$). The slope from these calibration experiments ($\varepsilon$) gives an estimate of the error in the single point calibration and how it increases when the measurement is farther from the value of the single calibration point. Overall, $u_{extrap}$ is proportional to the fractional difference between the mole fraction of the air sample and that of the ambient-level calibration tank. The average and standard deviation of $\varepsilon$ also provide estimates of the

systematic and random components of the error in the single-point calibration method (Table 7).

We also estimated the error associated with the single-point calibration strategy using Eq. 8 and various estimates of $\varepsilon$ for 3 cases: (1) the average and standard deviation of $\varepsilon$ from all 9 analyzers, (2) the average $\varepsilon$ from 7 analyzers (excluding LJO and VIC), and (3) an instrument specific estimate of $\varepsilon$ from the LJO site (Table 5). Next, we estimated the error assuming an hourly average air measurement of 500 ppm $CO_2$ and 6000 ppb $CH_4$ (i.e. roughly 100 ppm $CO_2$ and 4000 ppb

$CH_4$ enhancement above the "near ambient" calibration standard). Finally, we corrected air data from the LJO and VIC sites using an "Alternate Calibration Method," during times when a limited number of measurements of a high mole-fraction $CO_2$ and/or $CH_4$ standard were available for analysis (see Appendix, Figures A2 and A3). Overall, the difference between the single-point (default) calibration method and the "Alternate Calibration Method" are <0.2 ppm $CO_2$ and <5 ppb $CH_4$ for the majority of air measurements. We also estimated the maximum correction using both approaches (the "Alternate Calibration

Method" and a correction and error based on $u_{extrap}$), and the results are summarized in Table 6.

While the initial results are very promising, and the corrections tend to be small, there is a large degree of variability in the estimates of $\varepsilon$ for individual analyzers. The value of $\varepsilon$ can be different for different analyzers and can also change over time for a single analyzer (Tables 6 and 7 and Figures S1 and S2). Based on the experiments discussed here, our current calibration strategy could be modified to correct the concentration data using the mean value of $\varepsilon$ found from all the

analyzers and estimating an uncertainty in that correction. However, our approach for estimating $\varepsilon$ is based on relatively small statistical sample of analyzers. Furthermore, the two estimates we do have from the LJO and VIC field sites only rely on one additional calibration point other than the calibration tank, making it difficult to estimate a robust fit for these analyzers. An estimate of $\varepsilon$ for each analyzer in the field (or from a larger statistical sample of analyzers) is needed to provide a robust estimate of the mean $\varepsilon$ to correct the air sample data. Values of $\varepsilon$ could also be estimated for the analyzers

deployed in the field, for example, by deploying a suite of calibration standards with varying concentrations of $CO_2$ and $CH_4$ (e.g. a round-robin). We have chosen not to correct the data and keep it tied to the single-point calibration until more experimental evidence can be obtained. In the future, the surface network will move to a 2-point calibration strategy. This





will rely on the availability of high-mole fraction tanks for deployment in the field, and a calibration uncertainty that is lower uncertainty than our current estimates for $u_{extrap}$.

**6.1.2 Uncertainty associated with water vapor ($u_{h2o}$)**

The presence of water vapor in the sample air contributes to the uncertainty in the CRDS measurements. Below we describe three potential sources of uncertainty in the measurements due to water vapor: 1) the coefficients used to determine the water vapor correction, which can vary from instrument to instrument, 2) bias due to imperfect drying, and 3) random noise in the $H_2O$ measurement reported by the CRDS analyzer, which ultimately gets incorporated in the water vapor correction (Rella et al., 2013).

      The Picarro CRDS analyzers use a factory default water vapor correction model that relies on the parameters
derived by Chen et al. (2010):

$$\frac{CO_{2wet}}{CO_{2dry}} = 1 + aH_{rep} + bH_{rep}^2 \qquad (9)$$

$$\frac{CH_{4wet}}{CH_{4dry}} = 1 + cH_{rep} + dH_{rep}^2 \qquad (10)$$

where $H_{rep}$ is the water vapor mixing ratio reported by the analyzer, $(CO_2)_{wet}$ and $(CH_4)_{wet}$ are the uncorrected CRDS, wet-gas mole fractions reported by the analyzer, $(CO_2)_{dry}$ and $(CH_4)_{dry}$ are the dry-gas mole fractions, while a, b, c, and d are
experimentally determined parameters (where a = -0.012000, b = -0.0002674, c = -0.00982, and d = -0.000239). This correction is currently being applied to the analyzers in the LA network. Users are free to design and perform their own experiments and derive parameters specific to each instrument (Nara et al., 2012; Rella et al., 2013; Welp et al., 2013). However, while an instrument specific correction of water vapor could potentially lead to reduced uncertainty, prior laboratory studies have also found that the benefits of an instrument specific correction are small at low water vapor levels
(Nara et al., 2012; Rella et al., 2013).

      The Nafion drying system described in Section 2.3 and by Welp et al. (2013) allows us to stabilize the water vapor concentrations in the sample gas stream ($H_{rep}$ in Eqs. 9-10) to 0.1±0.01%. With this drying system, the uncertainty in the water vapor correction drops to 0.015 ppm for $CO_2$ and 0.21 ppb for $CH_4$ when using the factory parameters described above (Rella et al., 2013; Welp et al., 2013).

The use of a Nafion dryer could also potentially introduce a bias due to imperfect drying. A slight permeation of $CO_2$ and $CH_4$ can occur across the membrane, especially when the Nafion membrane is wet (e.g. Ma and Skou, 2007; Welp et al., 2013). In our measurement setup, running the dry standard gases through the Nafion dryer significantly reduces this




bias effect. The water vapor concentration from the dry standard gas runs is similar to that of the preceding air measurements. We find that the water vapor mole fraction in the air measurements after a standard is run drops by 0.01% (from 0.10% to 0.09%). A similar effect has been described by Rella et al. (2013). We estimate the Nafion bias in our system based on this 0.01% variability in water vapor to be -0.011 ppm for $CO_2$ and 0.00028 ppb for $CH_4$ based on

laboratory experiments performed at the SIO laboratories with the same Nafion drying system used in the field. Details about the laboratory experiments are available in the Supplementary materials (see Figure S3).

A final source of uncertainty regarding water vapor correction comes from the variability of the water vapor measurement on the CRDS analyzers. We estimate this to be 0.014 ppm for $CO_2$ and 0.069 ppb for $CH_4$ at the water vapor concentrations of our measurements (Rella et al., 2013; Welp et al., 2013).

The total uncertainty due to water vapor ($u_{h2o}$) is the quadrature sum of the water vapor correction uncertainty, the Nafion-induced bias due to changes in water vapor, and the variability (noise) of the water vapor measurements. Therefore, we estimate $u_{h2o}$ is 0.0233 ppm for $CO_2$ and 0.221 ppb for $CH_4$ across the network and it is assumed to be constant at all times.

### 6.1.3 Uncertainty derived from target tank measurements ($u_{TGT}$)

We define $u_{TGT}$ in Eq. 6, where the target tank is treated as an unknown and the measured value is compared to the tank assignment to calculate the root mean square error (RMSE):

$$u_{TGT} = \sqrt{(Xcorr_{TGT} - Xassign_{TGT})^2} \qquad (11)$$

where, $Xcorr_{TGT}$ is the corrected target tank measurements and $Xassign_{TGT}$ is the assigned value of the target tank by the calibration laboratory (NOAA/ESRL or SIO). The assigned values are constant over the lifetime of the cylinder and are

determined based on laboratory measurements traceable to the WMO scales. Errors in the tank assignments are typically small and would result in a bias in the measurement, rather than a random error (see Section 6.1.6). To calculate $Xcorr_{TGT}$, the uncorrected CRDS target tank concentration readings are treated as an unknown sample and are corrected using Eq. 2. For each target tank measurement, $u_{TGT}$ is calculated as the RMSE (Eq. 11) over 11 target measurements centered on the measurement time (this is usually a 10-day period). Then, this time-dependent $u_{TGT}$ is interpolated in time onto all the air

measurements. Overall, $u_{TGT}$ is equivalent to a RMSE and includes errors in the assigned value of the calibration tank and the target tank, and also encompasses other errors (e.g. the instrument precision and the calibration standard baseline uncertainty), as well as additional and possibly unknown errors due to delivery of air to the analyzer downstream of the Valco valve. Drift in either the calibration or target cylinders will also manifest as an increasing $u_{TGT}$. In this way, $u_{TGT}$ is useful as a diagnostic of instrument performance.





### 6.1.4 Analyzer precision ($u_p$)

The analyzer precision ($u_p$) is defined as the standard deviation of the 10-minute daily calibration standard tank measurement:

$$u_p = \sigma_{cal} \tag{12}$$

where $\sigma_{cal}$ is the standard deviation of the uncorrected CRDS, dry mole fraction measurements for the calibration tank at roughly 2.5-second resolution. Our definition of $u_p$ is different from that described by Andrews et al., (2014), where the analyzer precision was defined as the standard error of the calibration measurements. To use the standard error, we must assume statistical independence of the measurements and estimate a maximum value for N, the number of samples in the average that reduce the uncertainty.

We performed an Allan deviation analysis to estimate stability of the Picarro CRDS analyzer due to noise processes. The Allan deviation is the square root of the Allan variance (Allan, 1966, 1987) and was plotted as a function of averaging time for calibration runs at the LJO site during January 2016 (Figure S4). During this month, the calibration tank was run 28 times through the CRDS analyzer for 30 minutes at each time (10 minutes longer than the normal calibration run period, for quality check purposes). We omitted the first 10 minutes of data and performed the Allan deviation analysis on

the next 20 minutes of data for each of the 28 calibration runs. We found that the instrument variability does not average with a slope of -1/2 as would be expected for a white noise profile, indicating correlation in the noise at various longer time scales. The deviation (noise) therefore does not decrease as the inverse square root of the averaging time ($\sqrt{N}$), as it would for white noise. Filges et al. (2015) found a comparable result using similar CRDS units. Figure S4 shows the Allan deviation analysis for a subset of six (for figure clarity) of these calibration runs over the course of the month, also indicating

that the characteristics of the noise in the analyzer varies. The deviation does decrease with averaging time, but not in a consistent manner. Therefore, we have chosen not to compute the standard error in the mean by dividing the standard deviation by the square root of the number of measurements, because the characteristics of the noise in the analyzer vary with time and the data does not fit the criterion of the measurements being truly independent. We therefore quantify the precision of the analyzers as the 2.5-second standard deviation independent of averaging time, recognizing that it is likely an

overestimate of the analyzer precision. This uncertainty for $CO_2$ and $CH_4$ is small compared to other sources, so we chose to retain it, considering that in the future (or for other species, such as CO), we will model the precision in a more robust manner.

### 6.1.5 Calibration baseline uncertainty ($u_b$)

To estimate the calibration baseline uncertainty ($u_b$) we follow a process similar to that described by Andrews et al.

(2014). First, we calculate three different possible time series of the calibration tank measurement ($X'$) to estimate $S_{cal}$ (the instrument sensitivity measured for the calibration tank). The first is an interpolation onto air data using every calibration run





($S_{cal}$). The second and third time series use alternate sampling of the calibration tank time series (i.e. by either odd or even sampling of every other daily calibration run) to interpolate $X'$ onto the time series of the air sample data (see Figure S5). Next, we calculate the dry air mole fraction corrected at each point using each of these three different time series. The maximum uncertainty, $u_{bmax}$, is estimated as the standard deviation of the three corrected mole fractions (black solid line,

Figure S5). The actual uncertainty, $u_b$, is equal to this maximum value ($u_{bmax}$) at the halfway point in time between subsequent calibration runs, and goes to zero at the time of a calibration run, since that that time the calibration value is known exactly. Thus, $u_b$ is equal to $u_{bmax}$ weighted by the time difference between an air sample measurement and the adjacent calibration run (dashed line, Figure S5).

### 6.1.6 Uncertainty in calibration tank assignments

Absolute scale accuracy includes uncertainties in the values assigned to the primary calibration standards, as well as scale propagation errors (Andrews et al., 2014). Here we report an expanded uncertainty (95% C.L., approximately 2σ): 0.20 ppm at 400 ppm $CO_2$ (WMO X2007 scale) and 3.5 ppb at 1850 ppb $CH_4$ (WMO X2004A scale), where the total uncertainty is a relatively small function of the measured mole fraction. However, in our case, all measurements are calibrated relative to the same (WMO) scale, so scale reproducibility is the relevant metric for assessing measurement compatibility over time

and between sites. Similar to Andrews et al. (2014), the reported scale reproducibility is 0.06 ppm for $CO_2$, and 1.0 ppb for $CH_4$ (2σ) (*B. Hall, personal communication*). We use the 1σ scale reproducibility ($u_{scale}$) in the calculation of $U_{air}$ (0.03 ppm $CO_2$, and 0.31 ppb $CH_4$)

Cylinder drift has not been discussed and could also impact this component of the measurement uncertainty. Andrews et al. (2014) report a mean difference between pre- and post-deployment tank calibrations of $CO_2$ and $CH_4$. $CO_2$

has rarely been observed to drift in cylinders, while $CH_4$ standards are very stable. Andrews et al. (2014) report a mean difference between pre- and post-deployment tank calibrations of 0.02±0.05 ppm $CO_2$ (post- minus pre-deployment from 177 tanks analyzed over approximately 10 years). $CH_4$ standards are generally very stable and field calibration residuals reported for $CH_4$ had not indicated any drift in the tanks (for $CH_4$ absolute stability is reported as 0±0.1 ppb yr$^{-1}$ (Dlugokencky, 2005; Dlugokencky et al., 1994). $CH_4$ standards were generally stable, (Andrews et al., 2014; Dlugokencky,

2005). At the time of this study, none of our field calibration cylinders for the LA surface network had final calibrations; however routine field measurements of standard tanks to date do not indicate significant drift in either gas.

### 6.2 Uncertainty due to background

We used the standard deviation of the fit residuals to define an uncertainty in the background mole fraction as follows:

$(U_{BG})^2 = (X_{RMSE})^2$                                                                          (13)



where $X_{RMSE}$ is the standard deviation of the fitted curve residuals. For the SCI reference curve fit, $U_{BG}$ is 1.1 ppm $CO_2$ and 11.7 ppb $CH_4$.

### 6.3 Comparison of uncertainty estimates

Figure 7 shows the time-dependent analytical uncertainty estimates for the LJO site during 2015 and Table 7 gives

the average values for each term. We assigned fixed values for $u_{scale}$ (0.03 ppm $CO_2$ and 0.31 ppb $CH_4$) and $u_{h2o}$ (0.0233 ppm $CO_2$, 0.221 ppb $CH_4$). Overall, $u_{h2o}$, and $u_{scale}$ are small components of $U_{air}$, the overall measurement uncertainty. We do not have time dependent estimates of all the uncertainty terms used in calculating $U_{air}$ for every analyzer.

Under normal operating conditions, the calibration baseline uncertainty ($u_b$) and the analyzer precision ($u_p$), are also negligible. The average $u_b$ is 0.0042 ppm and 0.054 ppb for $CO_2$ and $CH_4$ respectively, with no significant outliers (based on

the average for 11 analyzers deployed in the field). Similarly, $u_p$ is a very small component of the overall uncertainty. Similarly, the values for analyzer precision across the network are similar to those derived from the LJO analyzer under normal operating conditions (roughly 0.024 ppm $CO_2$, 0.22 ppb $CH_4$ for the 20-minute average air observations, and 0.011 ppm $CO_2$ and 0.12 ppb $CH_4$ for the 1-minute average air observations). Both $u_b$ and $u_p$ can become non-negligible components of the uncertainty if there are problems with either the CRDS analyzer, or the delivery of calibration gas to the

analyzer. For example, the standard deviation of some calibration runs may be higher than the values reported for the LJO analyzer suggest, either because of analyzer noise increasing due to hardware or software problems, analyzer drift during a calibration run, or because a limited number of calibration measurements were available to calculate an average due to analyzer problems. Therefore, the values derived from the LJO analysis represent the minimum quantities we expect for $u_p$, which is representative of the precision from a well-performing analyzer.

Overall, $u_{extrap}$, provides an estimate of the uncertainty due to the single-point calibration method, which is the largest component of uncertainty in the air measurements (Figure 7 and Table 7). We find that $u_{extrap}$ is linearly dependent on the difference between the mole fraction of the air sample and that of the ambient-level calibration tank, at least over the range of mole fractions tested (see Supplementary materials, Figures S1 and S2). As described earlier, we do not have instrument-specific estimates of $\varepsilon$ for every analyzer to use in estimating $u_{extrap}$. Therefore, we assumed constant values for $\varepsilon$

based on the average of the 9 analyzers shown in Table 6. During 2015, the average $u_{extrap}$ value estimated for the LJO analyzer is 0.047 ppm $CO_2$ and 0.46 ppb $CH_4$. The magnitude of $u_{extrap}$ is larger for air data with higher mole fractions, and scales as a percentage of the difference in the mole fraction of the air sample above the assigned value of the calibration tank. On average, the uncertainty due to $u_{extrap}$ results in an uncertainty in the enhancement on the order of 0.0025 ppm/ppm (0.25%, or 0.25 ppm for a 100 ppm enhancement) for $CO_2$ and 0.003 ppb/ppb (0.3% or 0.30 ppb for a 100 ppb enhancement)

for $CH_4$. Based on analysis of the LJO data during 2015, the average value of $U_{air}$ is 0.070 ppm $CO_2$ and 0.72 ppb $CH_4$ (Table 8). Overall, these experiments show that the single-point calibration introduces rather small errors in the final mole





fraction assignments for $CO_2$ or $CH_4$, and especially relative to the enhancement above background (Figure 7 and Tables 4–8).

We used Eq. 4 to estimate the uncertainty in the enhancement signal using the estimates of $U_{air}$ and $U_{BG}$ for the LJO analyzer and the SCI background estimate, respectively. Since $U_{air}$ is time varying, the uncertainty in the enhancement is
also time-dependent. On average, uncertainty in the enhancement is roughly 1.1 ppm and 11.7 ppb, for $CO_2$ and $CH_4$, respectively for the LJO air data. Overall, the uncertainty due to the assumptions about the background condition is the largest component of the error in the enhancement. However, on an annual average basis, the total uncertainty is generally less than roughly 5 and 15% of the enhancement in downtown LA for $CO_2$ and $CH_4$, respectively.

**7 Summary and Conclusions**

Concerns about rising greenhouse gas levels have motivated many nations to begin monitoring or mitigating emissions, motivating the need for robust, consistent, traceable greenhouse gas observation methods in complex urban domains. Observations from organized urban greenhouse gas monitoring networks such as the LA surface network are emerging elsewhere (e.g. Shusterman et al., 2016; Turnbull et al., 2015; Xueref-Remy et al., 2016). To date, most of these research efforts have been largely disconnected. More information flow between existing urban observational networks and
the science and applications communities is needed to understand greenhouse gas emissions from cities. Data and methods for greenhouse gas monitoring in urban regions should be fully disclosed and documented with a small degree of latency to make the best use of these atmospheric data for emissions verification and/or for informing policies more generally.

In this study, we describe the instrumentation and calibration methods used for the Los Angeles Megacity surface network. The measurement and sample module system described here provide robust, near-continuous and unattended
measurement of $CO_2$ and $CH_4$ at urban and suburban monitoring stations in the South Coast Air Basin. A total of eleven analyzers have been deployed thus far, and most have been operational for more than 1.5 yrs. We reported the sampling configuration, algorithms to compute calibrated $CO_2$ and $CH_4$ mole fractions, and methods for estimating the local enhancement above background and uncertainties.

We presented an observation-based method for estimating background mole fractions using measurements from
four remote, "extra-domain" sites. Our approach to background determination is useful for exploring variability in the enhancement signals. Relative to the enhancements observed at most sites, there is near equivalence of continental and marine boundary layer background conditions, except during summer months, when continental sites may not be relevant for estimating background due to prevailing on-shore flow conditions in the basin. One strength of our observation-based strategy for background determination is the relatively short latency with which background observations can be evaluated
(hours to days). This will be important as greenhouse gas research networks such as the LA network transition from research networks into monitoring networks and will allow near real-time estimation of local greenhouse gas enhancements.





The stability criteria discussed here could also be used to identify periods that are optimal for flux inversion. For example, it may not be useful to select background observations when influences from outside the domain cause large gradients or fluctuations within the domain. Similarly, periods that are impacted by recirculation effects are not ideal for identifying background and thus are also not useful for estimating fluxes, and the measurement stability criteria may also be useful for

identifying such periods.

We calculated $CO_2$ and $CH_4$ enhancements in the LA megacity from during 2015 using a marine background estimate. An urban site near Downtown LA has a median enhancement of roughly 20 ppm $CO_2$ and 150 ppb $CH_4$ during all hours, and roughly 15 ppm $CO_2$ and 80 ppb $CH_4$ during midday hours (roughly 12-16:00 LT, local time), which is the typical period of focus for flux inversions. "Suburban" sites show moderate, but slightly smaller enhancements, with median values

of roughly and roughly 5-10 ppm $CO_2$ and 30-70 ppb $CH_4$ during midday hours. Overall, the largest $CO_2$ and $CH_4$ enhancements were observed at the USC site near Downtown Los Angeles.

We also described the components of the analytical uncertainty that we believe to be most important for urban studies. The uncertainty in the enhancement was estimated using both the uncertainty in the air sample data collected form the measurement system and the uncertainty in the background mole fraction. The algorithm discussed here can also help

determine periods when uncertainties in the observation are small and are therefore most useful for atmospheric inversion studies. The acceptable threshold for the measurement uncertainty in part depends on the question of interest, and how large the signal is relative to a local background (i.e. the enhancement).

Our analysis shows that the uncertainty in the single-point calibration method ($u_{extrap}$) is the largest component of the measurement uncertainty. Overall, $u_{extrap}$, the uncertainty in the single-point calibration strategy, scales as a function of

the enhancement in the air data (roughly 0.3% of the enhancement for both $CO_2$ and $CH_4$). Based on our error analysis, $u_{extrap}$ depends on the response or sensitivity of the analyzer, which is time varying. Our assessment of $u_{extrap}$ could be further improved with more estimates of the correction factor ($\varepsilon$) from a larger statistical sample of analyzers. Currently, our ability to fully evaluate the magnitude of the correction to the air data is limited by the availability of high concentration standards in the field. In the near future, the LA measurement network will begin using analyzer specific estimates of the correction

factor based on periodic measurements with high mole fraction tanks, which will allow correction of the random and systematic components of the uncertainty associated with the single-point calibration strategy.

While measurement uncertainty is important for estimating gradients between sites, accurate background determination and uncertainties in atmospheric transport will likely be more important for estimating urban enhancements and using observations in flux inversions. Overall, the uncertainty associated with background is larger than the analytical

uncertainty; however both the analytical and background uncertainty are likely to be smaller than the uncertainty due to atmospheric transport, a topic that we have only discussed briefly to provide context for the observations presented in this study. Our results suggest that reducing the uncertainty to less than 5% of the enhancement will require detailed assessment



of the impact of meteorology on background conditions over a range of conditions (e.g. following Feng et al., 2016). Future modelling efforts for the LA Megacity Carbon Project may require equivalent attention to meteorological validation, as has been demonstrated here for the greenhouse gas observations, due to uncertainties in atmospheric transport.

5 "Top down" flux inversions relying on in situ greenhouse gas observations require accurate determination of urban enhancements relative to a local background. We calculated an expected atmospheric signal of Los Angeles carbon emissions assuming emissions are distributed evenly over the roughly 17,100 $km^2$ are of the South Coast Air Basin (SCB), an average wind speed of 2 m/s (based on annual average wind speed observed at the USC observation site) equivalent to a transit time of ~18 h), an average mixed layer depth of 1 km (Rahn and Mitchell, 2016; Ware et al., 2016) and estimated emissions, (Pacala et al., 2011). Estimated annual emissions of 144 Tg $CO_2$ $y^{-1}$ would raise $CO_2$ mole fractions by roughly 10 10 ppm (based on Hestia-LA 2012, see e.g. Gurney et al., 2012, 2015). Assuming an annual emissions estimate of 0.4 Tg $CH_4$ $y^{-1}$ in the SCB based on a top-down study, $CH_4$ mole fractions would be enhanced by roughly 75 ppb (Wong et al., 2015). These estimates are consistent with the midday enhancements observed over downtown LA during 2015 (Figure 6 and Tables 4–5), and those reported previously (Newman et al., 2013, 2016; Wong et al., 2015).

In the future, urban greenhouse gas monitoring networks such as the LA surface network could also be used to 15 understand episodic sources or disturbance events such as fires, gas leaks, etc., which are difficult to capture with bottom-up approaches. This will also require background estimation in near real-time. Co-monitoring of tracers (e.g. $CO_2$ and CO enhancements, calibrated with [14]C measurements) is also planned as part of future work and will allow continuous or near-continuous estimation of fossil carbon signals in Los Angeles (Miller et al., 2015). Establishing greenhouse gas enhancements and emissions trends over a period of several years could help assist in determining the effectiveness of local 20 control measures and mitigation strategies. As part of future work, we plan to use forward and inverse modelling studies and tracer-tracer analyses in conjunction with the calibrated $CO_2$ and $CH_4$ observations from the LA surface network presented here to estimate greenhouse gas emissions fluxes, determine spatial and temporal emissions trends, and to attribute those fluxes to specific sectors and/or sources.



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

comments that significantly improved the manuscript. We also thank E. Dlugokencky and A. Andrews for providing the Pacific Marine Boundary Layer Reference, which is constructed using measurements from the NOAA Global Greenhouse Gas Reference Network.  We thank C. Sweeney and T. Newberger at NOAA/ESRL for calibration data on the series of Picarro G2401 analyzers presented in this study and B. Hall for providing calibration gases and for advice regarding uncertainty in the WMO/NOAA scales. We also thank A. Cox, W. Paplawsky, and T. Lueker at the SIO calibration

laboratories for support regarding site operations and calibration tanks. Earth Networks provided invaluable support for installation of sample modules and calibration gases at many of the sites. We would like to thank several Earth Networks staff, including B. Angel and C. Fain for keeping sites maintained and online and for regular status updates throughout the course of this study, D. Bixler and B. Biggs for network support, and J. Aman for regular quality control checks. A portion of this research was carried out at the Jet Propulsion Laboratory, California Institute of Technology, under contract with the

National Aeronautics and Space Administration. Additional support was provided by the NIST Greenhouse Gas and Climate Science Measurements Program and the NOAA Atmospheric Chemistry, Carbon Cycle, and Climate Program. FH's research was supported by an appointment to the NASA Postdoctoral Program at JPL, California Institute of Technology, administered by Universities Space Research Association under contract with NASA. Certain commercial equipment, instruments, or materials are identified in this paper in order to specify the experimental procedure adequately. Such

identification is not intended to imply recommendation or endorsement by the National Institute of Standards and Technology, nor is it intended to imply that the materials or equipment identified are necessarily the best available for the purpose. © 2016 All Rights Reserved.





**Figure captions**

**Figure 1.** Map of the Los Angeles Megacity and locations of the greenhouse gas monitoring network sites. Site locations are shown by the black squares (see Table 1 for details). The South Coast Air Basin (perimeter of the black line) is a geopolitical boundary including non-desert portions of the Los Angeles, Riverside, San Bernardino Counties and all of Orange County (defined by the interior back lines). Background image shows surrounding topography plotted with the average monthly nightlight radiance data from VIIRS during March 2016 (units $nW/cm^2/sr$) as a proxy for population density (http://ngdc.noaa.gov/eog/viirs/download_monthly.html). Continuous measurements from the MWO, SBC, PVP, and CIT sites are not included as part of this study, however, MWO flask data are included as part of the background analysis.

**Figure 2.** Time series plots showing the calibrated one-hour average dry air mole fractions for $CO_2$ (left panels) and $CH_4$ (right panels) in units parts per million (ppm) from nine CRDS analyzers in the LA Megacity Network. Data were collected between Jan 1, 2013 and Dec 31, 2015. Atmospheric $CO_2$ and $CH_4$ observations were corrected using the single-point calibration method. Site codes, from top:. University of Southern California/Downtown LA (USC), Compton (COM), California State University, Fullerton (FUL), Ontario (ONT), Granada Hills (GRA), University of California, Irvine (IRV), La Jolla (LJO), Victorville (VIC), and San Clemente Island (SCI). The length of each record reflects the commissioning date of each site. The gaps in the records indicate periods where the instruments were not operational or data quality was determined to be poor. Note that the y-axis scale is different for VIC and SCI ($CO_2$ and $CH_4$) and GRA ($CH_4$ only).

**Figure 3.** Time series of 1-hour average observations from the San Clemente Island (SCI, top), Victorville (VIC, middle), and La Jolla (LJO, bottom) sites between 2014 and 2015. Hourly average $CO_2$ (left panels) and $CH_4$ (right panels) measurements were filtered using stability criteria described in the test, roughly following the preliminary selection criteria described by Thoning et al., (1989). The CCGCRV curve fitting algorithm was then used to fit the selected data iteratively, removing $CO_2$ and $CH_4$ outliers $>2\sigma$ (see Appendix). The final filtered dataset (red points) and smooth curve fits (lines) are also shown.

**Figure 4.** Comparison of background estimates for Los Angeles for $CO_2$ (left panels) and $CH_4$ (right panels) at various sites during 2014 to 2015. Upper panels: Smooth curve results for Victorville (VIC, cyan); San Clemente Island (SCI, blue); La Jolla (LJO, magenta); Mt Wilson (MWO, black); and a 2-D Pacific marine boundary layer curtain estimate (Pac. MBL, yellow, red, and light blue dashed lines shown results for at 33.4°, 36.9°, and 40.5° N, respectively). The SCI, VIC, and LJO curves were generated using data selected based on stability criteria. The MWO curve was generated using night-time flask data collected every 3-4 days. Lower panels: Background estimates plotted as a difference from the MBL curtain at 40.5° N.

**Figure 5.** Back trajectories estimated for the previous 24-hours, ending in Pasadena, CA (red circle) at 14:00 PST. Results are shown for January, March, May, July, September, and November 2015 (from top left to bottom right).

**Figure 6.** Boxplot of enhancements ($\Delta CO_2$ and $\Delta CH_4$) in the LA megacity during 2015. Results are shown for $\Delta CO_2$ (upper panels) and $\Delta CH_4$ (lower panels) and for all hours (left panels) and midday hours (12-16:00 LT, right panels). Boxes outline the 25[th] and 75[th] percentiles of the sample data, respectively and red horizontal lines show the median values at each site. The sites are arranged in order by latitude going from north to south (plotted top to bottom): Victorville (VIC), Granada Hills (GRA), Ontario (ONT), Downtown LA (USC), California State University Fullerton (FUL), Compton COM), University of California, Irvine (IRV), San Clemente Island (SCI) and La Jolla (LJO). Note: Outliers are included in the statistics, but are not shown here. Figure S6 shows the same results, with outliers plotted. Observations for the ONT site began in Sept 2015, while all other results are annual averages.

**Figure 7.** Time series of uncertainties in the La Jolla (LJO) air observations. $u_p$ is the analyzer precision, $u_{TGT}$ is the uncertainty derived from the target tank measurements, $u_b$ is the calibration baseline uncertainty, and $u_{extrap}$ is the extrapolation uncertainty, or the uncertainty due to the single-point calibration strategy. $u_{extrap}$ was estimated using a mean $\varepsilon$ for 9 analyzers see text and Supplemental materials). The total analytical uncertainty in the air measurements ($U_{air}$) is calculated as described by Eqs. 5-7.





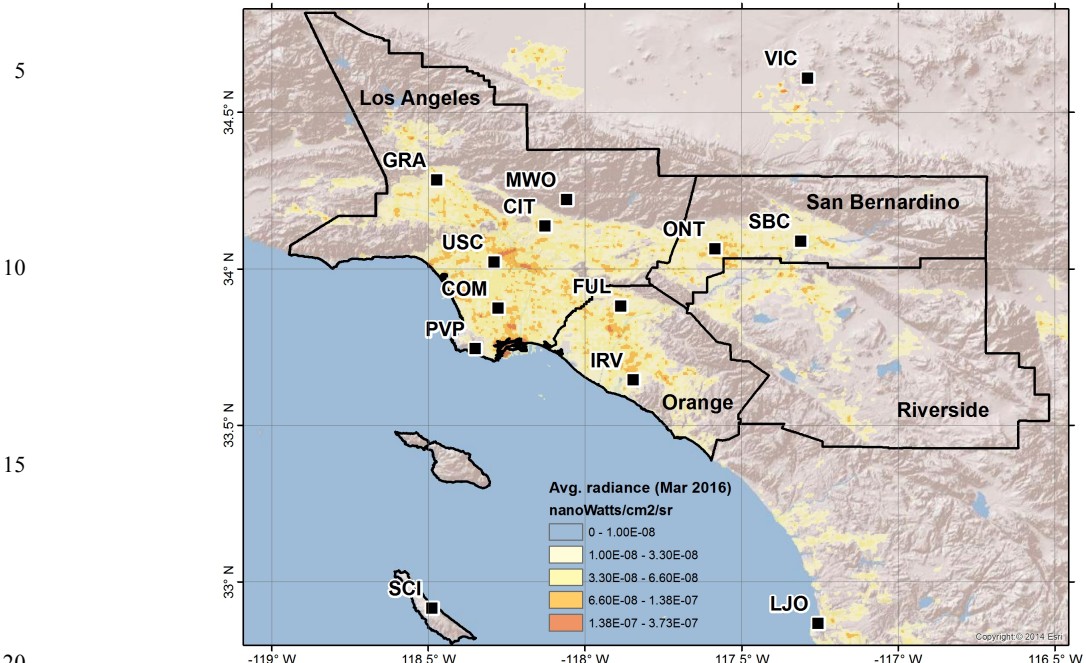

**Figure 1.** Map of the Los Angeles Megacity and locations of the greenhouse gas monitoring network sites. Site locations are shown by the black squares (see Table 1 for details). The South Coast Air Basin (perimeter of the black line) is a geopolitical boundary including non-desert portions of the Los Angeles, Riverside, San Bernardino Counties and all of Orange County (defined by the interior back lines). Background image shows surrounding topography plotted with the average monthly nightlight radiance data from VIIRS during March 2016 (units nW/cm$^2$/sr) as a proxy for population density (http://ngdc.noaa.gov/eog/viirs/download_monthly.html). Continuous measurements from the MWO, SBC, PVP, and CIT sites are not included as part of this study, however, MWO flask data are included as part of the background analysis.





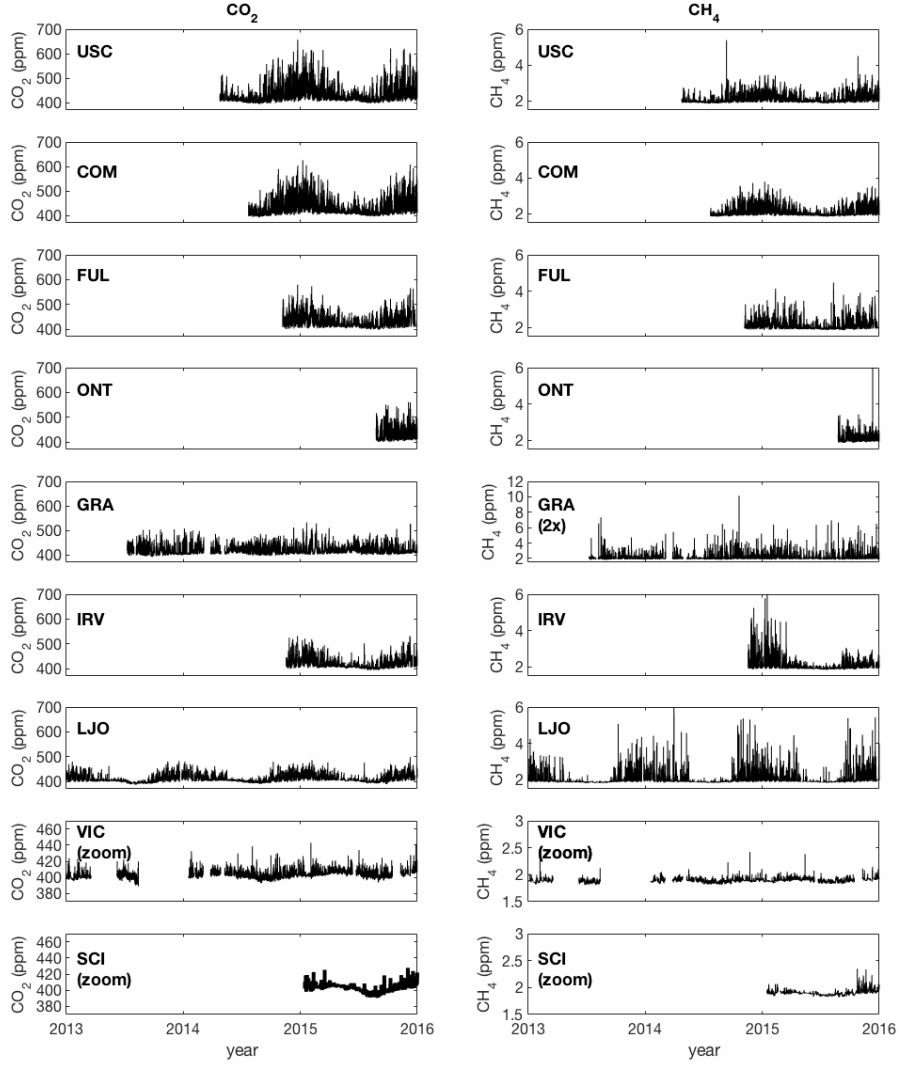

**Figure 2.** Time series plots showing the calibrated one-hour average dry air mole fractions for $CO_2$ (left panels) and $CH_4$ (right panels) in units parts per million (ppm) from nine CRDS analyzers in the LA Megacity Network. Data were collected between Jan 1, 2013 and Dec 31, 2015. Atmospheric $CO_2$ and $CH_4$ observations were corrected using the single-point calibration method. Site codes, from top:. University of Southern California/Downtown LA (USC), Compton (COM), California State University, Fullerton (FUL), Ontario (ONT), Granada Hills (GRA), University of California, Irvine (IRV), La Jolla (LJO), Victorville (VIC), and San Clemente Island (SCI). The length of each record reflects the commissioning date of each site. The gaps in the records indicate periods where the instruments were not operational or data quality was determined to be poor. Note that the y-axis scale is different for VIC and SCI ($CO_2$ and $CH_4$) and GRA ($CH_4$ only).





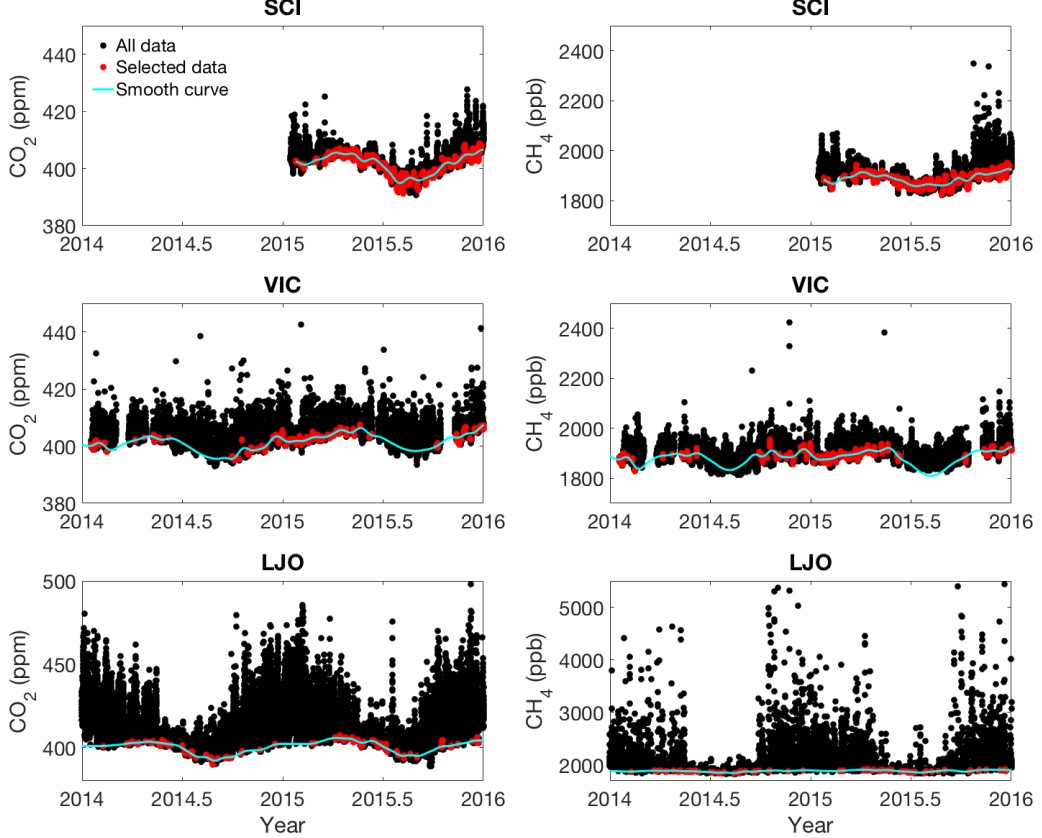

**Figure 3.** Time series of 1-hour average observations from the San Clemente Island (SCI, top), Victorville (VIC, middle), and La Jolla (LJO, bottom) sites between 2014 and 2015. Hourly average $CO_2$ (left panels) and $CH_4$ (right panels) measurements were filtered using stability criteria described in the test, roughly following the preliminary selection criteria described by Thoning et al., (1989). The CCGCRV curve fitting algorithm was then used to fit the selected data iteratively, removing $CO_2$ and $CH_4$ outliers $>2\sigma$ (see Appendix).
5   The final filtered dataset (red points) and smooth curve fits (lines) are also shown.





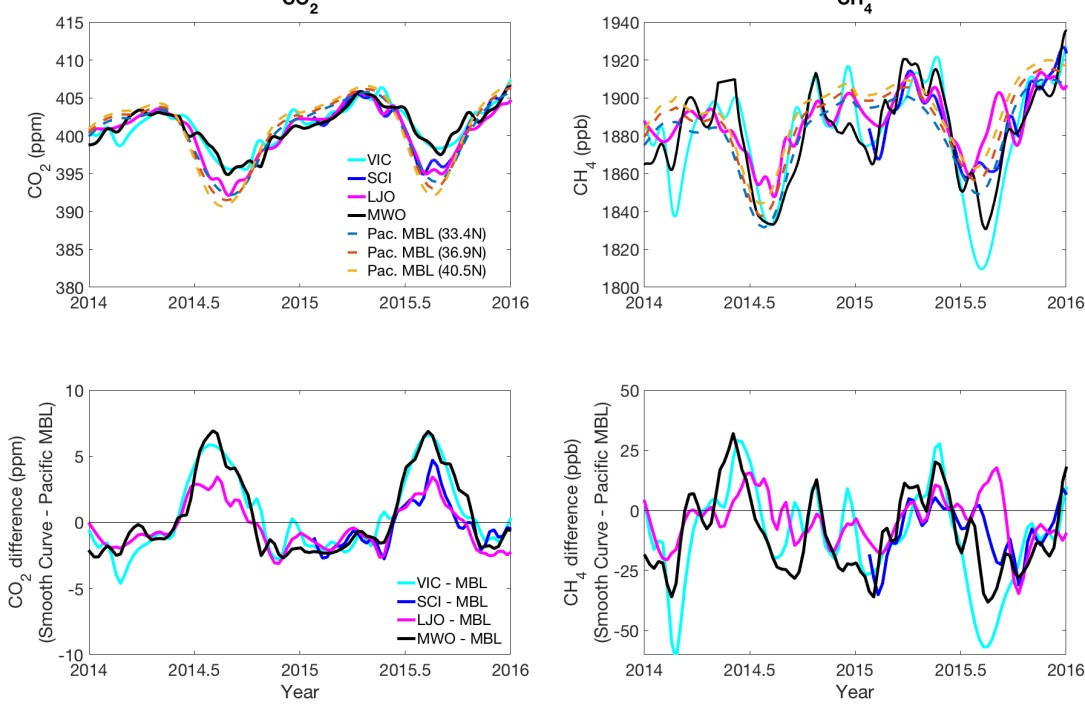

**Figure 4.** Comparison of background estimates for Los Angeles for $CO_2$ (left panels) and $CH_4$ (right panels) at various sites during 2014 to 2015. Upper panels: Smooth curve results for Victorville (VIC, cyan); San Clemente Island (SCI, blue); La Jolla (LJO, magenta); Mt Wilson (MWO, black); and a 2-D Pacific marine boundary layer curtain estimate (Pac. MBL, yellow, red, and light blue dashed lines shown results for at 33.4°, 36.9°, and 40.5° N, respectively). The SCI, VIC, and LJO curves were generated using data selected based on stability criteria. The MWO curve was generated using night-time flask data collected every 3-4 days. Lower panels: Background estimates plotted as a difference from the MBL curtain at 40.5° N.





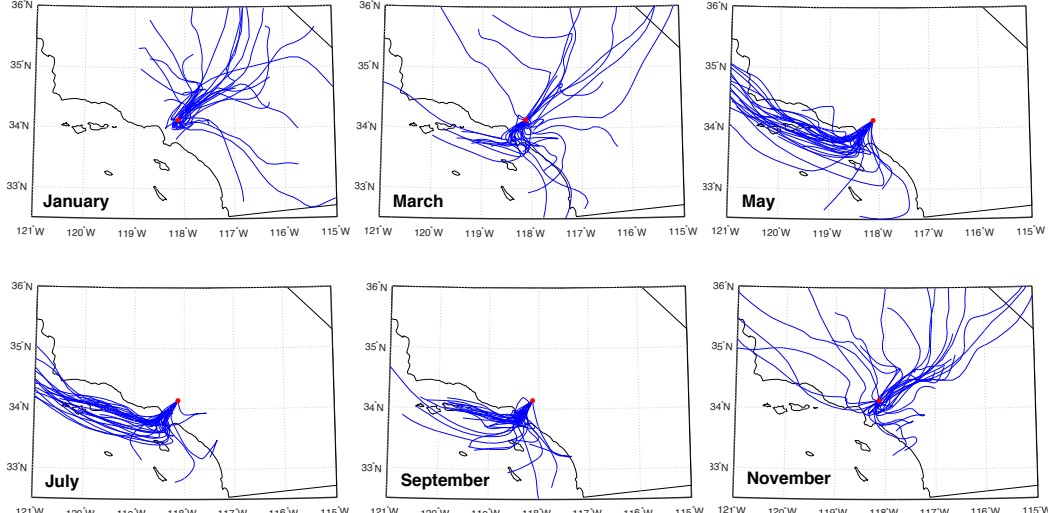

**Figure 5.** Back trajectories estimated for the previous 24-hours, ending in Pasadena, CA (red circle) at 14:00 PST. Results are shown for January, March, May, July, September, and November 2015 (from top left to bottom right).



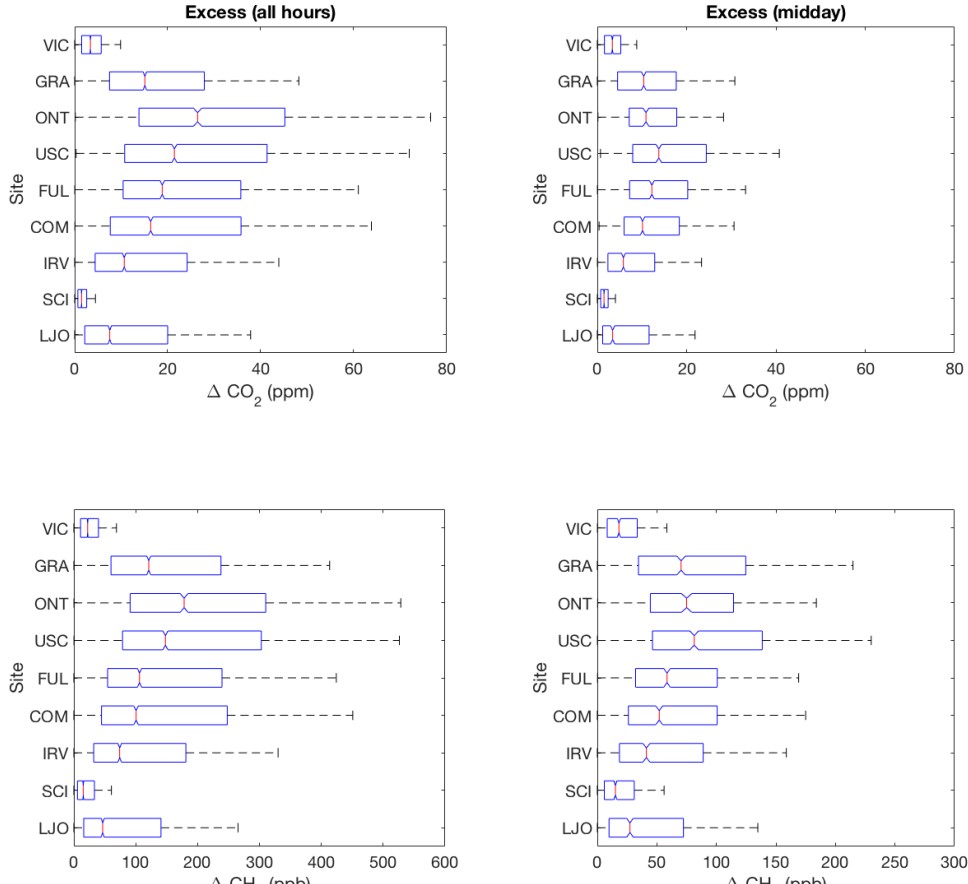

**Figure 6.** Boxplot of enhancements ($\Delta CO_2$ and $\Delta CH_4$) in the LA megacity during 2015. Results are shown for $\Delta CO_2$ (upper panels) and $\Delta CH_4$ (lower panels) and for all hours (left panels) and midday hours (12-16:00 LT, right panels). Boxes outline the 25th and 75th percentiles of the sample data, respectively and red horizontal lines show the median values at each site. The sites are arranged in order by latitude going from north to south (plotted top to bottom): Victorville (VIC), Granada Hills (GRA), Ontario (ONT), Downtown LA (USC), California State University Fullerton (FUL), Compton COM), University of California, Irvine (IRV), San Clemente Island (SCI) and La Jolla (LJO). Note: Outliers are included in the statistics, but are not shown here. Figure S6 shows the same results, with outliers plotted. Observations for the ONT site began in Sept 2015, while all other results are annual averages.





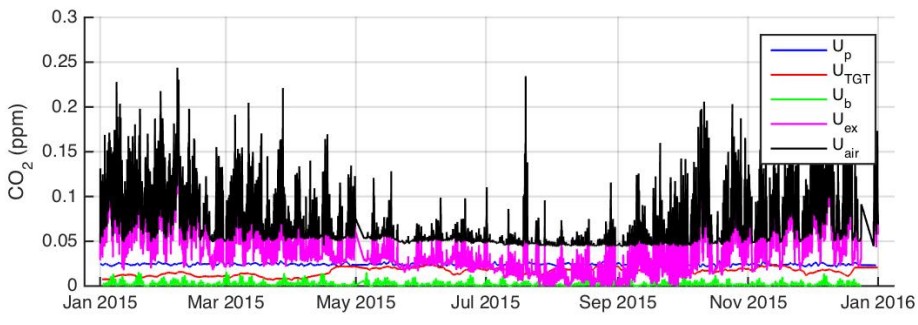

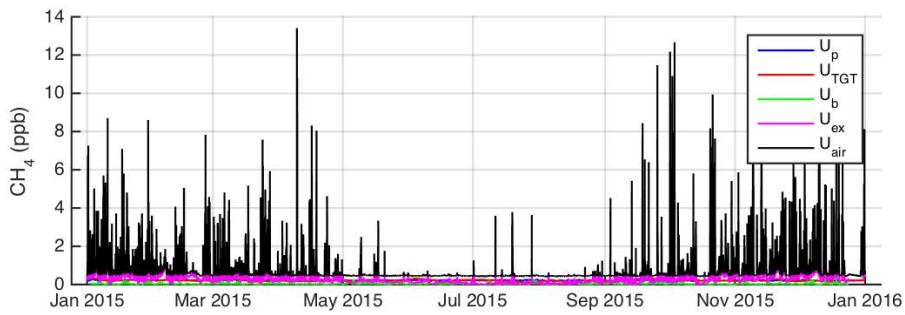

**Figure 7.** Time series of uncertainties in the La Jolla (LJO) air observations. $u_p$ is the analyzer precision, $u_{TGT}$ is the uncertainty derived from the target tank measurements, $u_b$ is the calibration baseline uncertainty, and $u_{extrap}$ is the extrapolation uncertainty, or the uncertainty due to the single-point calibration strategy. $u_{extrap}$ was estimated using a mean $\varepsilon$
5  for 9 analyzers see text and Supplemental materials). The total analytical uncertainty in the air measurements ($U_{air}$) is calculated as described by Eqs. 5-7.





**Table 1.** Site information for the Los Angeles Megacity Carbon Project surface network. Continuous measurements from the CIT, MWO, SBC, and PVP sites are not included as part of this study.

| Code | Full Site Name | Elev. (m agl) | Elev. (m asl) | Lat (°N) | Long (°W) | Analyzer |
|------|----------------|-------|-------|-----|------|----------|
| LJO | La Jolla (Scripps Pier)[1,2] | 13 | 0 | 32.87 | 117.25 | Picarro G2301 |
| CIT | Pasadena (Caltech, Arms Laboratory)[1,3,4] | 10 | 230 | 34.14 | 118.13 | Picarro G1101-i |
| CIT | Pasadena (Caltech, Millikan Library)[3,4] | 48 | 230 | | | Picarro G2401 |
| PVP | Palos Verdes Peninsula [1] | 3 | 320 | 33.74 | 118.35 | |
| MWO | Mt Wilson[1,5] | 3 | 1670 | 34.22 | 118.06 | Picarro G2201-i |
| VIC | Victorville* | 100/100/50 | 1370 | 34.61 | 117.29 | Picarro G2301 |
| SBC | San Bernardino *[1] | 27/58 | 300 | 34.09 | 117.31 | Picarro G2301 |
| GRA | Granada Hills*[1] | 51/51/31 | 391 | 34.28 | 118.47 | Picarro G2401 |
| USC | University of Southern California (Downtown LA)[1,4] | 50 | 55 | 34.02 | 118.29 | Picarro G2301 |
| | | | | | | Picarro G2401 |
| COM | Compton* | 45/45/25 | 9 | 33.87 | 118.28 | Picarro G2401 |
| FUL | Fullerton (CSU Fullerton)[1,4] | 50 | 75 | 33.88 | 117.88 | Picarro G2401 |
| IRV | Irvine (UC Irvine)[1,4] | 20 | 10 | 33.64 | 117.84 | Picarro G2301 |
| SCI | San Clemente Island* | 27 | 489 | 32.92 | 118.49 | Picarro G2401 |
| ONT | Ontario* | 41/41/25 | 260 | 34.06 | 117.58 | Picarro G2301 |

(*) Tower sites (VIC, GRA, COM, SCI, ONT, and SBC). Data from the SBC site are not included as part of this analysis, but this site also has an LGR $N_2O/CO$-EP analyzer on site measuring $N_2O$ and CO.

(1) Indicates flask collection site for $^{14}C$ observations (during current or past studies). The PVP site also includes a PP-Systems CIRAS-SC continuous $CO_2$ analyzer, which has been described previously (Newman et al., 2013).

(2) LJO: $CO_2$ flask observations began near this site in 1979, but are not included as part of this study. The Earth Networks
configuration was implemented in Jan 2012 and the inlet is located on Scripps pier.

(3) CIT: Flask observations at Caltech Arms began in 1998 and continuous measurements from the Picarro G1101-i (isotopic $CO_2$ analyzer) began in March 2001 have been described previously (Newman et al., 2008; 2013; 2016) and the Arms laboratory site also has an LGR $N_2O/CO$-EP analyzer on site measuring $N_2O$ and CO. A Picarro G2401 analyzer was installed at Millikan library in Dec. 2015 and includes a 4-corner rooftop sampling strategy identical to other rooftop sites.
Results from both CIT analyzers are not included in this study due to the short period of overlap in the records during 2015.

(4) Rooftop sites (USC, FUL, IRV, and CIT). All the rooftop sites described in this study follow a 4-corner sampling scheme similar to that described by McKain et al., 2015. One exception is the Caltech Arms laboratory site, which is a building site with a different configuration. Each analyzer samples from the 4 corner air inlets for roughly 15 minutes. Wind data are analyzed to compute hourly averages using only the data from the "upwind" corner of the building,
determined based on the highest wind speed measurement. Each analyzer is plumbed to the four corners of the building, with observations staggered by roughly 30 minutes. The USC site has two Picarro analyzers, model G2301 and G2401, which are referred to as USC1 and USC2, respectively, in the text. Rooftop inlet height indicates the total height above the surface (building + mast).

(5) MWO: Flask data have been collected by NOAA/ESRL since 2010 and are included as part of the background analysis in
this study. At the time of this study, there were three continuous analyzers installed at the CLARS facility near MWO, which are managed by the Air Resources Board: Picarro G2201-i analyzer measuring $CH_4/^{13}CH_4/CO_2$, Picarro G5310 measuring $N_2O/CO/CO_2$, and LGR Model 913-0015 measuring $N_2O/CO$.



**Table 2.** Statistics on the hourly average $CO_2$ observations for the nine sites shown in Figure 2 Annual average, SD, 16[th], 50[th] (median) and 84[th] percentiles, minimum and maximum values, and RMS were computed based on hourly average observations collected during calendar year 2015. Results are shown for all hours and midday hours (roughly 12-16:00 LT).

| $CO_2$ (ppm) | | VIC | GRA | ONT* | USC | FUL | COM | IRV | SCI | LJO |
|---|---|---|---|---|---|---|---|---|---|---|
| **All hours** | **mean** | 404.7 | 421.4 | 434.0 | 434.8 | 429.0 | 430.5 | 419.4 | 402.4 | 412.9 |
| | **1σ S.D.** | 3.8 | 17.0 | 25.2 | 31.2 | 23.3 | 30.3 | 19.3 | 4.3 | 14.9 |
| | **min** | 393.5 | 399.0 | 400.0 | 397.5 | 398.1 | 395.9 | 392.6 | 390.7 | 388.8 |
| | **16th** | 401.4 | 407.5 | 410.9 | 410.0 | 409.4 | 407.4 | 403.9 | 397.9 | 400.4 |
| | **median** | 404.4 | 416.3 | 428.1 | 424.4 | 421.3 | 419.5 | 413.0 | 403.1 | 407.3 |
| | **84th** | 407.9 | 435.8 | 457.1 | 460.7 | 451.2 | 457.4 | 436.9 | 406.1 | 428.8 |
| | **max** | 442.6 | 532.6 | 561.1 | 621.8 | 572.9 | 625.8 | 531.9 | 427.7 | 498.2 |
| | **RMS** | 0.8 | 3.9 | 3.4 | 7.0 | 5.1 | 6.9 | 4.2 | 1.0 | 3.4 |
| **Midday** | **mean** | 404.5 | 414.6 | 415.4 | 421.6 | 418.6 | 418.0 | 412.0 | 402.4 | 407.9 |
| | **1σ S.D.** | 3.7 | 12.8 | 11.8 | 17.5 | 14.9 | 16.9 | 13.5 | 4.4 | 10.6 |
| | **Min** | 395.9 | 399.2 | 400.0 | 397.5 | 398.7 | 396.9 | 392.6 | 391.2 | 392.5 |
| | **16th** | 401.3 | 404.9 | 406.2 | 408.2 | 407.2 | 406.0 | 401.4 | 397.9 | 398.5 |
| | **median** | 404.0 | 411.2 | 412.3 | 416.5 | 414.4 | 412.9 | 408.2 | 403.1 | 405.1 |
| | **84th** | 407.5 | 423.5 | 423.3 | 435.0 | 430.0 | 429.3 | 422.7 | 406.1 | 417.9 |
| | **max** | 442.6 | 521.6 | 487.8 | 530.0 | 498.8 | 558.1 | 494.9 | 425.2 | 468.1 |
| | **RMS** | 0.8 | 2.9 | 1.6 | 3.9 | 3.3 | 3.9 | 2.9 | 1.0 | 2.4 |

*Statistics for the ONT site are based on measurements from Sept-Dec 2015 only.





**Table 3.** Statistics on the hourly average $CH_4$ observations for the nine sites shown in Figure 2. Annual average, SD, $16^{th}$, $50^{th}$ (median) and $84^{th}$ percentiles, minimum and maximum values, and RMS were computed based on hourly average observations collected during calendar year 2015. Results are shown for all hours and midday hours (roughly 12-16:00 LT).

| $CH_4$ (ppb) | | VIC | GRA | ONT* | USC | FUL | COM | IRV | SCI | LJO |
|---|---|---|---|---|---|---|---|---|---|---|
| **All hours** | **mean** | 1902.5 | 2103.9 | 2126.1 | 2126.5 | 2079.3 | 2090.7 | 2045.7 | 1901.4 | 2009.5 |
| | **1σ S.D.** | 34.6 | 331.3 | 231.5 | 227.9 | 218.0 | 240.8 | 246.7 | 39.9 | 247.0 |
| | **min** | 1824.9 | 1828.8 | 1860.3 | 1864.9 | 1849.7 | 1848.6 | 1845.6 | 1823.3 | 1838.3 |
| | **$16^{th}$** | 1869.7 | 1927.3 | 1956.6 | 1946.2 | 1923.8 | 1914.9 | 1902.5 | 1866.4 | 1883.7 |
| | **median** | 1899.2 | 2003.6 | 2073.6 | 2047.2 | 1998.0 | 1998.4 | 1966.7 | 1897.6 | 1925.2 |
| | **$84^{th}$** | 1934.1 | 2228.1 | 2287.1 | 2321.3 | 2245.9 | 2296.3 | 2183.5 | 1928.9 | 2108.6 |
| | **max** | 2383.1 | 6946.1 | 8675.7 | 4511.1 | 4475.0 | 3788.6 | 8432.4 | 2348.3 | 5439.4 |
| | **RMS** | 7.4 | 76.0 | 31.4 | 51.1 | 47.5 | 55.1 | 53.5 | 8.9 | 55.7 |
| **Midday** | **mean** | 1898.6 | 1985.6 | 1990.7 | 2009.9 | 1978.2 | 1977.2 | 1962.6 | 1900.9 | 1935.3 |
| | **1σ S.D.** | 32.9 | 130.5 | 93.3 | 116.4 | 100.2 | 109.8 | 101.5 | 37.9 | 77.5 |
| | **min** | 1832.7 | 1828.8 | 1862.5 | 1864.9 | 1849.7 | 1848.9 | 1845.6 | 1824.7 | 1838.3 |
| | **$16^{th}$** | 1866.5 | 1902.9 | 1924.7 | 1923.6 | 1907.5 | 1901.6 | 1889.7 | 1866.4 | 1877.3 |
| | **median** | 1896.7 | 1949.8 | 1969.3 | 1973.0 | 1947.6 | 1943.7 | 1929.2 | 1897.3 | 1911.3 |
| | **$84^{th}$** | 1928.6 | 2056.3 | 2042.2 | 2095.7 | 2041.2 | 2050.3 | 2036.4 | 1927.9 | 1997.6 |
| | **max** | 2105.3 | 3567.8 | 2634.0 | 2677.9 | 2710.1 | 3109.6 | 2960.0 | 2231.4 | 2758.3 |
| | **RMS** | 7.0 | 29.9 | 12.7 | 26.2 | 22.0 | 25.2 | 22.1 | 8.4 | 17.5 |

*Statistics for the ONT site are based on measurements from Sept-Dec 2015 only.



**Table 4.** Statistics on the annual average $\Delta CO_2$ observations for eight of the sites shown in Figure 2. Annual average, SD, $16^{th}$, $50^{th}$ (median) and $84^{th}$ percentiles, minimum and maximum values, and RMS were computed based on the hourly average enhancements calculated during calendar year 2015. Results are shown for all hours and midday hours (roughly 12-16:00 LT). Results from USC are shown for the G2401 analyzer only. ONT results are not shown because measurements were only available from Sept-Dec 2015.

| $\Delta CO_2$ | | VIC | GRA | USC | FUL | COM | IRV | SCI | LJO |
|---|---|---|---|---|---|---|---|---|---|
| **All hours** | mean | 4.16 | 19.81 | 30.76 | 26.22 | 26.61 | 17.01 | 2.05 | 12.78 |
| | 1σ S.D. | 3.52 | 16.76 | 28.65 | 22.07 | 27.51 | 17.51 | 2.26 | 13.54 |
| | min | 0.00 | 0.01 | 0.31 | 0.02 | 0.05 | 0.00 | 0.00 | 0.00 |
| | 16th | 0.96 | 5.06 | 7.94 | 7.83 | 5.74 | 2.90 | 0.42 | 1.27 |
| | median | 3.38 | 15.10 | 21.44 | 18.83 | 16.34 | 10.66 | 1.48 | 7.55 |
| | 84th | 7.24 | 35.02 | 53.62 | 46.76 | 49.85 | 32.52 | 3.36 | 26.60 |
| | max | 40.94 | 126.66 | 222.78 | 171.36 | 203.71 | 126.97 | 22.68 | 93.31 |
| | RMS | 0.96 | 5.38 | 9.01 | 6.76 | 8.84 | 5.27 | 0.56 | 3.90 |
| **Midday** | mean | 3.90 | 12.76 | 18.75 | 16.30 | 15.16 | 10.03 | 1.98 | 7.81 |
| | 1σ S.D. | 3.22 | 11.04 | 15.94 | 13.76 | 15.24 | 11.86 | 2.05 | 9.65 |
| | min | 0.00 | 0.01 | 0.70 | 0.02 | 0.39 | 0.01 | 0.00 | 0.00 |
| | 16th | 0.98 | 2.93 | 6.17 | 5.57 | 4.83 | 1.42 | 0.49 | 0.74 |
| | median | 3.36 | 10.39 | 13.77 | 12.21 | 10.13 | 5.84 | 1.51 | 3.43 |
| | 84th | 6.43 | 22.29 | 31.09 | 25.96 | 24.20 | 18.85 | 3.08 | 16.73 |
| | Max | 40.94 | 99.38 | 125.14 | 94.53 | 153.23 | 93.41 | 21.52 | 66.63 |
| | RMS | 0.88 | 3.52 | 5.04 | 4.26 | 4.91 | 3.51 | 0.51 | 2.65 |




**Table 5.** Statistics on the annual average $\Delta CH_4$ observations for eight of the sites shown in Figure 2. Annual average, SD, $16^{th}$, $50^{th}$ (median) and $84^{th}$ percentiles, minimum and maximum values, and RMS were computed based on hourly average enhancements calculated during calendar year 2015. Results are shown for all hours and midday hours (roughly 12-16:00 LT). Results from USC are shown for the G2401 analyzer only. ONT results are not shown because measurements were only available from Sept-Dec 2015.

| $\Delta CH_4$ | | VIC | GRA | USC | FUL | COM | IRV | SCI | LJO |
|---|---|---|---|---|---|---|---|---|---|
| **All hours** | mean | 29.14 | 217.93 | 223.66 | 183.27 | 188.72 | 144.49 | 25.82 | 132.10 |
| | 1σ S.D. | 26.13 | 335.50 | 212.60 | 209.17 | 219.63 | 196.09 | 32.83 | 251.93 |
| | min | 0.02 | 0.02 | 0.09 | 0.02 | 0.02 | 0.01 | 0.00 | 0.00 |
| | 16th | 7.06 | 40.39 | 60.23 | 40.91 | 30.63 | 21.62 | 3.56 | 9.54 |
| | median | 22.32 | 121.01 | 148.00 | 106.12 | 100.56 | 74.05 | 15.20 | 46.60 |
| | 84th | 50.34 | 330.38 | 402.54 | 331.11 | 372.08 | 267.85 | 44.26 | 220.54 |
| | max | 481.64 | 5080.21 | 2608.17 | 2610.95 | 1617.43 | 2814.2 | 449.34 | 3516.8 |
| | RMS | 6.92 | 63.79 | 61.37 | 48.57 | 52.86 | 38.71 | 5.93 | 34.60 |
| **Midday** | mean | 24.53 | 100.04 | 113.55 | 85.24 | 84.11 | 72.92 | 24.16 | 54.90 |
| | 1σ S.D. | 23.45 | 118.90 | 105.76 | 88.78 | 98.04 | 92.55 | 29.93 | 69.00 |
| | min | 0.07 | 0.02 | 0.09 | 0.15 | 0.02 | 0.01 | 0.02 | 0.02 |
| | 16th | 5.48 | 23.35 | 35.75 | 23.98 | 18.04 | 12.29 | 3.59 | 5.86 |
| | median | 18.18 | 70.37 | 81.36 | 58.55 | 51.98 | 41.20 | 15.18 | 27.43 |
| | 84th | 42.15 | 162.80 | 184.90 | 138.30 | 140.54 | 128.30 | 40.24 | 114.67 |
| | max | 190.22 | 1659.42 | 786.13 | 805.08 | 860.92 | 1056.50 | 317.35 | 510.06 |
| | RMS | 5.61 | 36.70 | 33.39 | 27.24 | 31.05 | 27.35 | 7.62 | 19.58 |



**Table 6.** Estimates of $\varepsilon$, the slope component of $u_{extrap}$, the uncertainty due to the single-point calibration strategy, based on laboratory experiments and field data. Two values are from analyzers deployed at the LJO and VIC sites using limited measurements of a high mole fraction tank deployed in the field. Additional estimates of $\varepsilon$ were collected from laboratory calibrations at NOAA/ESRL using CRDS analyzers with similar model numbers (see Figures S1 and S2). Two different sets of calibration results are available for one of the analyzers (CFKBDS-2007a and -2007b).

| no. | Analyzer model | $CO_2$ slope ($\varepsilon$) (ppm/ppm) | $R^2$ | $CH_4$ slope ($\varepsilon$) (ppb/ppb) | $R^2$ |
|---|---|---|---|---|---|
| 1a | CFKBDS-2007a | 0.0064 | 0.98 | 0.0038 | 0.99 |
| 1b | CFKBDS-2007b | 0.0056 | 0.96 | 0.0029 | 0.98 |
| 2 | CFKBDS-2008 | 0.0059 | n/a | 0.0029 | n/a |
| 3 | CFKBDS-2059 | 0.0017 | 0.56 | 0.0031 | 1.0 |
| 4 | CFKADS-2067 | 0.0024 | 0.82 | 0.0029 | 0.99 |
| 5 | CFKBDS-2091 | 0.0005 | 0.11 | 0.0031 | 0.99 |
| 6 | CFKBDS-2096 | 0.0017 | 0.77 | 0.0028 | 0.99 |
| 7 | CFKBDS-2099 | 0.0001 | 0.07 | 0.0026 | 0.99 |
| 8 | LJO | 0.0027 | n/a | 0.0012 | n/a |
| 9 | VIC | 0.0018 | n/a | 0.0060 | n/a |





**Table 7.** Estimates of the uncertainty in the single-point calibration method based on laboratory experiments and limited measurements of high concentration standards deployed in the field. Corrections to the air data estimated assuming a 100 ppm $CO_2$ and 4 ppm $CH_4$ enhancement above the "near ambient" calibration standard and various estimates for $\varepsilon$.

| | $CO_2$ slope ($\varepsilon$) (ppm/ppm) | $CH_4$ slope ($\varepsilon$) (ppb/ppb) | $CO_2$ correction given a 100 ppm enhancement (units: ppm) | $CH_4$ correction given a 4 ppm enhancement (units: ppb) |
|---|---|---|---|---|
| **Analyzer 1 (a and b)** | 0.0060±0.0005 | 0.0033±0.0006 | 0.60±0.05 | 13.2±2.4 |
| **Analyzers 1-7\*** | 0.0026±0.0024 | 0.0030±0.0002 | 0.26±0.24 | 12.0±0.8 |
| **Analyzers 1-9\*** | 0.0025±0.0021 | 0.0031±0.0012 | 0.25±0.21 | 12.4±4.8 |
| **LJO analyzer** | 0.0027 | 0.0012 | 0.27 | 4.8 |
| **"Alternate calibration method"\*\*** | n/a | n/a | 0.2 | <6 |

\*Two different calibrations available for analyzer 1 (a and b) were averaged together first, to get a single value for analyzer CFKBDS-2007, before averaging with the other analyzers. See Table 6.
\*\* See Appendix A2 and Figures A2 and A3.





**Table 8.** Summary of the average uncertainty estimates for the LJO analyzer during 2015. Each component of the total measurement uncertainty ($U_{air}$), where $u_{h2o}$ is the uncertainty due to the treatment of water vapor, $u_{TGT}$ is the uncertainty derived from the target tank measurements, $u_p$ is the analyzer precision, $u_b$ is the calibration baseline uncertainty, and $u_{extrap}$ is the extrapolation uncertainty, or the uncertainty due to the single-point calibration strategy. $u_{extrap}$ was estimated using a mean $\varepsilon$ for 9 analyzers see text and Supplemental materials). $\overline{U}_{air}$ is the total mean uncertainty in the air measurements and is calculated as described by Eqs. 5-7, and $\overline{U}_{Enhancement}$ is the average uncertainty in the enhancement.

| Uncertainty Estimates | $CO_2$ (ppm) | $CH_4$ (ppb) |
|---|---|---|
| $u_{h2o}$ | 0.0233 | 0.221 |
| $u_{scale}$ | 0.03 | 0.31 |
| $\overline{u}_{TGT}$ | 0.0166 | 0.2126 |
| $\overline{u}_p$ | 0.0242 | 0.2205 |
| $\overline{u}_b$ | 0.0028 | 0.0444 |
| $\overline{u}_{extrap}$ | 0.0477 | 0.4618 |
| $\overline{U}_{air}$ | 0.0699 | 0.7224 |
| $\overline{U}_{Enhancement}$ | 1.1 | 11.7 |



**Appendix A**

Here we describe: (A1) Data acquisition and QA/QC protocols; (A2) the "Alternate Calibration Method," which is a correction applied to the air data from LJO and VIC using a 2-point calibration method; (A3) Curve fitting parameters for the CCGCRV software to estimate background.

**A1 Data Acquisition and QA/QC**

**A1.1 Data acquisition**

The LA in situ network includes sites deployed by Earth Networks, Inc. (http://www.earthnetworks.com/OurNetworks/GreenhouseGasNetwork.aspx). The GCWerks software manages the data flow. This software was originally developed for use in the Advanced Global Atmospheric Gases Experiment (AGAGE)

network to provide a secure point of access to acquire data from remote instruments and has since been adapted for data management of CRDS analyzers using a Linux-based system (http://www.gcwerks.com). GCWerks also assists in the operation of the LA Megacity sites by reporting instrument diagnostics, sending user defined email alarms, applying pre-defined automated filter and flagging criteria to remove data impacted by instrument errors, and allowing graphical display of results. Several instrument diagnostics and quality control parameters including pressures, flow rates, cycle time, cavity

pressure and cavity temperature are monitored in GCWerks to track the instrument status and measurement quality. The calibration correction (using the default, single-point calibration method) is also applied by GCWerks based on daily runs of the calibration standard tank (see Section 2).

A master copy of GCWerks stored on the EN server ingests and merges the high-frequency CRDS greenhouse gas mole fractions and meteorological data, as well as the calibration information, port assignments, tank assignments, etc., on

an hourly basis. The imported data are stored as binary strip-chart files, which are over 30 times smaller than the hourly Picarro files for fast data processing and copying. Metadata associated with the LA measurement sites is also stored and maintained on the EN server, including the most current versions of the ports.log and standards files. The standards file contains all the information about the assigned calibration tank values. The ports.log indicates the assignments for each of the air inlets and calibration standards, as well as the time period to reject before calculating averaged values from the native

resolution Picarro data. Daily Picarro results are exported into yearly .csv files, which are checked offline against the standards files for additional updates (e.g. calibration tank assignments and data that requires manual flagging). Remote copies of GCWerks are also run by site operators for data exploration via VNC connection.

**A1.2 Data QA/QC and automated filters**

GCWerks applies some basic automated quality control flags, which filter and/or reject some of the Level 0 data





points. We apply automated filters to the high-frequency data in GCWerks before subsequent processing. Filtered data are displayed in GCWerks and are configured in the gcwerks.conf file. The filter criteria are listed in Table A1 and apply to individual high-resolution data points (letter codes: P, S, W, C, T).

Some data are also rejected to account for the stabilization period after the inlet is switched. The first 10 minutes of data are excluded from further analysis after switching from a calibration tank to an air inlet and the first minute is excluded when switching between air inlets. This allows for flushing time for the tank regulator and plumbing and stabilization of the measurement after valve switching. Data that is automatically or manually flagged or rejected data are also displayed in GCWerks with letter codes (F= flagged data and x = rejected data).

Manual data flagging is occasionally needed to ensure good quality control of the greenhouse gas data (e.g. to
ensure the flags were applied correctly in the previous step, or to address technical issues or instrument errors that are outside the scope of the automated filters, such as when a field technician visits the site and impacts the regular sampling protocol). After automated filters are applied, the data are screened manually using a parameter called N filtered in GCWerks to identify instances where automated filters have been applied. Data flags are applied by Earth Networks based on recommendations from the LA Megacity Data Working Group, a team of scientists from NASA's Jet Propulsion Laboratory,
Scripps Institution of Oceanography, National Institute of Standards and Technology, and Earth Networks. The manual flags are applied on the EN server to indicate those data that are not recommended for further scientific evaluation or interpretation within the scope of the project.

The corrected data is generated by GCWerks at 1-minute average intervals and generated as a .csv file for export and further analysis outside of the GCWerks framework. These .csv report files are uploaded to a primary Earth
Networks/GCWerks server and are synced nightly and later used to compute the hourly average (Level 3) product.

The measurements discussed in Sections 3 and 4 are mainly the 1-hour average $CO_2$ and $CH_4$ air observations from each individual inlet height (for tower sites), and from a combination of the 4 corner inlets (for rooftop sites), which we term Level 3 data.  For rooftops, we used a method similar to McKain et al. (2015) to calculate the 1-hour average air observations using only "upwind" observations (determined from the 1-minute average data).  Additionally we used wind
speed and direction observations to verify the "upwind" side of the building.  We constructed an "upwind" index for each 1-minute $CO_2$ or $CH_4$ observation.  The measurement was determined to be "upwind" if the building corner had the highest wind speed and the wind direction also corresponded to the same side of the building.

**A2 Alternate calibration method**

An "Alternate Calibration Method" was explored using a linear fit between two tanks (one "near-ambient" tank and
one ''high concentration'' standard tank) where data was available. Each calibration run is used to derive a slope and





intercept, which are then interpolated in time:

$$Xcorr\ alternate = m * X'_{air} + b \qquad \text{(Eq. A1)}$$

where the slope m and intercept b. The Alternate Calibration Method uses the high concentration standard to determine the slope, m, and intercept, b, while the default (single-point calibration) method assumes a zero reading at zero measurement.

Therefore, the Alternate Method becomes equivalent to the default calibration method if $b = 0$ and the slope m does not vary with mole fraction, so that $m = 1/S_{cal}$ for all points. Both methods of calibration assume linearity, in that the slope m (or 1/S), is a constant over all mole fractions.

Air data from the LJO and VIC sites were corrected using the two methods to quantify the effect of different calibration methods on the final air mole fraction data. Both of these sites had limited measurements of a high mole fraction

(span) tank available at the time of this study. Figures A2 and A3 show the difference in air data from the LJO analyzer corrected with the 2-point calibration method and the single-point calibration method for $CO_2$ (upper panels) and $CH_4$ (lower panels). Figure A3 shows similar results for air data collected from the VIC analyzer. Overall, the single-point calibration method underestimates the $CO_2$ levels by about 0.2 ppm out of 100 ppm and underestimates the $CH_4$ levels by about 6 ppb out of 6000 ppb, or about 1 part in 1000 for the LJO site (Figure A2). The results were similar when the same analysis was

performed using air data from the VIC analyzer (Figure A3).

**A3 Curve fitting parameters for CCGCRV**

For the SCI, VIC, and LJO in situ observations, the following fit parameters were used in CCGCRV: short-term cutoff filter=30 days, long-term cutoff filter=667 days, npoly=3, nharm=4. We fit the data iteratively using the curve-fitting algorithm by continually excluding outliers greater than $\pm 2\sigma$ from the smooth curve fit until no more outliers were found. A

multi-species filter was also applied, so that if either $CO_2$ or $CH_4$ were outliers from the smooth curve, then both observations were omitted.

For the MWO flask data, the following fit parameters were used in CCGCRV: short-term cutoff filter=30 days, long-term cutoff filter=667 days, npoly=3, nharm=4. In order to filter the time series, we fit the data iteratively, continually excluding outliers greater than $\pm 2\sigma$ from the smooth curve fit until no more outliers could be removed, as described above.





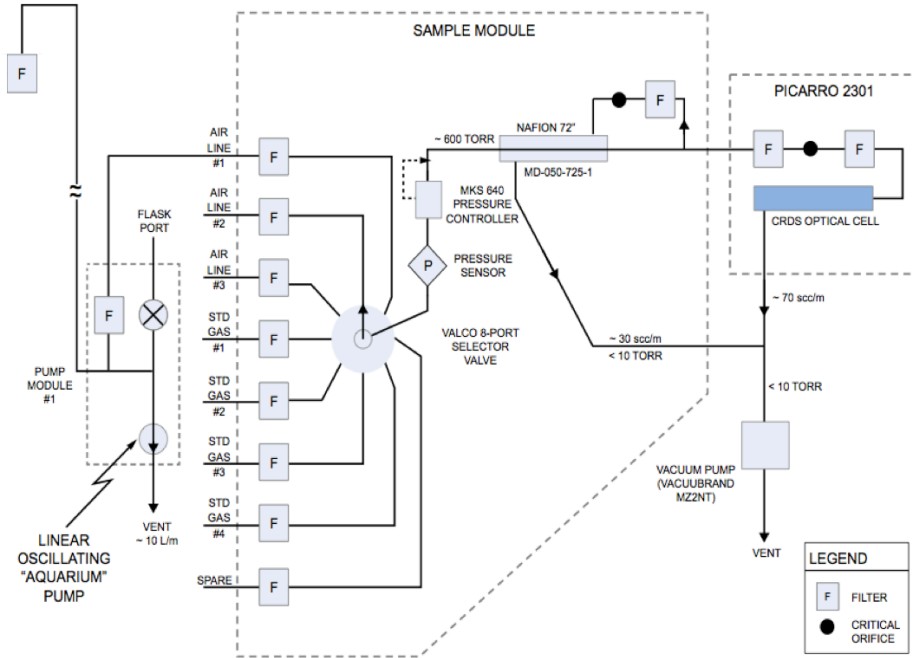

**Figure A1.** Diagram showing the standard gas-handling configuration for an Earth Networks greenhouse gas monitoring tower sampling from a single air inlet (see text). Figure and caption adapted from Welp et al. (2013).



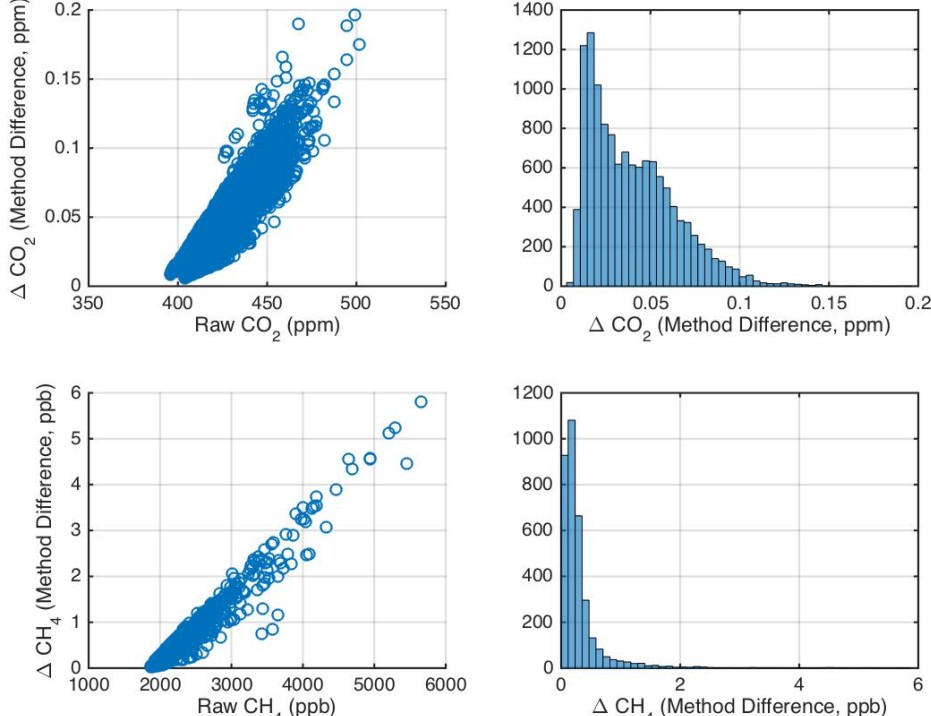

**Figure A2.** "Alternate Calibration Method" plotted versus raw $CO_2$ and $CH_4$ air sample measurements for the LJO site. A limited number of measurements of a high mole fraction $CO_2$ and $CH_4$ standard were available at the LJO field site between October 2015 and March 2016. "Method Difference" indicates the difference in the correction of the air data using a single-point relative to a 2-point calibration. Histograms show that the uncertainty associated with using a single-point calibration is <0.2 ppm $CO_2$ and <4 ppb $CH_4$ for air measurements collected during this period.





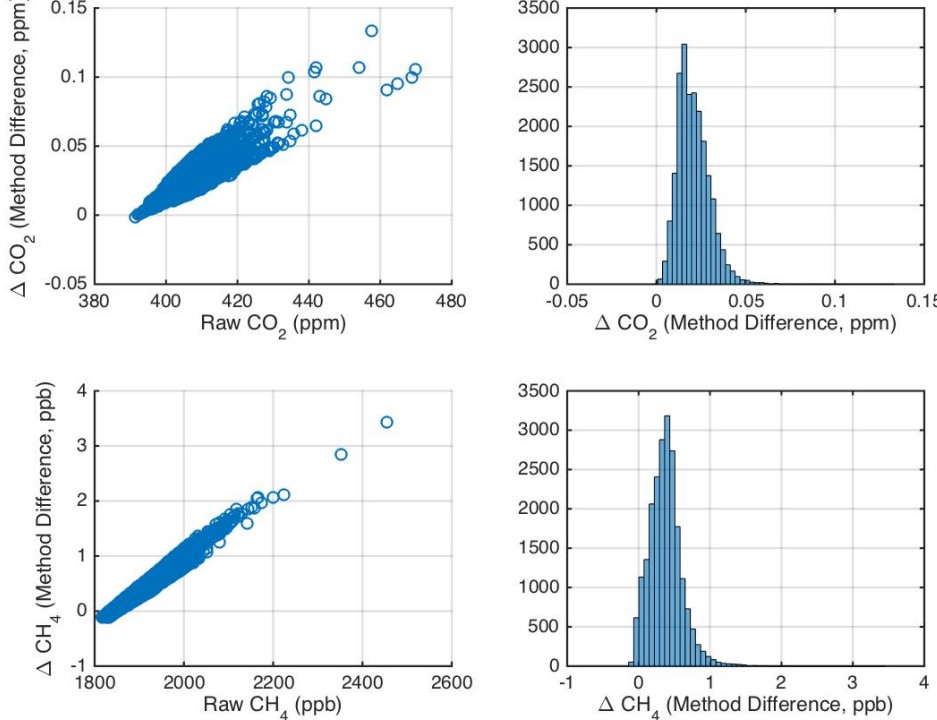

**Figure A3.** Same as Figure A2, except for the VIC site, which also had limited measurements of a high mole fraction $CO_2$ and $CH_4$ standard available at the time of this study. Histograms show that the uncertainty associated with using a single-point calibration is <0.15 ppm $CO_2$ and <4 ppb $CH_4$ for the majority of the air measurements during this period. Overall, the corrections are slightly smaller because the $CO_2$ and $CH_4$ enhancements at VIC are smaller relative to LJO.





**Table A1.** Automated flagging and filtering criteria applied to the CRDS measurements using the GCWerks software. Data meeting the filter criteria are flagged to identify periods when the CRDS analyzer may be subject to large errors and greenhouse gas observations collected during such periods were excluded from further analysis. Flags and filters are applied to high-resolution (roughly 2.5 second) CRDS readings.

| Symbol | Filter name | Frequency | Criteria |
| --- | --- | --- | --- |
| **P** | Cavity pressure[1] | 3-5 sec | Cavity pressure out of range (139.9-140.1 Torr) |
| **T** | Cavity temperature | 3-5 sec | Cavity temperature out of range 44.98-45.02 °C |
| **W** | High water | 3-5 sec | Water value too high (>10%) |
| **C** | Cycle time[1,2] | 3-5 sec | Cycle time too high (>8 seconds/cycle) |
| **S** | Standard deviation | Varies | For measured compound or water only |
| | | | Calibration standards only: applies to 20-min window, filters values >3σ |
| | | | Air data only: applies narrow 2-minute moving window, filters values >10σ |

5  (1) Indicates the filter contains a user defined or specified value within GCWerks.

(2) Cycle time is defined as the time between subsequent trace gas measurements. Note: For periods in which a data point is separated from adjacent points by more than the specified maximum cycle time, all 3 points will be filtered. For LA the default maximum for cycle time filter is 8 seconds. Representatives from Picarro Inc. recommend the cycle time value should not normally exceed 5 seconds (C. Rella, *personal communication*).

10  (3) For each measured compound (and water) any data points outside of the user-defined number of standard deviations are automatically filtered. The filter is applied recursively until no more points are filtered in each mean. For calibration tank measurements, the remaining 10-minute period (after the initial 10 minute rejection period) is filtered at once (i.e. no moving window) using a 3σ SD filter. For air data, a narrow 2-minute moving window is used to only filter extreme outliers (>10σ SD, default) that may result from instrument errors. The moving windows overlap 1-minute, and the center 1-minute is
15  filtered, while ends (first and last 30 seconds of each air measurement) are not filtered.