# Peer review of "Carbon Dioxide and Methane Measurements from the Los Angeles Megacity Carbon Project: 1. Calibration, Urban Enhancements, and Uncertainty Estimates"

_Atmospheric Chemistry and Physics, 2016_

## Referee Comment (RC1) · Anonymous Referee #1 · 16 Nov 2016

**1   Overview:**

Review of "*Carbon Dioxide and Methane Measurements from the Los Angeles Megacity Carbon Project: 1. Calibration, Urban Enhancements, and Uncertainty Estimates*" by Verhulst *et al.*

Verhulst *et al.* presents a new greenhouse gas monitoring network in Los Angeles and is generally well-written. I would recommend publication with some minor revisions to clarify several points.

[Figure]

**2 Minor comments:**

**2.1 IRV site**

On page 11 (3rd paragraph), the authors discuss why IRV is inconsistent with the results reported by Feng et al. (2016) and attribute it to the spring/summer data used in Feng et al. It seems that this should be easy to test. Are the real measurements from spring/summer consistent with the results from Feng et al?

**2.2 Background values**

On page 15 (1st paragraph), the authors discuss their background concentration selection. Criteria 2 seems like it would be difficult to obtain due to meteorology. Won't daytime PBL growth will impact the concentrations? Even if you have no local sources you could still have fairly large hour-to-hour variations due to meteorology. This criteria would make sense if you were, say, comparing the observed and modeled concentrations (if the residuals were small then you've captured all of the sources & meteorology), but that would be more involved than the implied goal of this section: to derive a simple background estimate to facilitate near-real time analysis.

Following onto this, in the following paragraph it looks like the authors use 6-hours as their time range. A plot of the diurnal cycle (maybe monthly averaged) could be useful to understand the time ranges that should be used.

**2.3 Background uncertainty**

It seems like using some combination of background sites could be a better method for getting the background uncertainty. For example, computing a running standard

deviation (over a moving window) of the differences between the various background sites. How would that compare to your background uncertainty estimate?

It would also be nice to see a bit more discussion on the background uncertainty, especially since it is found to be the largest source of uncertainty. There are about 7 pages devoted to the analytical uncertainty and 2 sentences devoted to the background uncertainty.

**3  Specific comments:**

Page 2, Line ∼25: Maybe include a reference to the San Francisco/Berkeley network (e.g., Shusterman et al., 2016).

Page 11, Line 30: wouldn't lower night time PBLs mean higher sensitivity? Isn't this a big part of why we have a diurnal cycle that is out of phase with emissions (emissions peak during the day but surface concentrations are near a minimum due to the PBL).

Page 18, Line 10: wouldn't the converse [that SCI and LJO are not relevant background choices when flow is from the continent] also be true? Or the more general statement that the background data used should be based on the prevailing flow patterns (similar to how McKain et al., 2015 chose their background values)?

---

## Short Comment (SC1) · 15 Dec 2016

I had a question about the use of the "annual averages" which was for example 2009.9 +/- 116.4 ppb $CH_4$ at one site. My assumption would be the standard deviation is calculated on a set of annual averages, but it seems to be the standard deviation of the 1-hour data (or similar). Shouldn't this be referred to as the mean 1-hr average?

Some additional minor comments:

Eq. 1 gives the equation for Scal [as used in Section 6.1.5] not S

[Figure]

Eq. 11: the overbar should extend over the squared - otherwise the square root and squared cancel

The variable names are a little confusing on first read. There are six terms: uncorrected values, corrected values, and the assigned tank values + each of these with the "cal" subscript. Three possible ideas to help clarify: - Page 9 / L29 sentence should end with "of the calibration standard" - Supplement: there is a reminder what $X_{assigncal}$ means, but a reminder of the meaning of $X_{assign}$ and $X_{corr}$ would also be helpful - Figure A3: "uncorrected" instead of "raw" so one knows for sure it is referring to the same thing

Page 27: unclear what "fitted curve residuals" refers to

Figure S2 caption: "same suite of tanks as Figure S1"

Page 27 /L18-26: Could be reworded for clarity (the point about CH4 standards being stable is made three times)

Figure 7: This is an interesting idea for a graph but unfortunately it is difficult to make out the details, especially for the CH4 plot. Also items in the legend are all uppercase which is not consistent with other usage

I am not sure the use of overbars in Table 8 and its caption is fully consistent

---

## Referee Comment (RC2) · Anonymous Referee #2 · 31 Jan 2017

In the manuscript "Carbon Dioxide and Methane Measurements from the Los Angeles Megacity Carbon Project: 1. Calibration, Urban Enhancements, and Uncertainty Estimates," Verhulst et al. describe a network of 11 $CO_2$ and $CH_4$ measurements on towers and building tops in and near LA. They carefully detail their site selection and calibration strategies. In LA, the choice of background is particularly important because there is predominantly flow from the ocean in the spring/summer, whereas in the fall/winter the flow pattern in more variable. In the end, they found the SCI site to best represent the background air and used that in the calculation of $CO_2$ and $CH_4$ enhancements. The median midday urban enhancements reported are large (5.8 -

13.8 ppm CO2 and 40.2 - 81.4 ppb CH4 for the urban/suburban sites)compared to both the estimated variability in the background and to the measurement uncertainties. Of course, the magnitude of these enhancements is dependent on the measurement heights, and the readers should be reminded of that. The authors also carefully describe the measurement uncertainties, and alternate calibration strategies.

This paper is very well-written and within the scope of Atmospheric Chemistry and Physics.

I recommend this paper for publication in Atmospheric Chemistry and Physics, with technical/minor revisions, listed below.

Abstract: "large CO2 and CH4 enhancements relative to a marine background to estimate"- remove "to"

Abstract: "roughly 20 ppm CO2 and 150 ppb CH4 during all hours" – specify averaged over all hours? Higher enhancements at night.

Abstract: "The largest component of the measurement uncertainty is due to the observations being elevated relative to the single-point calibration method." I understand the point of this sentence, but don't understand "elevated". Reword?

Section 2.3: also have G2201-i and G1101-i instruments listed in table, but not in text

Section 2.3: Did I miss how the instruments were calibrated in the lab, prior to deployment? If they weren't, I would expect significant slope errors.

Section 2.4: The sentiment of the first sentence is repeated a few sentences later -reword?

Section 2.4: Were the Scripps tanks calibrated to X2007 for CO2 and X2004A for CH4, like the NOAA tanks?

Section 2.4, second paragraph: Missing period in the last sentence.

Section 2.4: The high mole fraction tank is mentioned in passing in the third paragraph. Suggest mentioning prior to this.

Section 3.1: First sentence is pretty obvious. Reword?

Tables 2 and 3: Could these be changed into a figure? Hard to process as tables.

Section 4.1: Why are the criteria for SD within one hour 0.3 ppm and 5 ppb CH4? And why hour-to-hour difference 0.25 ppm CO2 (why in general, and why slightly different than one-hour SD?) Why no hour-to-hour criteria for CH4?

Section 4.3: It was surprising to me that after such detailed analysis, you ended up using SCI as background for all times.

Section 6.1.6, second paragraph: "CH4 standards are very stable" is said three times within this paragraph – suggest rewording.

Section 6.1.6: "routine field measurements of standard tanks to date do not indicate significant drift in either gas" – do you determine this through the target tank results? Clarify.

Consider moving Tables 5 and 6 to supplement.

---

## Referee Comment (RC3) · Anonymous Referee #3 · 2 Feb 2017

General Comments

This manuscript describes the instrumentation and calibration methods used for the Los Angeles Megacity surface observation network. The authors present a method for calculating background mole fraction conditions using sites outside the domain of the South Coast Air Basin and utilize this background to calculate urban enhancements above background. Finally, they describe the components of the analytical uncertainty that affect the observations. In general the manuscript is well written and the experimental methods are described in detail. This is an important methodological descrip-

tion of a major project and is therefore an important contribution and within the scope of Atmospheric Chemistry and Physics.

While I have a few suggestions regarding the background selection criteria and numerous technical suggestions, the bulk of the manuscript is technically sound and I would recommend publication after these minor concerns are addressed.

One complaint is that there is an Appendix A1-3 as well as Supplemental Materials. It would simplify the manuscript if the authors changed the supplemental information to an Appendix B, or if they added Appendix A1-3 to the Supplemental Materials. It seems overly complicated to have both an Appendix and supplemental information.

Specific Comments

Abstract Ln 30: The time span should be explicitly stated, not "roughly". This can be fixed by removing the word "roughly".

Pg 2, Ln 13: "Carbon fluxes can be estimated using "top-down" or "bottom-up" methods." – Why use "or" in this statement? It is possible to use both top-down and bottom-up methods to evaluate emissions, and it is also possible to use both methods in an iterative process that leads to an optimized solution. I think a discussion of how these methods can be used together is missing from this paragraph.

Pg 2, Ln 30-Pg 3 Ln 3: The statements starting with "More information flow…" seems like it doesn't belong in the introduction since its making a recommendation. Looking at the conclusions, this information is there prominently and that is what it should be so readers pay attention to it, but I don't think you need to also have the recommendation in the introduction. It would be fine to describe how methods should be documented and how there is a long latency in most observational networks currently (etc) since that is the current state of affairs, but leave the recommendations for the discussion/conclusions (in my opinion).

Pg 3, Ln 7: You don't need to cite the URL a second time since it was listed on the prior

page.

Pg 3, Ln 7: Is the LA project considered a "pilot" project? What would a full blown project be if the LA project is only a "pilot"?

Pg 3, Ln 14: Is there a more accurate population estimate than "> 15 million"? Could it be 20 million? What does the US Census Bureau say?

Pg 3, Ln 22: The sentence starting with "Urban and suburban areas. . ." seems redundant to the next paragraph where this is discussed in greater detail. I seems like you could just remove this sentence and have this paragraph describe the met conditions and the topography, while the next paragraph describes possible emission sources & complexities. Pg 4, Ln 1-5: You should also list the complication of using ethanol in gasoline. This will have a different d13C signature than fossil gas because its derived from C4 grass (corn) and is an additional important complication for understanding $CO_2$ emissions.

Pg 5, Ln 1: "Enhancements" should be defined when you first use it. Also, "robust" seems to be an odd descriptor. Perhaps "large enhancements" would be better?

Pg 5, Ln 2-3: The word "roughly" is redundant to listing a range of mole fractions. Also, for CH4 surface mole fractions it would be better to list a range like was done for CO2 instead of saying "10's to 100's of ppb" in order to keep the text internally consistent.

Pg 5, Ln 31: The word "extensible" seems odd. Would it be better to say ". . .are intended to PROVIDE A BLUEPRINT FOR other surface observation. . ."? Also, don't you provide recommendations about how to improve your calibration setup? From the conclusions: "In the near future, the LA measurement network will begin using analyzer specific estimates of the correction factor based on periodic measurements with high mole fraction tanks, which will allow correction of the random and systematic components of the uncertainty associated with the single-point calibration strategy." It might be nice to mention that you will be providing suggestions for future deployments
as well here.

Methods: Pg 6, Ln 19-20: missing the word 'to' here: "…were often inaccessible due TO permitting or other restrictions."

Pg 7, Ln 23: What temperature is the heated box kept at? Is it kept at a constant temperature (ie – is there a fan that cools it down when its too hot) or is there just a heater for when it is cold?

Pg 9, Ln 17: Forgot a period at the end of the sentence.

Pg 9, Ln 19-20: I think this is the first reference to high mole fraction tanks, but the filling procedure for these tanks has not been described. A brief description of how high mole fraction tanks are filled and calibrated is warranted, probably as part of the preceding paragraphs that talk about the filling and calibration of ambient mole fraction tanks.

Pg 9, Ln 20-23: I assume that the first 10 minutes of calibration data are rejected because that is longer than the turnover time of air in the CRDS and the instrument is coming to an equilibrium. If so, it would be nice to state this.

Pg 9, Ln 14: This calibration method assumes a linear response in the analyzer, and I think this is an important assumption to state clearly in the manuscript somewhere in this section. Also, I think this sentence is slightly misworded: "The sensitivity (S) of the high mole fraction tank is also tracked over time, providing a check on the analyzer stability at higher mole fractions." I don't think you are evaluating the "stability" but instead you are evaluating the "accuracy."

Pg 10, Ln 23-24: The first line of the Results is incomplete: "Atmospheric CO2 and CH4 mole fractions can vary on timescales ranging from less than 1 hour, to annual, and inter-annual cycles." Of course there can also be decadal, centennial, and millennial variations as well. Perhaps this could be changes to say that "The atm CO2 & CH4 in our data sets contain variations from less than one hour to annual and iner-annual", or

something like that.

Pg 10, ln 28: "levels" should be "mole fractions" for better clarity.

Pg 11, ln 1: You don't need the "roughly" since you define the time range and are using hourly data, unless the hourly window changes from day to day.

Pg 11, ln 1: Also, is the S.D. defined using the hourly data? Its clear that these statistics are for "afternoon hours," but it is not clear that it is using hourly data (vs minute average data, or daily afternoon averages). This could easily be clarified in the text.

Pg 11, Ln 13-14: "We find that SCI and VIC are the cleanest sites in terms of their annual $CO_2$ and $CH_4$ variability." – It is unclear what "cleanest" means in this context. Does it mean they have the lowest variability?

Pg 12, Ln 11: "...controlling the variability of $CO_2$ (and $CH_4$) observations..." should be "...controlling the variability of TRACE GAS observations..."

Pg 12, Ln 14-15: This brief description of a box model framework for how the PBL affects mixing ratios is missing the assumption that transport in and out of the boundary layer remains approximately constant.

Pg 12, Ln 17-18: I disagree with this statement: "The reduced variability in the $CO_2$ and $CH_4$ observations during midday hours is in part due to the stability of the PBL depth during the mid/late afternoon." The stability is due to the larger size of the PBL during the middle of the day (compared to the night), not necessarily the "stability of the PBL depth." Because the PBL is larger, a particular flux will be more dilute in a larger 'box' and therefore you'll see lower variability. If my understanding is incorrect it would be good to have a citation here to back this claim up. After reading a few more lines I see this is mentioned on lines 22-24. However, I think the size of the PBL is a bigger factor than the "stability of the PBL depth."

Pg 12, Ln 31: Again, "$CO_2$ (and $CH_4$)" should be "trace gas." Wind speed affects other trace gases as well (CO, NOx, etc).

Pg 13, Ln 10: The use of capital delta CO2 or capital delta CH4 is slightly contro-versial. The problem is that the capital delta symbol is used to represent radiocarbon measurements, so its use to represent local enhancement could be confusing. The research community has not agreed on what the appropriate symbol to use to repre-sent local enhancement of GHGs, so the authors should feel free to use what they feel is appropriate. However, they should be aware that this is something that should be agreed upon within the community and the authors should only use this symbol if they intentionally think this is how it should be displayed, because others will follow their ex-ample. Other options that I am aware of are writing out "CO2 enhancement" or "excess CO2" or "CO2-ex".

Pg 13, Ln 18: I don't know that you need to continually state that the units are ppm or ppb. Its clearly stated in multiple places before this.

Pg 13, Ln 19: The authors are calculating background mole fractions, not estimating them.

Pg 13, Ln 22-25: Listing the lat/lon is unnecessary since these sites are described in Table 1. Only list the lat/lon if the site is not already in Table 1. Also, the site codes should be used here instead of describing the sites.

Pg 14, Ln 15-16: Again, lat/lon in unnecessary and Table 1 should instead be cited.

Pg 15, Ln 9: The word "very" in unnecessary.

Pg 15, Ln 11-12: I wouldn't describe the selection criteria used by Thoning et al 1989 as "preliminary," its just what they used.

Pg 15, Ln 23: "1670 m agl" should be "1670 m asl", unless this site is on a VERY large tower. :-) The authors could also just refer to this as being on a mountaintop and refer to table 1 for the elevation.

Pg 16, Ln 1: MBL should be defined as Marine Boundary Layer.

Pg 16, Ln 13: "roughly" is not needed since the exact latitudes are listed.

Pg 16, Ln 13-14: The selection of these MBL latitudes deserves more explanation. Obviously 33.4N is the latitude of the SCB, so that makes sense. But why look at two other latitudes north of that, but not a latitude south of that? Also, 40.5N is ~770km north of the SCB, so is this really a good comparison to make?

Pg 16, Ln 25-26: This is surprising to me since I don't know of a mechanism that would draw CO2 down over land during the winter. My guess is that this difference may have to do with how the 2-D MBL was computed (there is greater smoothing in the 2D MBL curves than in their background curves), or perhaps the latitude chosen. Looking at Fig 4 (top left) closely, the springtime MBL at 33.4 is closer to the authors background estimates than the one from 40.5N. Given this, I don't understand why the authors chose to subtract their background estimates from the 40.5N MBL in the bottom panels of Fig 4 instead of using the MBL from 33.4N, which is the latitude of the SCB. Either the authors should explain the rational for this selection or possibly change the latitude they are using in the bottom panels of Fig 4.

Pg 16, Ln 27-28: It looks to me like they are pretty similar in the summer of 2014, but very different from each other in the summer of 2015. Also, I have the same question/comment about using the MBL from 40.5N instead of 33.4N for CH4 as well.

Pg 17, Ln 2-5: Yes, the summertime CH4 from LJO is very different from the other sites! This is really clear in Fig 3, so there should be a reference to that figure in this section. In fig 3 it is clear that LJO has measurements as high as 5ppm which is very high for a background site. There are some very abrupt changes in the LJO data set that may indicate the start or end of a physical process (landfill being covered or not). By having the selection criteria reject observations if there is too much variability in both species they may be making their criteria overly selective at this site.

Pg 17, Ln 7-10: These would be slightly different if the 33.4N latitude was used. Also, it may be described later, but for the % enhancement figures it should be stated that

this is during the afternoon (if that is the case).

Pg 17, Ln 10: After reading this section I went back and looked at Fig 3. It seems to me that the background selection criteria is too strict for VIC and LJO since there are large parts of the year when there are no selected data. This is problematic, for example, at VIC in the summer of 2015 when there are no selected data and the smooth CH4 curve dips below all of the data. The CCGCRV software relies on high and low pass filtering of the data to compute the smoothed curves, but if there is no data it doesn't work! Based on this, I would recommend that the authors slightly relax their selection criteria so that they have at least some data throughout the year. Otherwise they are interpreting a smoothed curve that isn't based on anything. Another approach could be to mask out their background during times when they don't have data to constrain the smoothed curve, but that would defeat their objective of having a background curve all the time. Looking back at Pg 15, Ln 11-21 made me look again at Thoning et al 1998, and they plotted the distribution of their data at Mauna Loa in their Fig 1 showing that most of their data had an hourly variability <0.3 ppm. I wonder what it would look like if the authors examined a similar plot for their data? My guess is that there is greater variability in the data in this work than there was at Mauna Loa (Mauna Loa is a very good background site!) and perhaps a less stringent selection criteria is needed so that they can have more data present throughout the year. After reading further, this issue is discussed on pg 18 ln 1-10, but the authors conclude that VIC is just not suitable as a background station during the summertime due to onshore flow. In that case, the smoothed VIC background curve shouldn't be discussed. However, I still feel that there could be a compelling case to make the selection criteria less stringent than that used for Mauna Loa.

Pg 17, Ln 13: The "LA Basin" should probably be SCB for consistency.

Pg 17, Ln 18: Did you select a specific site in Pasadena? It might be appropriate to list the lat/lon on this location (even though the specifics don't really matter that much). If the back trajectories originate from a site, that would be ideal to note. Also,

what meteorology did they use to drive the HYSPLIT model? The authors state they were using trajectories that ended at 14:00 PST, but were they using monthly averaged wind fields, or were they selecting meteorology from representative days? This should be explicitly described because the decisions about met fields would affect the back trajectories greatly, and hence the interpretation of the seasonal patterns of dominant transport.

Pg 17, Ln 32: The plot showing the back trajectories only spans from 32.5N to 36N, so this seems to also question why the authors looked at MBLs extending all the way up to 40N.

Pg 18, Ln 20: There is an extra parenthesis next to Feng 2016.

Pg 19, Ln 19: The sentence starting "Figure 6 shows..." should explicitly state that the data is averaged for 2015 (as opposed to the whole record). This is stated in the Figure caption, but its ambiguous in the text.

Pg 19, Ln 22-28: These two paragraphs are redundant to Table 4 & 5 and could be removed.

Pg 20, Ln 2-3: The sentence starting "Prior studies..." should have citations with it. Also another factor in the greater wintertime enhancements is the lower PBL heights, as shown in Strong et al. (2011), Figure 4a.

Strong, C., C. Stwertka, D. R. Bowling, B. B. Stephens, and J. R. Ehleringer. "Urban Carbon Dioxide Cycles within the Salt Lake Valley: A Multiple-Box Model Validated by Observations." Journal of Geophysical Research: Atmospheres 116, no. D15 (2011): n/a–n/a. doi:10.1029/2011JD015693.

Pg 20, Ln 11: How were the "outliers" defined? Often outliers are rejected, but it seems that none of these were? It would be good to state this explicitly if the authors use the "outlier" terminology.

Pg 22, Ln 4: There is an extra "and" on this line.

Pg 30, Ln 7-10: Way too many instances of the word "roughly" in this paragraph. Roughly is even duplicated on line 10! The authors could simply remove every instance of "roughly" and it would read better, or they could simply report the specific values and that would be fine also.

Pg 30, Ln 17: "enhancement" is already defined and used in prior paragraphs, so this can just be stated like this "...and how large the signals is relative to the enhancement."

Pg 31, Ln 20 – The sentence starting with "As part of future work..." is a run-on sentence that is a bit confusing, so it should be re-written. Also, I have a small quibble with using the phrase "we plan to..." here and in many other places in the manuscript (eg Pg 19, Ln 3; pg 26 Ln 26, etc). I think that writing "we plan do this or that" lays claim to doing work in the future and discourages others from examining work that needs to be done. Instead, it would be better to say "this or that should be done in the future." This is because investigators change, authors change, plans change, collaborators change, etc. What if another research group comes up with a new method and you collaborate with them to examine fluxes? This could be re-written as: "As part of future work, forward and inverse modelling studies as well as tracer-tracer analysis SHOULD BE USED TO...", or something like that depending on how the run-on sentence is re-worded.

Appendix A1: Pg 56, ln 13-17: The text reads: "Data flags are applied by Earth Networks based on recommendations from the LA Megacity Data Working Group, a team of scientists from NASA's Jet Propulsion Laboratory, 15 Scripps Institution of Oceanography, National Institute of Standards and Technology, and Earth Networks. The manual flags are applied on the EN server to indicate those data that are not recommended for further scientific evaluation or interpretation within the scope of the project." It would be useful for other research groups working on urban GHG measurements to describe what these manual data flags are, or provide a citation for the conclusions of this working group. This is a very important addition.

---

## Author Comment (AC1) · 4 Apr 2017

This short comment addresses a need for clarification on the use of "annual average" in the text. As stated in the text and captions for Tables 2-5 in the manuscript, the annual averages (and standard deviations) were computed for each site based on a single year of data – collected in 2015 – using the one hour observations. We have further clarified the captions for Tables 2-5 to make sure this point is clear. Additionally, we modified the text on P. 11 (line 6) to clearly state that the annual averages are computed based on one hour observations collected during 2015. To prevent any confusion, we now report

only the annual average in this section of the text. This short comment also noted a few other minor issues with regards to grammar and the equations. We thank the reviewer for these comments, which helped improve overall clarity of the manuscript. All changes have been addressed in our revised manuscript.

---

## Author Comment (AC2) · 5 Apr 2017

Response to Referee #1 Comments

1. Overview

We would like to thank the reviewer for insightful comments on the manuscript. Overall, the reviewer's comments on the background analysis and background uncertainty generated some internal discussion and prompted us to make edits that improved the content and clarity of the paper. Below we provide a summary of our responses to each comment and relevant changes made to the manuscript.

2. Minor Comments:

2.1 IRV site

Author response: The modeling study conducted by Feng et al., (2016) focused on modeled CO2 pseudo-data from during May/June 2010. We began collecting observations from the IRV site in Nov. 2014. While it is not possible to do a direct comparison with the results of Feng et al., we do expect the IRV site to experience less trace gas variability due to stronger onshore flow conditions during spring/summer months. The text has been edited to further clarify this point.

MS Changes: We edited the text beginning on P.11 (line 28) as follows: "Feng et al. (2016) used a forward modelling framework to explore variability in modelled CO2 mole fractions during the CalNex period (May-June 2010). Their results, based on modelled CO2 pseudo-data, are generally in agreement with the observations from the SCI and VIC sites. Feng et al. (2016) also showed that the IRV site was relatively clean with respect the modelled pseudo-CO2 data. As shown in Figure 2, during spring/summer months, sites such as IRV and LJO typically show less trace gas variability relative to winter months due to more persistent onshore flow. However, during the rest of the year, the IRV site shows CO2 and CH4 mole fractions in the same range as other suburban sites, such as GRA and FUL (Figure 2, Tables 2 and 3). The LJO site is outside the innermost model domain used Feng et al. (2016) and was not discussed as part of that study. Future work should focus on comparing modelled and observed CO2 and CH4 mole fractions during different meteorological conditions, but using periods with overlapping model and measurement results from the same time period."

2.2 Background values

Author response: The background selection criteria were designed to select observations with a small degree of variability. Overall, the selection criteria are based on the assumption that "clean" background air masses (i.e., those that are not impacted by local sources) exhibit small variability within one hour, with stable conditions persisting

for several hours. A site sampling marine or continental background air is not expected to exhibit significant changes in trace gas variability due to small shifts in meteorology (i.e. wind speed or direction, PBL growth, etc.). By contrast, the same changes in meteorology may lead to increased trace gas variability if the site is influenced by nearby emissions sources. Although not described previously, we have considered possible impacts of PBL growth on the background analysis when preparing this manuscript. Based on this comment (and other reviewer comments) we decided to add a section to the Supplementary materials to provide more details about the background selection criteria. In this new section, we also specifically address the topic of diurnal variability (see below).

MS changes: New text added to the Supplementary materials, P. 4 (beginning on line 29): "We have considered possible impacts of PBL growth on the background analysis. As described in the main text, we use only nighttime flask samples for the MWO background estimate because this site is more sensitive to the LA Basin during daylight hours due to growth of the PBL and upslope winds. However, our filtering criteria for SCI, LJO, and VIC do not account for diurnal variations, e.g., due to variations in the planetary boundary layer height or due to potential daytime drawdown of $CO_2$ due to photosynthetic uptake. Initially, we made plots of the monthly average diurnal variability for the SCI, LJO, and VIC sites. However, it was not apparent how the diurnal cycle would aid in the interpretation of background because most of the time the diurnal changes at these sites are dominated by impacts from local emissions (especially at LJO and to a lesser extent at the other two background sites due to outflow). At the marine background sites (LJO and SCI), it is the growth the marine boundary layer (MBL) rather than the PBL over the land, that is relevant to the interpretation of background. However, the MBL growth effect is most relevant when a site located very far off-shore, such that nighttime continental outflow is not present. Under these conditions, changes in the MBL with time of day are likely to be very small. The LJO is near sea level and is within the MBL, but is frequently impacted by local sources. The SCI site can be either within or above the MBL due to its elevation ($\sim$489 m asl), but is still occasionally

impacted by continental outflow. For these reasons, we do not limit the background consideration to certain times of day. The agreement between the SCI and LJO marine background estimates (within ∼±1 ppm CO2 and ∼±10 ppb CH4) suggests that there is not a large gradient between the CO2 and CH4 levels in the surface MBL and above the MBL. In summary, for the SCI, LJO, and VIC background sites, our underlying assumption is that if the PBL (or MBL) grows, it will not further dilute the CO2 or CH4 levels or cause additional large variations if the site is truly sampling background conditions. . . . Overall, we have achieved a reasonable level of convergence between the background estimates for three sites with very different variability in CO2 and CH4 mole fractions. A metric of success exhibited by our results is that the background reference curve estimates agree within ∼±1 ppm CO2 and ∼±10 ppb CH4 for the marine sites (LJO and SCI) and continental sites (MWO and VIC, see Figures 4 and S10)."

2.3 Background uncertainty:

Author response: We have significantly revised the method to quantify a time-dependent background uncertainty estimate for each site based on this comment and our internal review. As shown in Figure 3 in the manuscript, the blue curves are the background estimates and the red points are used to calculate the blue curve. During periods when the curve is constrained by observations, we assign the uncertainty based on the monthly average residual between the observations and the smooth curve. Due to filtering, there were some large gaps in the observations (red points) that were used to calculate the background curves. The CCGCRV curve fitting algorithm interpolates over these gaps, resulting in an additional interpolation uncertainty. We now assign a larger uncertainty during data gaps such that the uncertainty will default to the maximum annual average residual during periods with gaps. Finally, it is worth mentioning that based on a comment from Reviewer #3, we revised our filtering criteria such that the largest gaps are now ∼1 month (see our response to Reviewer #3's comments). Overall, we feel these changes have significantly improved the discussion of background and uncertainty.

MS changes: We changed the text in Section 6.2, P. 28 (lines 1-13) as follows (note "…" was inserted where equations are given: "We define the time-varying uncertainty in the background estimate as …(Eq. 13), where … is the absolute value of the monthly average residual of the selected background observations (red points, Figure 3) from the smooth curve result. Due to the method used to filter the observations, there are some gaps in the background observations. The background reference curves interpolate over observation gaps, however, the portions of the curve that are not constrained by observations are more uncertain relative to other periods. For data gaps longer than one month, it is not possible to estimate … . Since there are no observations to constrain the curve, we assign an interpolation uncertainty based on the maximum annual average residual. In other words, if there are long observation gaps, the interpolation uncertainty will default to the maximum residual based on periods when observations were available. The time-varying uncertainty estimates for the SCI, VIC, and LJO reference curves are shown in Figure S10. During 2015, the annual average uncertainty in the SCI smooth curve estimate is 1.4 ppm $CO_2$ and 11.9 ppb $CH_4$ (Table 6). This amounts to roughly 10% and 15% of the median midday enhancement near Downtown LA (i.e., at the USC site) for $CO_2$ and $CH_4$, respectively."

We also added Figure S10 to the Supplementary materials, which shows the background and time-dependent uncertainty estimates for SCI, VIC, and LJO. All the other text related to background uncertainty was also updated, including the annual average U_BG estimate from SCI in the abstract and in Table 6. Figure S10 will be attached to this review as a separate PDF file.

3. Specific comments

Page 2, Line 25: The reference was added to the Introduction.

Page 11, Line 30: A shallower nighttime PBL will lead to higher $CO_2$ and $CH_4$ enhancements and higher sensitivity to local surface emissions. We edited the text (now on P.12, lines 11-13) as follows: "Many of the $CH_4$ spikes throughout the GRA record

occur at night, suggesting contributions from a nearby source. Shallower PBL heights at night will lead to higher trace gas enhancements and higher sensitivity to local surface emissions (e.g., Djuricin et al., 2010; Turnbull et al., 2015)."

Page 18, Line 10: Yes, the converse is also true that SCI and LJO would not be relevant choices for background when flow is from the continent. We modified the text on P. 18 (now on lines 1-4) as follows: "Overall, the VIC and MWO sites may not be relevant choices for background during summer, when onshore flow patterns dominate. Conversely, SCI and LJO may not be relevant choices for background when flow is from the continent. In future studies, background data could also be selected based on the prevailing flow patterns in the region of interest (e.g., McKain et al., 2015)."
* * *
[Figure]

**Fig. 1.** Fig S10. Background observations (black circles), smooth curve estimates (cyan lines) and uncertainty estimates (red dashed lines) for San Clemente Island (a-b), La Jolla (c-d), and Victorville (e-f).

---

## Author Comment (AC3) · 5 Apr 2017

Response to Referee #2 Comments

1. Overview

We would like to thank the reviewer for their valuable comments, many of which we incorporated into a revised version of the manuscript. Below we provide responses to the specific comments and a summary of changes to the manuscript, where applicable.

2. Technical/minor comments

[Figure]

Abstract:

Author Response (Comment 1): The extra "to" was removed.

Author Response (Comment 2): We have clarified the sentence as follows: "The USC site near Downtown LA exhibits median hourly enhancements of ~20 ppm $CO_2$ and ~150 ppb $CH_4$ during 2015, and ~15 ppm $CO_2$ and ~80 ppb $CH_4$ during midday hours (12-16:00 LT, local time), which is the typical period of focus for flux inversions."

Author Response (Comment 3): We reworded the sentence as follows: "The largest component of the measurement uncertainty is due to the single-point calibration method; however, the uncertainty in the background mole fraction is much larger than the measurement uncertainty."

Section 2.3:

Author Response (Comment 1): The G2201-i and G1101-i analyzers are not mentioned in the main text because results from these analyzers are not included as part of this study. We list them in Table 1 because they are relevant as prior measurement sites. The footnotes in Table 1 were reorganized and reworded throughout for clarity. No additional changes were made to the text.

Author Response (Comment 2): Correct. As we discuss in the text, it was not possible to calibrate all the instruments in a laboratory prior to field deployment. As the reviewer comment notes, this is not ideal, which is why we dedicate a significant part of our analysis to investigating the error in single-point calibration method and the slope (epsilon) estimates for various Picarro analyzers with similar model numbers. As stated in the text on page 22, lines 25-26: "Our approach relies on independent estimates of $\varepsilon$ (epsilon), the slope parameter, to determine the magnitude of the systematic and random components of the (extrapolation) error in our calibration method." For more details, see Section 6.1.1 Extrapolation uncertainty, and results in Tables S2 and S3 and Figures S5 and S6). No additional changes were made to the text.

[Figure]

Section 2.4:

Author Response (Comment 1): The repeated sentence was removed.

Author Response (Comment 2): Yes, all tanks used in the LA network are calibrated to the same NOAA/WMO scales (X2007 for CO2 and X2004A for CH4). The text on page 9, lines 17-19 was edited as follows: "The SIO standards are filled using a similar procedure, except tanks are filled with natural coastal air from Scripps Pier in La Jolla, California, and the tanks are also calibrated against standards on the same WMO-scales."

Author Response (Comment 3): The missing period in the second paragraph was added.

Author Response (Comment 4): In response to this comment, we moved up the first reference to the high mole fraction tanks to the first paragraph in Section 2.4: "In addition to the ambient-level calibration and target tanks, the VIC and LJO sites had high mole-fraction standard tanks installed at the time of this study. These tanks were prepared by NOAA/ESRL and calibration assignments were provided prior to deployment (roughly 500 ppm CO2 and 2600 ppb CH4)."

Section 3.1:

Author Response: This sentence was also pointed out by another reviewer. We decided that the content was not critical to this paper, we removed the first sentence of Section 3.1.

Tables 2 and 3:

Author Response: Tables 2 and 3 provide the statistics of the CO2 and CH4 measurements. We feel this quantitative information would be lost in figures and is more appropriately presented in table format. No changes were made to the manuscript.

Section 4.1:

Author Response: For the purposes of this study, there was an effort to ascertain criteria for selecting background both CO2 and CH4 observations. The CO2 background selection criteria for S.D. within 1 hour (0.3 ppm CO2) is based selection criteria used by Thoning et al., (1989, JGR) to select background CO2 observations at Mauna Loa, HI. There are no similar criteria published for CH4, so we came up with our own based on the observed variability at our sites. In general, CH4 exhibits more hour-to-hour variability (relative to its baseline) compared to CO2. Therefore, different filtering parameters were needed to limit the variability while not excluding too much data. Since we are using an hour-to-hour criteria for CO2, the hour-to-hour variability is already somewhat restricted, so imposing an additional hour-to-hour criteria for CH4 did not seem appropriate. In response to this comment – and another similar comment from reviewer #3 – we performed additional tests on the data filter criteria used for each site. To summarize those changes, we included a new figure in the supplement (Figure S4, see below), showing histograms of the 1 hour S.D. for CO2 and CH4 observations from SCI, VIC, and LJO. Overall, Figure S4 shows that a large fraction of measurements from all three sites have hourly standard deviations <0.3 ppm CO2 and <3 ppm CH4. During 2015, 70%, 42%, and 30% of the data had a 1 hour S.D. <0.3 ppm CO2, 67%, 57%, and 42% of the data had a 1 hour S.D <3 ppb CH4, and 60%, 35%, and 29% of the data met both criteria for the SCI, VIC, and LJO sites, respectively. Because a significant fraction of the data from each site is within these limits (∼30% or more), we applied them to all three sites. This means our CH4 filter limit was reduced from 5 ppb to 3 ppb CH4 (which is more reasonable considering the average analytical uncertainty in the CH4 observations was only ∼1 ppb at LJO during 2015, see Table 6 in the revised manuscript). For SCI, all other original filter criteria were retained (one hour S.D. of 0.3 ppm CO2, hour-to-hour stability cutoff of 0.25 ppm CO2 based on Thoning et al., 1989, and 6 hours of persistent "background" conditions). Overall, we found that the LJO and VIC were most sensitive to filter criteria 2 and 3 (the hour-to-hour stability and number of consecutive hours with stable conditions). We performed tests varying the hour-to-hour stability between 0.25 and 0.5 ppm CO2 and the number of consecutive

hours from 3 to 6 hours and analyzed the results. For LJO, the original filter criteria did not produce large gaps (i.e., >1 month). Furthermore, increasing the allowable hour-to-hour stability or decreasing the number of consecutive hours sometimes resulted in a few anomalously high $CO_2$ and/or $CH_4$ observations being included in the result, which was unfavorable (and likely due to a persistent polluted air mass passing over the site rather than clean background air). For these reasons, we decided to keep the same filter criteria for both LJO and SCI. For VIC, we noticed that applying the same filtering criteria as SCI and LJO produced large gaps in the selected background observations, sometimes over an entire season in summer months, which would make the background estimate highly uncertain. In order to reduce gaps in the VIC background observations to <=1 month, we used the following criteria: 1) hour-to-hour stability equal to 0.5 ppm $CO_2$ and 2) number of consecutive hours with stable conditions equal to four hours. With these changes, there are no longer significant gaps in the $CO_2$ or $CH_4$ records used to generate the smooth curve fits. Overall, we believe the final results from our revised background analysis are very reasonable for the intended purposes. Furthermore, the agreement between the background estimates from the marine and continental sites –which all exhibited very different variability in $CO_2$ and $CH_4$ mole fractions – serves as a metric of success in our approach.

Summary of MS changes for this comment:

P. 15 (Section 4.1, line 17-28): "LJO and SCI "Marine" Background and VIC "Continental" Background Estimates: The LJO, SCI, and VIC air observations were filtered according to statistical criteria based on the variability in the hourly average data (see Supplementary materials). As shown in Figure 3, the $CO_2$ and $CH_4$ observations from SCI exhibit much less variability compared to VIC and LJO. Figure S4 shows histograms of the hourly standard deviations for the SCI, VIC, and LJO observations. As discussed earlier, the variability in the LJO record is more like an urban/suburban site than a background site. This is primarily due to along-shore transport from the north and the proximity to other local sources (including a large landfill immediately to the

east). After applying the selection criteria respective to each site, the CCGCRV curve fitting software was used to estimate a "smooth curve" fit to the selected observations (Thoning et al., 1989; http://www.esrl.noaa.gov/gmd/ccgg/mbl/crvfit/crvfit.html). The curve-fitting parameters are described further in the Supplementary materials. The full time series, selected data and "smooth curve" results are shown in Figure 3 and the final smooth curve results for each site are shown in Figure 4 (panels a-b). We discuss the uncertainty in the smooth curve estimates in Section 6.2 (see also, Figure S10)."

Supplementary materials (Section 3, P. 4-5): Added details described above regarding the filtering criteria for each site. Also, added Figure S4 showing histograms of the 1 hour S.D. for the $CO_2$ and $CH_4$ observations from SCI, LJO and VIC.

We also removed the following sentences as the gaps at VIC are no longer relevant to the discussion in this paper:

Abstract: "We also show that continental sites may not be relevant for selecting background observations during summer months due to the prevalence of onshore flow, which could transport $CO_2$ and $CH_4$ from the LA Basin to relatively remote sites."

P. 18, Section 4.3: "For VIC, there is virtually no $CO_2$ or $CH_4$ data meeting the selection criteria during the summer and early fall months."

Section 4.3:

Author Response: We used the SCI background estimate to calculate the $CO_2$ and $CH_4$ enhancements because this turns out to be the site that samples marine background air most frequently near LA, and additionally, the results from SCI look plausible compared to other marine background estimates. While this may seem somewhat obvious after reading the manuscript, the analysis we conducted really served to explore whether SCI is a reasonable marine background site, which had not been demonstrated previously in the literature. In our analysis, we demonstrate that SCI is a good background site by analyzing results from SCI in comparison to the LJO and Pacific

[Figure]

MBL background curves, as well as two other potential continental background sites. Historically, LJO has been used as a background site for other gases; however, the continuous observations show LJO is frequently impacted by local sources of $CO_2$ and $CH_4$ (see Figure 2 in the manuscript). Applying our filtering criteria to both SCI and LJO offered a great test case for our methodology. A metric of success exhibited by our results is that we have achieved a reasonable level of convergence between all the background estimates (e.g., marine sites: LJO vs SCI and continental sites: MWO vs VIC), and overall the differences between the background estimates are now relatively small (see Figure 4 in the manuscript). To further address this comment, we changed the text on P. 19 (lines 10-13) in the manuscript, as follows: "SCI is the most representative of local marine background conditions for both $CO_2$ and $CH_4$ throughout the year. The LJO background curve also helps confirm that the background estimate from SCI is reasonable. Therefore, we use SCI as the background reference site to calculate $CO_2$ and $CH_4$ enhancements for the LA surface sites (see Section 5)."

Section 6.1.6:

Author Response (Comment 1): We removed the redundant text in the second paragraph.

Author Response (Comment 2): This comment refers to the fact that standard tanks that are measured during "pre-deployment" and "post-deployment" field checks. This is determined independently at the NOAA/ESRL laboratories, and these results do not indicate significant drift (see, e.g., Andrews et al., 2014). For clarification, we changed the text (now on P. 27, lines 22-24) as follows: "Andrews et al. (2014) report a mean difference between pre- and post-deployment tank calibrations of $CO_2$ and $CH_4$ for tanks prepared by the NOAA/ESRL laboratories."

Tables 5 and 6

Author Response: We assume the reviewer was referring to moving Tables 6 and 7 to the Supplement (the statistics for epsilon, or the slope estimates from individual

analyzers) since Tables 5 and 6 show very different results from one another. We moved Tables 6 and 7 to the Supplement (now Tables S2 and S3). As noted, these tables complement Figures S4 and S5.
* * *
[Figure]

[Figure]

**Fig. 1.** Fig S4: Histograms of the S.D. of hourly CO2 (panels a and b) and CH4 (panels c and d) observations from SCI (blue), VIC (red), and LJO (grey). Left panels show all data (right panels, zoomed).

---

## Author Comment (AC4) · 6 Apr 2017

Response to Anonymous Referee #3 Comments

1. General Comments:

The authors would like to thank anonymous referee #3 for a thorough review and many insightful comments that improved the manuscript. In response to one of the broader general comments, we agree it does not make sense to have both an Appendix and Supplement. We decided to transfer all Appendix materials into the Supplement, with changes made throughout the text to reflect updated figure and table numbers. An-

other general comment related to our background estimation method. As discussed in our responses to the previous two reviews, we have performed additional tests related to this topic. In our response to Reviewer #2, we include a new supplementary figure (Figure S4), showing histograms of the 1 hour S.D. for the $CO_2$ and $CH_4$ observations from the three background sites (SCI, LJO and VIC). We also made some minor changes to the background selection criteria for the three LA regional background sites (also discussed here), which we believe significantly improved the quality of the background estimates. All details regarding the background data selection methods are now included in the Supplementary materials, Section 3. Below we discuss these and other specific comments in more detail. Our responses are arranged by section of the manuscript. The reviewer also noted a number of very helpful grammatical and stylistic edits that we have addressed. These are summarized at the end of our response.

2. Specific Comments:

2.1 Abstract and Introduction

Pg 2, Ln 13: We have modified the text in the Introduction to include a discussion of using both top-down and bottom-up methods together to evaluate emissions (see P.2, 2nd paragraph). We also added several references (including: Lauvuax et al., 2016; Asefi-Najafabady et al., 2014; and Gurney et al., 2005).

Pg 2, Ln 30-Pg 3 Ln 3: The sentence noted was removed.

Pg 3, Ln 7: The Los Angeles project was described as a "pilot" because this is one of the first projects of its kind. For clarity, we changed the word "pilot" to "testbed".

Pg 3, Ln 14: We decided to use an estimate of 16.3 million residents for the SCB region (roughly 17,100 km2) based on the following reference: The California Almanac of Emissions and Air Quality - 2013 EditionÂăRep., California Air Resources Board, Sacramento, CA (CARB, 2014). Ultimately, the population estimate should take into account the land area. For example, in 2015, the "Combined Statistical Area" for the

greater Los Angeles area was 18.7 million. Combined statistical area for the greater Los Angeles area is defined as the sum of population in the five SCB counties, and including the non-urban desert areas of Riverside and San Bernardino Counties. A quick check suggests that there's roughly 1 million people in the desert areas of Riverside and San Bernardino (Victor Valley (Victorville + Apple Valley + Hesperia): 400k, Coachella Valley: 400k, Imperial Valley: 200k). Therefore, the estimate we are using is slightly smaller than the "Combined Statistical Area" because it does not include the non-urban desert portions of the Los Angeles, Riverside and San Bernardino Counties in the SCB.

Pg 3, Ln 22: We agree. The redundant sentence was removed, with some other minor stylistic edits to the text on P. 3 lines 29-31 for clarity.

Pg 4, Ln 1-5: This comment relates to the ethanol content in gasoline used in California, which the ARB reports has been approximately 10% by volume since the beginning of 2010. Ethanol derived from C4 grasses (i.e. corn) increases the ratio of 13C/12C from combustion of gasoline, which adds complexity to fossil $CO_2$ attribution. Newman et al., (2016) discuss this topic in great detail. We changed the text in the Introduction, P. 5 ($\sim$lines 12-14) as follows: "In California, gasoline is approximately 10% ethanol by volume. Ethanol that is derived from biofuel (i.e. from C4 grasses, such as corn) will increase the ratio of atmospheric 13C/12C when gasoline is combusted, adding complexity to the attribution of fossil $CO_2$ emissions (Djuricin et al., 2010; Newman et al., 2016)."

Pg 5, Ln 1: We updated the text to reflect these changes. On P. 5, we define the first instance of the term "enhancement."

Pg 5, Ln 2-3: The word "robust" was changed to "large" and "roughly" was removed in both instances. We kept "10s to 1000's of ppb CH4" because this still accurately reflects the range of CH4 mole fraction reported from multiple studies in the LA Basin.

2.2 Methods Section

[Figure]

Pg 7, Ln 23: In our system, the temperature of the Earth Networks sample module is controlled at 38 degrees C at all sites. To address the second part of this comment, the Earth Networks sample module has a heater which is strong enough to raise the temperature in the box by up to about 30C. Therefore, the optimal range of control for the heater is when the ambient (room) temperature is between 8C to 38C. The GCWerks software controlling the sample module was modified to implement a PID algorithm to keep the temperature control to within about one tenth of a degree under normal operating conditions. In GCWerks, we also monitor the variables "sample temperature," which is the readout of the sample module temperature (after control), and "ambient temperature, which is the temperature immediately outside the sample module. These variables can be used to look at the stability of the sample module and room temperature over time. As the reviewer inferred, we do not actively cool the sample module and only apply heating. Therefore, we rely on the ambient air temperature just outside of the sample module to be within the 8C to 38C range so that the only primary cooling mechanism required is the loss of heat from the sample module to the room. While this could introduce potential problems during an extreme summer heatwave, the majority of the instruments are located in temperature controlled shelters so the ambient room temperature is nearly always between 8C and 38C. To address this concern, we included the temperature at which the heated box is maintained (38C) in the text on P. 7 (line 30).

Pg 9, Ln 19-20: The high mole fraction tanks were purchased from NOAA/ESRL. The tank filling procedure for all tanks (near ambient and high mole fraction) is now described in the first paragraph in Section 2.4. For reference, the NOAA/ESRL tank filling procedures are described in much more detail here: https://www.esrl.noaa.gov/gmd/outreach/behind_the_scenes/standards.html. The text on P. 9 (lines 12-18) was changed as follows: "In addition to the ambient-level calibration and target tanks, the VIC and LJO sites had high mole-fraction standard tanks installed at the time of this study. These high mole fraction tanks were prepared by NOAA/ESRL and calibration assignments were provided prior to deployment (roughly

500 ppm CO2 and 2600 ppb CH4). The NOAA/ESRL high mole fraction tanks are prepared by adding a 10% CO2-in-air mixture to natural air during the pressurization of the cylinder at Niwot Ridge, Colorado (and a similar procedure is used for CH4). The cylinder is then moved to the NOAA calibration laboratory in Boulder, CO where it is calibrated relative to NOAA/WMO secondary standards."

Pg 9, Ln 20-23: Yes, we reject the first 10-minutes to account for the stabilization period after the inlet is switched, i.e. to account for the turnover of air in the CRDS coming to equilibrium (see also Welp et al. 2013). This was mentioned in the supplement, however, we also modified the main text for clarity (now on P. 10, lines 2-4): "The first 10 minutes of each tank run are rejected and only the data from the last 10 minutes of any are used in the calibration of CO2 and CH4 mole fractions to account for the stabilization of air in the CRDS after the inlet is switched (Welp et al., 2013)."

Pg 9, Ln 14: Yes, we are evaluating both accuracy and stability, where stability is the accuracy over time. We added a sentence on P.9 (line 21), as follows: "This calibration method assumes a linear response in the analyzer."

2.3 Results Section

Pg 10, Ln 23-24: We decided to remove the first sentence of Section 3.1 and made some minor revisions to the text of the paragraph that follows to begin this section. The next sentence was revised as follows: "Figure 2 shows the 1 hour average observations collected from nine sites in the Los Angeles surface network between January 1, 2013 and December 31, 2015 and Tables 2 and 3 show the annual average CO2 and CH4 variability based on hourly observations collected during 2015."

Pg 11, ln 1: We included the word "hourly" to clarify that the S.D. is defined using the hourly data. We also modified the Table 2 and 3 captions so it is clear that we using hourly data to compute the annual statistics.

Pg 11, Ln 13-14: Yes, the VIC and SCI sites exhibit the lowest variability with regards

to CO2 and CH4 mole fractions. The text was modified slightly to clarify this point. We changed the text in the second paragraph of section 4.1 as follows: "Victorville and San Clemente Island (VIC and SCI) show less variability in CO2 and CH4 mole fractions compared to the other sites within the SCB (Figure 2)."

Pg 12, Ln 14-15: Yes, the assumption was implied. We modified the text so this is now explicitly stated on P.12 (now on lines 18-20) as follows: "Given a constant flux, and assuming that transport in and out of the boundary layer remains approximately constant, the trace gas mole fraction observed within the PBL will increase or decrease as the PBL height falls or rises, respectively."

Pg 12, Ln 17-18: We agree, the PBL height, rather than stability, is the more important factor controlling the magnitude of the enhancements. A similar comment was made by reviewer #1 in the same section of the text, so we have modified the text on P. 12 to address both comments, see lines 14-15: "Shallower PBL heights at night will lead to higher trace gas enhancements and higher sensitivity to local surface emissions (e.g., Djuricin et al., 2010; Turnbull et al., 2015)."...and lines 23-24: "The reduced variability in the CO2 and CH4 observations during midday hours is in part due to the larger height of the PBL during the mid/late afternoon."

Pg 13, Ln 10: A consensus on the notation for CO2 and CH4 enhancements is indeed a good question for the broader community and perhaps something that should be discussed as part of upcoming workshops that include urban measurements. The terms enhancement and excess are interchangeable and we agree with the reviewer that the capital "delta" notation could be confusing to some readers. We decided to adopt the notation "CO2xs" in alignment with the work of Newman et al. (2013; 2016). All instances of "delta" notation were removed and were changed to CO2xs or CH4xs (including figures, tables, and captions).

Pg 13, Ln 18: This is a minor technical comment. We decided to leave the units in for clarity. Sometimes readers may not start with the beginning of a paper, so we feel

it is important to note that we are using the mole fraction and indicate the units used throughout the paper in this section.

Pg 13, Ln 19: This is a stylistic point. We have decided to leave the title of Section 4 as is as it describes our process for evaluating background mole fractions for the Los Angeles region (as "estimation" is the process of approximately calculating or evaluating something.

Pg 13, Ln 22-25 and Pg 14, Ln 15-16: We also decided to leave in the site details in these locations for clarity. We also added the site codes for the LJO and PVP sites.

Pg 15, Ln 23: Correct, MWO is a mountaintop site and "m asl" is the correct notation!

Pg 16, Ln 13-14; Pg 16, Ln 25-26; Pg 16, Ln 27-28; and Pg 17, Ln 7-10: Here we respond to four comments all related to Figure 4, which show the MBL curves and a comparison of the background estimates. In Figure 4 we show 3 MBL curves, one near the latitude of SCB and the other two from latitudes north of the Basin. Generally, the climatological flow into Southern California during onshore flow conditions is from the north rather than the south, before reaching the California Bight and flowing inland. This is illustrated in the HYSPLIT back trajectories shown in Figure 5, and is the primary reason we chose to show MBL estimates for two latitudes north of the SCB region. Aside from this, the exact latitudes for the MBL curves are linked to model resolution. Another related comment in this set refers to the lower panels of Figure 4, where we show the difference of each smooth curve estimate from a "control" case. In the manuscript, this difference was computed from the MBL curve at 40.5 N. Overall, the differences in the $CO_2$ and $CH_4$ mole fractions between the three MBL curves at 40.5, 36.9 and 33.4 degrees N are relatively small (see Figure 4, upper panels). Therefore, the choice of subtracting each curve from the 40.5N latitude was somewhat arbitrary. Per the reviewer's suggestion, we decided to subtract each curve from the MBL estimate at 33.4 N to update the analysis. In response to the last comment in this set, the estimates for the background uncertainty were computed as the "percentage

of the enhancement" for mid-afternoon hours only.

Summary of MS changes related to these comments: We updated Figure 4 and modified the text in Section 4.2 (P. 16-17) to indicate that we subtracted each background estimate from the MBL curve at 33.4 N: "The average absolute difference between the Pacific MBL estimate at 33.4° N and each background estimate from SCI, LJO, VIC, and MWO for the period shown in Figure 4 is: 0.8, 0.7, 1.7 and 1.5 ppm CO2, and 8.0, 8.9, 10.1, and 13.7 ppb CH4, respectively. The average absolute differences between the background estimates from SCI and LJO and the Pacific MBL estimate from 33.4° N are <1 ppm CO2 and <10 ppb CH4, suggesting that both sites are useful for deriving marine background estimates for CO2 and CH4 when the appropriate filtering criteria are used." We also updated the text in Section 6.2 (P. 28, ∼lines 16-18): "During 2015, the annual average uncertainty in the SCI smooth curve estimate is 1.4 ppm CO2 and 11.9 ppb CH4 (Table 6). This amounts to roughly 10% and 15% of the median mid-afternoon enhancement in Downtown LA (USC) for CO2 and CH4, respectively." And in the Abstract (P. 2, ∼lines 3-4): The background uncertainty for the marine background estimate is ∼10% and ∼15% of the mid-afternoon enhancement near Downtown LA for CO2 and CH4, respectively.

Pg 17, Ln 2-5: LJO is a coastal site with essentially no local upwind sources over south-westerly sector. Generally, the CO2 and CH4 observations from LJO show significantly more variability than the San Clemente Island and Victorville sites. As discussed in the manuscript, the variability at LJO is more like an urban/suburban site than a background site. This is primarily because the site has very strong sources in other wind sectors (and the reason we see measurements as high as 5ppm, which is indeed very high for a background site). The proximity of the LJO site to local sources (including a large landfill immediately to the east and along-shore transport from the north) as well as the meteorology (which sometimes brings very clean air to the site) explains why this site is both heavily impacted by local emissions but is sometimes useful as a background site. Regarding the variability in both species, CO2 and CH4 can be coemitted from urban emissions sources, such as landfills, gas fired power plants, etc. Therefore, both CO2 and CH4 levels (as well as CO and other trace gases) may be impacted when the site is influenced by an urban air mass. By requiring small variability in both CO2 and CH4 levels as part of our filtering criteria, we are assuming that either gas could indicate influence from an urban air mass. While we did not find that the selection criterion based on both CO2 and CH4 variability was too strict, we did make small modifications to the data selection criteria as described in our response to comments from Reviewer #2 as well as the next comment. For further changes related to the background topic, please see our response to the comment below as well as our Response to Comments from Referee #2.

Pg 17, Ln 10: We agree, Mauna Loa is a very good background site and our criteria for the LA background sites need not be identical to this site. Based on reviewer comments and our internal review, we performed several additional tests with the background selection criteria. In our response to Reviewer #2, we included a new figure in the Supplement (Figure S4) with histograms of the 1 hour S.D. for the CO2 and CH4 observations from SCI, LJO and VIC, similar to the analysis of Thoning et al.. During 2015, 70%, 42%, and 30% of the data had a 1 hour S.D. <0.3 ppm CO2, 67%, 57%, and 42% of the data had a 1 hour S.D <3 ppb CH4 filter criteria, and 60%, 35%, and 29% of the data met both criteria for the SCI, VIC, and LJO sites, respectively. Based on this analysis, we reduced our CH4 one hour S.D. filter limit from 5 ppb to 3 ppb CH4 and retained the original CO2 filter criteria (0.3 ppm CO2). Next, we performed tests by varying the number of consecutive hours with stable conditions from 3 to 6 hours and the cutoff for the hour-to-hour variability between 0.25 to 0.5 ppm CO2. We found that the LJO and VIC observations were most sensitive to the filter parameters, especially criteria 2 and 3, the hour-to-hour stability and number of consecutive hours. For VIC, our results are in agreement with Reviewer #3's inference that the limits were too strict. We found that requiring 6 or more consecutive hours of stable conditions at VIC resulted in large data gaps over the entire season during summer months, making the background estimate highly uncertain during this period. We also found that

gaps in the VIC background observations were reduced to <1 month when the following criteria were used: 1) hour-to-hour stability equal to 0.5 ppm CO2 and 2) number of consecutive hours with stable conditions equal to four hours. For LJO, the original filter criteria did not produce gaps >1 month during 2015-2016. Furthermore, using less strict criteria for LJO resulted in a few anomalously high CO2 and/or CH4 observations being included in the result, which is unfavorable (and likely due to a persistent polluted air mass passing over the site rather than clean background air). For these reasons, we decided to keep the same filter criteria for both LJO and SCI. With these changes, there are no longer significant gaps in the records used to generate the smooth curve fits. Overall, we believe the results presented are now very reasonable, as exhibited by the improved agreement between the background estimates. Finally, we note that while the filtering criteria are now less strict, there are still differences between the marine (i.e. LJO and SCI) and continental (i.e. MWO and VIC) background estimates that cannot be explained by the data filtering methods.

Summary of MS changes for this comment: Supplementary materials: See Figure S4 showing histograms of the 1 hour S.D. for the CO2 and CH4 observations from SCI, LJO and VIC. Also, see new text detailing the filtering criteria in Section 3 (P. 4-5). In Section 4.1 of the main text, we now refer to details about the filtering criteria in the Supplement. In Section 4.2, P. 17 (∼lines 3-6), we modified the text as follows: "The cause of the larger differences between the continental (i.e., VIC and MWO) and marine (i.e.. SCI, LJO, and Pacific MBL) background estimates is not clear. Future modelling studies could investigate whether a time-dependent background selection method – e.g., based meteorological information and the origin of incoming air mass – can be used to determine the appropriate background site under some of the more common meteorological regimes in the SCB." We also removed the following sentence that was no longer relevant in Section 4.3, P. 18: "For VIC, there is virtually no CO2 or CH4 data meeting the selection criteria during the summer and early fall months (Figure 3)."

Pg 17, Ln 18: Yes, the back trajectories are for the Caltech site in Pasadena, CA using NAM12 winds. We modified the text in Section 4.3, P. 17 (line 17-18) as follows: "We computed twenty-four hour back trajectories for winds arriving at the CIT site in Pasadena at 14:00 LST using NOAA's HYSPLIT model (Figure 5; Stein et al., 2015; Rolph, 2016)." We also added more details to the Figure 5 caption, including the site coordinates, the model (HYSPLIT) and that we used hourly winds from NAM12.

Pg 17, Ln 32: As noted in our response to an earlier comment, all the background curves in Figure 4 are now subtracted from the MBL estimate at 33.4 degrees N. Figure 5 is mainly intended to show the general seasonal trends in the location of the incoming air masses to the SCB region. We mainly showed the 32.5-36 N latitude range because expanding the scale did not add much additional information and the zoomed-out view made visualizing the back trajectories more difficult. However, we note that this analysis has also been performed for other sites in the SCB and shows very similar results.

Pg 19, Ln 19: We modified the text to clarify that observations shown in Figure 6 are averaged for 2015, as opposed to the whole record.

Pg 19, Ln 22-28: We removed some of the redundant text and merged the remaining text from the first and second paragraphs of Section 5, which significantly improved the overall flow of this section.

Pg 20, Ln 2-3: The main issue noted relates to seasonal variations in mixing layer height. We prefer to cite the Ware et al., (2016) rather than Strong et al., (2011) for relevant information on mixing heights in Los Angeles and because winter conditions may impact mixing heights differently in LA and Salt Lake City. Recently, Ware et al., (2016) used backscatter data from a Mini Micropulse Lidar (MiniMPL) instrument located at Caltech in Pasadena, CA and provided a detailed assessment of mixing height observations from 2012-2014 near one of our rooftop sites. Their results suggest relationship between trace gas enhancements and mixing height in LA may be more complicated

than the reviewer's comment suggests. We added text to the manuscript on P. 20 (lines 11-21), as follows: "In general, increased summertime insolation is expected to produce a deeper afternoon mixed layer depth in summer relative to winter, which in turn would result in larger trace gas enhancements within the PBL during winter relative to summer. As discussed earlier, Ware et al. (2016) used backscatter data from a Mini-iMPL instrument located in Pasadena, CA to estimate mixing heights over two years from 2012-2014. They found the mean afternoon maximum mixing depth was 770 m agl in summer (June and August) and 670 m agl in winter (December–February). However, seasonal differences in mixing depth should also be considered in the context of the daily and weekly variability. Ware et al. (2016) show that the maximum depth of the afternoon mixing layer may differ by a factor of 2 from day-to-day. Furthermore, they also show that the within-season S.D. for the afternoon maximum mixing height is about 220 m, or approximately 30% of the mean afternoon maximum mixing depth in either summer or winter (which is larger than the observed average seasonal differences in mixing height). Overall, the large variability in mixing layer depth over different timescales suggests that the meteorological impacts on trace gas concentrations in the PBL can also be quite variable."

Pg 20, Ln 11: The word "outlier" was used incorrectly here and in the figure caption. The red pluses in Figure S11 indicate enhancements greater (or less than) the maximum whisker length, showing the full range of variability.

2.4 Summary and Conclusions Section

Pg 30, Ln 17: We are interested in the magnitude of the enhancement, which is the signal above background, not the signal relative to the enhancement. A minor modification was made to the text to make sure this point is clear

Pg 31, Ln 20: The run-on sentence was rewritten and the instances of the phrase "we plan to" were also revised as suggested.

2.5 Appendix/Supplement

Appendix A1: Pg 56, ln 13-17: Please note, all appendix materials are now included in the supplement, so the text noted here is in now located in Section 1.2 of the Supplementary materials. In general, manual flags are applied on a case-by-case basis. Mainly, the manually flagged data are identified during technician site visits, especially those that required modifications to the plumbing on the instrument of callbox (e.g. when calibration standards are replaced, when an analyzer is removed for repair, etc.). The text was modified to describe some specific instances when manual flags have been applied.

3. Typographical edits

All suggested typographical edits listed below were addressed in the revised manuscript:

Abstract Ln 30: The time span should be explicitly stated, not "roughly". This can be fixed by removing the word "roughly".

Pg 3, Ln 7: You don't need to cite the URL a second time since it was listed on the prior page.

Pg 6, Ln 19-20: missing the word 'to' here: ". . .were often inaccessible due TO permitting or other restrictions."

Pg 9, Ln 17: Forgot a period at the end of the sentence.

Pg 10, ln 28: "levels" should be "mole fractions" for better clarity.

Pg 11, ln 1: You don't need the "roughly" since you define the time range and are using hourly data, unless the hourly window changes from day to day.

Pg 12, Ln 11: ". . .controlling the variability of CO2 (and CH4) observations. . ." should be ". . .controlling the variability of TRACE GAS observations. . ."

Pg 12, Ln 31: Again, "CO2 (and CH4)" should be "trace gas." Wind speed affects other trace gases as well (CO, NOx, etc).

Pg 15, Ln 9: The word "very" in unnecessary.

Pg 15, Ln 11-12: I wouldn't describe the selection criteria used by Thoning et al 1989 as "preliminary," its just what they used.

Pg 16, Ln 1: MBL should be defined as Marine Boundary Layer.

Pg 16, Ln 13: "roughly" is not needed since the exact latitudes are listed.

Pg 17, Ln 13: The "LA Basin" should probably be SCB for consistency.

Pg 18, Ln 20: There is an extra parenthesis next to Feng 2016.

Pg 22, Ln 4: There is an extra "and" on this line.

Pg 30, Ln 7-10: Way too many instances of the word "roughly" in this paragraph. Roughly is even duplicated on line 10! The authors could simply remove every instance of "roughly" and it would read better, or they could simply report the specific values and that would be fine also.

---

## Author Response (AR1)

**Response to the Editor: acp-2016-850**

On behalf of all co-authors, I would like to thank the three anonymous reviewers for detailed and thoughtful comments on the manuscript: Carbon Dioxide and Methane Measurements from the Los Angeles Megacity Carbon Project: 1. Calibration, Urban Enhancements, and Uncertainty Estimates" *by* K. R. Verhulst et al.  The reviewer comments were both thoughtful and insightful and helped improve the clarity and content of the manuscript, especially with regards to the topic of background estimation.  We also thank another reviewer for a short comment that helped clarify text in the manuscript related to the annual averages.  We have posted separate responses to comments made by Referees 1-3 online and incorporated the suggested changes into a revised version of the manuscript.  Attached below is a copy of the revised manuscript, with all changes highlighted using "track changes." Where applicable, we included typographical and stylistic changes, as well as additional references suggested by the reviewers.  We also conducted our own internal reviews, which resulted some in stylistic and typographical edits throughout the text that are also tracked here. Additionally, we were able to update our latest carbon dioxide and methane datasets up to June 2016, and these data are now included in Figures 2-4.  Finally, in Table 1 we included the location of an additional site that was installed in November 2016 (Canoga Park) in Table 1. Although we are not able to present results from this site at this time, the location will become part of the regular Los Angeles in situ network going forward and we thought it would be appropriate to include the coordinates with the list of other sites in Table 1.

[revised manuscript text omitted]

10  method; (3) Data selection criteria and curve fitting parameters for the CCGCRV software used to estimate background; (4) Estimates of epsilon (the slope component of the extrapolation uncertainty) for $CO_2$ and $CH_4$ based on laboratory measurements using CRDS analyzer units similar to those deployed in the field; (5) Uncertainty due to permeability of the Nafion drier determined from laboratory experiments; (6) Results from an Allan deviation analysis conducted using daily calibration runs from the La Jolla analyzer during January 2016; (7) Example plots

15  showing the $CO_2$ and $CH_4$ calibration baseline uncertainty using three possible time series of Picarro sensitivity for the standard tank measurements from the La Jolla site during January 2016; (8) Results for the uncertainty associated with background ($U_{BG}$) for the San Clemente Island, Victorville, and La Jolla estimates.

**1) Data Acquisition and Quality Control**

**1.1) Data acquisition**

[revised manuscript text omitted]

25   by Earth Networks based on recommendations from the LA Megacity Data Working Group, a team of scientists from NASA's Jet Propulsion Laboratory, Scripps Institution of Oceanography, the National Institute of Standards and Technology, and Earth Networks. Manual flags are applied on a case-by-case basis.  The decision to flag the data is usually based on information or observations that suggest instrument issues.  Many of these cases include technician site visits that require modifications to the plumbing on the instrument of calbox.  For example, when a

30   calibration standard is replaced or if an analyzer is removed and/or replaced during repair, room air may enter the instrument, which is not of scientific interest.  In some instances, we identify problematic by first looking for large deviations in the cavity pressure, sample pressure, inlet pressure, and/or in the measured mole fraction data and then comparing the data alongside notes from technician logs.  The numerous parameters monitored by GCWerks help narrow down the cause of anomalous observations. The N filtered parameter is also monitored and cases where a

35   large number of data have been filtered are analyzed in more detail to determine if additional manual flags are

required.  The manual flags are applied on the EN server to indicate those data that are not recommended for further scientific evaluation or interpretation within the scope of the project.

The corrected data is generated by GCWerks at 1-minute average intervals and generated as a .csv file for export and further analysis outside of the GCWerks framework. These .csv report files are uploaded to a primary Earth Networks/GCWerks server and are synced nightly and later used to compute the hourly average (Level 3) product.

We primarily discuss the 1-hour average $CO_2$ and $CH_4$ air observations in Sections 3 and 4, which are from individual inlet heights (for tower sites), and from a combination of the 4 corner inlets (for rooftop sites).  These hourly average data are a Level 3 product, which is averaged from the uncorrected (2-5 s) Picarro data.  For rooftops, we used a method similar to McKain et al. (2015) to calculate the 1-hour average air observations using only "upwind" observations (determined from the 1-minute average data).  Additionally, we used wind speed and direction observations to verify the "upwind" side of the building.  We constructed an "upwind" index for each 1-minute $CO_2$ or $CH_4$ observation.  The measurement was determined to be "upwind" if the building corner had the highest wind speed and the wind direction also corresponded to the same side of the building.

**2) Alternate calibration method**

An "Alternate Calibration Method" was explored using a linear fit between two tanks (one "near-ambient" tank and one ''high concentration" standard tank) where data was available. Each calibration run is used to derive a slope and intercept, which are then interpolated in time:

*Xcorr alternate* $= m*X'_{air} + b$ (Eq. S1)

where the slope m and intercept b.  The Alternate Calibration Method uses the high concentration standard to determine the slope, m, and intercept, b, while the default (single-point calibration) method assumes a zero reading at zero measurement. Therefore, the Alternate Method becomes equivalent to the default calibration method if b = 0 and the slope m does not vary with mole fraction, so that m = 1/S for all points. Both methods of calibration assume linearity, in that the slope m (or 1/S), is a constant over all mole fractions.

Air data from the LJO and VIC sites were corrected using the two methods to quantify the effect of different calibration methods on the final air mole fraction data. Both of these sites had limited measurements of a high mole fraction (span) tank available at the time of this study. Figures S2 and S3 show the difference in air data from the LJO analyzer corrected with the 2-point calibration method and the single-point calibration method for $CO_2$ (upper panels) and $CH_4$ (lower panels). Figure S3 shows similar results for air data collected from the VIC analyzer. Overall, the single-point calibration method underestimates the $CO_2$ levels by about 0.2 ppm out of 100 ppm and underestimates the $CH_4$ levels by about 6 ppb out of 6000 ppb, or about 1 part in 1000 for the LJO site (Figure S2). The results were similar when the same analysis was performed using air data from the VIC analyzer (Figure S3).

**3) Background selection criteria and curve fitting parameters for CCGCRV**

Our data selection criteria for $CO_2$ loosely follow the discussion in Thoning *et al.* (1989), but differ slightly for the LA sites. As described in the text, our data selection approach relies on several criteria: (1) a small degree of variability within a one hour period, and 2) small hour-to-hour variability, and (3) persistence of the first two conditions for several hours. Based on these criteria, we exclude observations that are impacted by local emissions or recirculation effects. The selection criteria for SCI were as follows: (1) First, check for stability of the $CO_2$ and $CH_4$ observations within 1-hour and retain measurements if the 1-hour SD is <0.3 ppm $CO_2$ and <3 ppb $CH_4$; (2) Next, find small hour-to-hour changes in $CO_2$ concentration and retain measurements if the hour-to-hour difference is less than 0.25 ppm $CO_2$. No hour-to-hour criteria were used for $CH_4$; (3) Finally, retain only those observations with several (6 or more) consecutive hours that meet criteria 1 and 2.

To determine the filter criteria, we first evaluated the standard deviation of the one hour average observations (Figure S4). During 2015, 70%, 42%, and 30% of the data had a one hour S.D. <0.3 ppm $CO_2$, 67%, 57%, and 42% of the data had a one hour S.D <3 ppb $CH_4$ filter criteria, and 60%, 35%, and 29% of the data met both criteria for the SCI, VIC, and LJO sites, respectively. We began by applying these criteria to all 3 sites since a significant fraction of the data were within these limits. Next, we chose the hour-to-hour stability cutoff (0.25 ppm $CO_2$) based on Thoning *et al.* (1989). For the final criteria, we performed several tests by setting the number of consecutive hours between 3 and 6 hours and analyzing the remaining observations.

We found that the LJO and VIC observations were most sensitive to the filter parameters, especially the hour-to-hour stability and number of consecutive hours (criteria 2 and 3). For VIC, we found that requiring 6 or more consecutive hours of stable conditions resulted in large data gaps over the entire season during summer months, making the background estimate highly uncertain during this period. After several adjustments to the filter criteria, we were able to reduce the gaps in the VIC background observations to <1 month by applying the following changes: 1) increasing the hour-to-hour stability from 0.3 ppm $CO_2$ to 0.5 ppm $CO_2$ and 2) decreasing the number of consecutive hours with stable conditions from six hours to four hours. For LJO, the original filter criteria did not produce large gaps (>1 month). Furthermore, increasing the allowable hour-to-hour stability or decreasing the number of consecutive hours resulted in a few anomalously high $CO_2$ and/or $CH_4$ observations being included in the result, which is unfavorable (and likely due to a persistent polluted air mass passing over the site rather than clean background air). For these reasons, we applied the same criteria at LJO and SCI.

We have considered possible impacts of PBL growth on the background analysis. As described in the main text, we use only nighttime flask samples for the MWO background estimate because this site is more sensitive to the LA Basin during daylight hours due to growth of the PBL and upslope winds. However, our filtering criteria for SCI, LJO, and VIC do not account for diurnal variations, e.g., due to variations in the planetary boundary layer height or due to potential daytime drawdown of $CO_2$ due to photosynthetic uptake. Initially, we made plots of the monthly average diurnal variability for the SCI, LJO, and VIC sites. However, it was not apparent how the diurnal cycle would aid in the interpretation of background because most of the time the diurnal changes at these sites are dominated by impacts from local emissions (especially at LJO and to a lesser extent at the other two background

sites due to outflow). At the marine background sites (LJO and SCI), it is the growth the marine boundary layer (MBL) rather than the PBL over the land, that is relevant to the interpretation of background. However, the MBL growth effect is most relevant when a site located very far off-shore, such that nighttime continental outflow is not present. Under these conditions, changes in the MBL with time of day are likely to be very small. The LJO is near sea level and is within the MBL, but is frequently impacted by local sources. The SCI site can be either within or above the MBL due to its elevation (~489 m asl), but is still occasionally impacted by continental outflow. For these reasons, we do not limit the background consideration to certain times of day. The agreement between the SCI and LJO marine background estimates (within ~±1 ppm $CO_2$ and ~±10 ppb $CH_4$) suggests that there is not a large gradient between the $CO_2$ and $CH_4$ levels in the surface MBL and above the MBL. In summary, for the SCI, LJO, and VIC background sites, our underlying assumption is that if the PBL (or MBL) grows, it will not further dilute the $CO_2$ or $CH_4$ levels or cause additional large variations if the site is truly sampling background conditions.

After applying the selection criteria respective to each site, the CCGCRV curve fitting software was used to estimate a "smooth curve" fit to the remaining observations (Thoning et al., 1989; http://www.esrl.noaa.gov/gmd/ccgg/mbl/crvfit/crvfit.html). The following fit parameters were used in CCGCRV: short-term cutoff filter=80 days, long-term cutoff filter=667 days, npoly=3, nharm=4. Data were fit iteratively, continually excluding outliers greater than ±2σ from the smooth curve fit until no more outliers could be removed. A multi-species filter was also applied, so that if either $CO_2$ or $CH_4$ were outliers from the smooth curve, then both observations were omitted. The MWO flask data were fit using a similar approach (but only using nighttime flask data to prevent potential contamination due to upslope winds). For MWO, the following fit parameters were used in CCGCRV: short-term cutoff filter=30 days, long-term cutoff filter=667 days, npoly=3, nharm=4, and the data were fit iteratively excluding outliers as described above. The full datasets, selected data and "smooth curve" results are shown in Figure 3 in the main text and a comparison of the final smooth curve results for each site is shown in Figure 4 (top panels). Figure S10 shows the same results with uncertainty estimates calculated as described in Section 6.2 of the main text. Overall, we have achieved a reasonable level of convergence between the background estimates for three sites with very different variability in $CO_2$ and $CH_4$ mole fractions. A metric of success exhibited by our results is that the background reference curve estimates agree within ~±1 ppm $CO_2$ and ~±10 ppb $CH_4$ for the marine sites (LJO and SCI) and continental sites (MWO and VIC, see Figures 4 and S10).

**4) Uncertainty due to Nafion drier permeability**

The laboratory experiments described here were performed at the Scripps Institution of Oceanography to estimate the uncertainty in the water vapor correction due to bias caused by the permeation of $CO_2$ and $CH_4$ across the membrane of the Nafion drier. We measured two dry standard tanks for 1200 seconds each, alternating one directly after the other. The measurement system setup was identical to those used at our field sites, with a Nafion dryer located upstream of the instrument. The water vapor concentrations at the start of the measurements were 0.095%, reflective of the Nafion dryer conditions during the ambient air measurements just prior to this experiment. As the measurements continued, the Nafion dryer gradually dried out, which reduced the permeation of $CO_2$ and $CH_4$ across the membrane. This effect leads to a small increase in the $CO_2$ and $CH_4$ levels measured on the Picarro

analyzer (Figure S6). In this experiment, we assume that other factors such as instrument drift are negligible over the duration of this experiment (approximately 16 hours).

The uncertainty due to the Nafion permeation effect is derived from the slope of measured $CO_2$, $CH_4$ concentrations against water vapor concentrations during our experiment, as shown in Figure S6. $CO_2$ concentrations are found to decrease at a ratio of -1.15 ppm per 1% change in water vapor concentration in the range of 0 to roughly 0.095%, while $CH_4$ concentrations are found to be small at a ratio of 0.029 ppb per 1% water vapor concentration change within a range of 0.03 to roughly 0.095%. As the water vapor concentrations in our field measurements lie within a range of 0.01±0.001%, we estimate the potential bias introduced by the 0.001% range in water vapor concentrations to be -0.0115 ppm for $CO_2$ and 0.000029 ppb for $CH_4$.

Note that while the permeability of $CH_4$ through the Nafion membrane is shown to change dramatically in water vapor concentrations lower than 0.03%, this effect can be effectively ignored for our purposes considering the range of water vapor concentrations measured at our sites.

Also, since the relationship between permeation through the Nafion membrane and water vapor concentration has been established, it is also possible to correct for this bias and report an uncertainty on the confidence of our understanding of this relationship. This correction may potentially be added in the future, which would further reduce the uncertainties due to this effect.

[Figure]

**Figure S1.** Diagram showing the standard gas-handling configuration for an Earth Networks greenhouse gas monitoring tower sampling from a single air inlet (see text). Figure and caption adapted from Welp et al. (2013).

[Figure]

**Figure S2.** "Alternate Calibration Method" plotted versus uncorrected $CO_2$ and $CH_4$ air sample measurements for the LJO site. A limited number of measurements of a high mole fraction $CO_2$ and $CH_4$ standard were available at the LJO field site between October 2015 and March 2016. "Method Difference" indicates the difference in the correction of the air data using a single-point relative to a 2-point calibration. Histograms show that the uncertainty associated with using a single-point calibration is <0.2 ppm $CO_2$ and <4 ppb $CH_4$ for air measurements collected during this period.

[Figure]

**Figure S3.** Same as Figure S2, except for the VIC site, which also had limited measurements of a high mole fraction $CO_2$ and $CH_4$ standard available at the time of this study. Histograms show that the uncertainty associated with using a single-point calibration is <0.15 ppm $CO_2$ and <4 ppb $CH_4$ for the majority of the air measurements collected during this period. Overall, the corrections are slightly smaller because the $CO_2$ and $CH_4$ enhancements at VIC are smaller relative to LJO.

[Figure]

**Figure S4:** Histograms of the standard deviation of hourly $CO_2$ (panels *a and b*) and $CH_4$ (panels *c and d*) observations from the SCI (blue), VIC (red), and LJO (grey) sites. Left panels show histograms for all data. Right panels show the same data with the x-axis truncated for values >1.5 ppm $CO_2$ and >5 ppb $CH_4$.

[Figure]

**Figure S5.** Estimates for epsilon ($\varepsilon$), the slope component of the extrapolation uncertainty ($U_{extrap}$) for $CO_2$ based on measurements from seven Picarro CRDS analyzer units. All calibrations were performed on the same suite of tanks at the NOAA/ESRL calibration laboratory. $Xassign_{cal}$ is the assigned value of the calibration standard on the WMO scales (in this case the tank with $CO_2$ value closest to 400 ppm). The calibration standard was used to correct the uncorrected measurements of other standard gases using Eq. 2 in the main text. $Xassign$ is the assigned value of the other standard tanks on the WMO scale (i.e., the span tanks with varying concentrations), and $Xcorr$ is the calibrated data. The slope of the residual ($Xcorr - Xassign$) is plotted as a function of the concentration difference between each of the standard tanks and the assigned calibration tank ($Xassign_{span} - Xassign_{cal}$), and is a measure of $\varepsilon$. All tanks were calibrated on the WMO/NOAA scales at the NOAA/ESRL laboratory. All Picarro analyzers shown here are similar models to those deployed in the LA network. The same suite of standard tanks was run on each analyzer prior to deployment for various field campaigns (with the exception of CFKBDS2008). CFKBDS2007a and CFKBDS2007b indicate two different calibrations of the same analyzer. All regressions are forced through zero. Error bars show the scale reproducibility (1σ) for the tank values reported by NOAA/ESRL (0.03 ppm $CO_2$; *Andrews et al.*, 2014 and *B. Hall*, personal communication).

[Figure]

**Figure S6.** Estimates of epsilon ($\varepsilon$), the slope component of the extrapolation uncertainty ($U_{extrap}$) for $CH_4$ using the same suite of tanks as in Figure S4. All regressions are forced through zero. Error bars show the scale reproducibility (1σ) for tank values reported by NOAA/ESRL (0.31 ppb; $CH_4$ Andrews et al., 2014).

[Figure]

**Figure S7.** Results of Nafion permeation experiment. A slight increase in the measured $CO_2$ and $CH_4$ levels (shown here as uncalibrated CRDS readings) was found as the Nafion membrane dried out, leading to less permeation across the membrane throughout the experiment. We assume that these changes in the measured concentrations are not due to any other factors such as instrument drift. For simplification, only one of the two tanks measured during the experiment is shown. For $CO_2$, the relationship between measured concentrations and water vapor concentration is derived for the complete water vapor concentration range. For $CH_4$ we only consider the range of 0.03 to 0.095% $H_2O$ since the permeation effect is different at lower water vapor concentrations, and the water vapor concentration for the field measurements in our network are within the $0.01\pm0.001$% range.

[Figure]

**Figure S8.** Allan deviation analysis from a subset of the daily calibration runs collected on the LJO analyzer during January 2016 (every 5[th] run is plotted for clarity). The results show that the characteristics of the noise in the analyzer vary with time. In general, the results for the calibrations are not all the same and do not fit a white-noise profile (indicated by the dashed line with slope of -1/2), indicating correlation in the noise at various longer time scales.

[Figure]

**Figure S9.** Example showing three possible time series of Picarro sensitivity for the standard tank measurement (upper panels) and the impact on estimates of calibration baseline uncertainty (lower panels). Results are shown for LJO data collected during January 2016. Upper panels: S is the sensitivity of the standard at the times when the reference tank was sampled (black points), which is calculated as the ratio of the measured analyzer mole fraction for the reference gas and the tank's assigned value (see text). The sensitivity of the standard is linearly interpolated in time, as shown by the black lines for $CO_2$ (left) and $CH_4$ (right). The traces, labelled S1 (blue lines) and S2 (red lines), show two alternate realizations of the analyzer sensitivity based on different interpolation methods (e.g. interpolating at points halfway between the sequential standard tank measurements, leaving out every other point). Lower panels: Calibration baseline uncertainty ($U_b$ and $U_{bmax}$) calculated for $CO_2$ (left) and $CH_4$ (right). $U_b$ reduces to zero at the times when the calibration gas was run because the tank value is measured at that time.

[Figure]

**Figure S10.** Background observations (black circles), smooth curve estimates (cyan lines) and uncertainty estimates (red dashed lines) for San Clemente Island (panels *a-b*), La Jolla (panels *c-d*), and Victorville (panels *e-f*) for $CO_2$ and $CH_4$. During 2015, the annual average uncertainty in the SCI smooth curve estimate is 1.4 ppm $CO_2$ and 11.9 ppb $CH_4$.

[Figure]

**Figure S11:** Boxplot of enhancements ($CO_2xs$ and $CH_4xs$) in the LA megacity during 2015. Results for $CO_2xs$ (upper panels) and $CH_4xs$ (lower panels) are shown for all hours (left panels) and mid-afternoon hours (12-16:00 LT, right panels). The sites are arranged by latitude from north to south (top to bottom): Victorville (VIC), Granada Hills (GRA), Ontario (ONT), University of Southern California (USC), Fullerton (FUL), Compton (COM), Irvine (IRV), San Clemente Island (SCI) and La Jolla (LJO). Boxes outline the $25^{th}$ and $75^{th}$ percentiles of the sample data, respectively and red vertical lines show the median value at each site. The maximum whisker length is specified as 1.0 times the interquartile range (i.e. $[q_3 + w*(q_3 - q_1)]$ and $[q_1 - w*(q_3 - q_1)]$, where $q_1$ and $q_3$ are the $25^{th}$ and $75^{th}$ percentiles and w=1.0). Red pluses (+) indicate enhancements greater (or less than) the maximum whisker length to show the full range of variability. (Note: Results for the ONT site include observations for September –December 2015 only, while all other results are annual averages. Results from the USC site are shown for the G2401 analyzer only).

[revised manuscript text omitted]